# Flavor-switchable scaffold for cultured meat with enhanced aromatic properties

Milae Lee [1,3], Woojin Choi [1,3], Jeong Min Lee [2], Seung Tae Lee[2], Won-Gun Koh[1] & Jinkee Hong [1] ✉

Cultured meat is emerging as a new type of food that can provide animal protein in a sustainable way. Many previous studies employed various types of scaffolds to develop cultured meat with similar properties to slaughtered meat. However, important properties such as flavor were not discussed, even though they determine the quality of food. Flavor characteristics vary dramatically depending on the amount and types of amino acids and sugars that produce volatile compounds through the Maillard reaction upon cooking. In this study, a flavor-switchable scaffold is developed to release meaty flavor compounds only upon cooking temperature mimicking the Maillard reaction of slaughtered meat. By introducing a switchable flavor compound (SFC) into a gelatin-based hydrogel, we fabricate a functional scaffold that can enhance the aromatic properties of cultured meat. The temperature-responsive SFC stably remains in the scaffold during the cell culture period and can be released at the cooking temperature. Surprisingly, cultured meat fabricated with this flavor-switchable scaffold exhibits a flavor pattern similar to that of beef. This research suggests a strategy to develop cultured meat with enhanced sensorial characteristics by developing a functional scaffold which can mimic the natural cooking flavors of conventional meat.

Cultured meat has been attracting attention as a sustainable meat that can minimize animal slaughter and environmental pollutants. Many previous studies on cultured meat have focused on developing scaffolds to mimic the shape of meat. For example, Jeong et al.[1] reported a strategy to fabricate steak-type cultured meat using a three-dimensional (3D) printing technique. Moreover, Liu et al.[2] developed a cellularized microtissue to produce meatball-type cultured meat. Most previous studies have focused on strategies to mimic the properties of meat depending on the cell mass and differentiation. However, several food-related characteristics of meat, such as texture and flavor, are associated with blood and various biological tissues such as muscle, fat, and connective tissues. Because research on cultured meat is still at the stage of cell culture scale rather than tissue culture level[3–6], there are limitations in mimicking the organoleptic properties of meat.

The flavor is the key organoleptic property that determines the quality of meat[7]. In particular, we recognize the flavor components of meat generated during cooking, which include aldehydes, alcohols, and sulfur-containing compounds. These flavor compounds are produced when peptides react with reduced sugar at high temperatures (above 150 °C), in the so-called Maillard reaction[8]. Depending on the amino acid types and amount of proteins and sugars, different Maillard reaction products are formed, eventually determining the overall flavor of cooked meat[9]. The difference in amino acid profiles between in vitro tissues and traditional meat presents challenges in mimicking the Maillard flavor of traditional meat in the field of cultured meat[10,11]. Therefore, mimicking the Maillard flavor in cultured meat is also required to increase the sensorial similarity between cultured meat and traditional meat. Recently, many papers reported to improvement the flavor characteristics of cultured meat. For example, Joo et al.

[1]Department of Chemical & Biomolecular Engineering, College of Engineering, Yonsei University, Seodaemun-gu, Seoul, Republic of Korea. [2]Department of Applied Animal Science, Kangwon National University, Chuncheon-si, Gangwon-do, Republic of Korea. [3]These authors contributed equally: Milae Lee, Woojin Choi. ✉e-mail: jinkee.hong@yonsei.ac.kr

compared the taste of cultured muscle tissue obtained from chicken satellite cells to that of traditional chicken[10]. However, they revealed that it is challenging to mimic the taste of traditional meat in solely cultured tissue without any treatment due to the different amino acid composition. Liu et al. also reported that cultured fat had fruity and creamy flavor characteristics due to the lipid oxidation of fatty acids formed in cultured fat[12]. Recently, our group revealed that the enhanced proliferation and differentiation of murine and bovine myoblasts could induce relatively more meat-like flavors in cultured meat[5,13]. Nevertheless, for cultured meat to present the flavor characteristics of well-cooked meat products, a strategy to generate the Maillard reaction compounds during cooking is still needed.

The concerns regarding flavor are not limited to cultured meat, and synthetic flavor compounds have recently been employed to optimize the food quality in the conventional food industry. For instance, among the various Maillard reaction products formed during meat cooking, sulfur-containing compounds such as furfuryl mercaptan, featuring roasted meat-like flavors, have been introduced to produce the unique aroma of meat[14,15]. However, these synthetic flavor compounds are volatile, and can thus rapidly diffuse into air[16]. The problem of the burst release of flavor compounds becomes significant in cultured meat, owing to its fabrication process. The long cell culture period could accelerate the loss of flavor, thus leaving no residual flavor compounds within prepared cultured meat. The total cell culture period was 15 days in this work, and it may take 28 days or more for cell proliferation and differentiation depending on the culture system[17,18]. Therefore, a material science approach to generate the flavor compounds during the cooking process is essential.

In this study, we develop a cultured meat that generates grilled beef flavors upon cooking. In particular, we develop a switchable flavor compound (SFC) that can release the conjugated flavor group (which is the Maillard reaction compound) upon heated at the cooking temperature, 150 °C (Fig. 1a). The SFC mainly consists of flavor group and two binding groups. The flavor group is the volatile compound with meaty flavor which is furfuryl mercaptan in this research. Furfuryl mercaptan (IUPAC nomenclature: furan-2-ylmethanethiol) was chosen because it is known as a Maillard reaction product formed from cooked beef[19,20]. Moreover, the thiol-end group of furfuryl mercaptan enables the formation of the thermo-responsive disulfide bond. The binding group of SFC is the functional group which can react with methacryloyl, the binding group of the gelatin backbone. After introducing SFC into gelatin methacryloyl (GelMA), a three-dimensional and temperature-responsively beef flavoring scaffold is fabricated. In other words, this scaffold is composed of SFC introduced GelMA. Cultured meat with the switchable flavor compound (CM + SFC) is produced after inducing differentiation of bovine primary myoblasts in the scaffold (Fig. 1b). Because of the covalent bond between the binding group of SFC and GelMA, SFC is stably conjugated in the scaffold during cell culture condition. Then, it was hypothesized that the SFC can release the flavor group upon heating due to the dynamic disulfide exchange of the flavor group. Eventually, CM + SFC can replicate the Maillard reaction of conventional meat (Fig. 1c). When cultured meat is cooked at the Maillard reaction temperature, 150 °C, various flavor compounds are formed in our study. The volatile compounds were classified by two main flavor notes (Fig. 1d). Volatile compounds with fishy and pungent flavors are classified as off-flavors whereas the compounds with meaty, sulfurous, almond-like, floral, fatty, and fruity flavors were classified as pleasant flavors[21–26]. In this research, the Maillard reaction-mimicking scaffold can enrich the flavor characteristics of cultured meat by increasing the ratio of volatile compounds with pleasant flavor. This study focuses on an under-explored but important aspect of cultured meat production. We believe that the present strategy can contribute to the production of cultured meat by bridging the gap between the organoleptic properties of cultured meat and those of slaughtered meat.

## Results

### Fabrication of flavor-switchable scaffold

The switchable flavor compound (SFC) capable of generating the Maillard reaction products in response to heating and cooking was designed. Figure 2a shows the chemical structure of the SFC involving three distinct functional moieties ($R_1$, $R_2$, and $R_3$). $R_1$ and $R_2$ were responsible for robust binding with the gelatin matrix, while $R_3$ contributed to temperature-responsive flavoring function. In the case of the thermal responsive group ($R_3$), a disulfide bond to the roasted meat-flavored Maillard reaction product, furfuryl mercaptan, was introduced (Supplementary Fig. 1a). In particular, the disulfide bonds exhibit temperature responsivity through the disulfide exchange process[27,28]. The Raman signals at 486 cm⁻¹, 515 cm⁻¹, and 2570 cm⁻¹ confirm the disulfide bridge formation at the thiol-end of furfuryl mercaptan (Supplementary Fig. 1b). Two methacrylate groups were connected in the positions of $R_1$ and $R_2$ for serving as binding groups with gelatin methacryloyl (Supplementary Fig. 2a)[29]. As shown in Supplementary Fig. 2b, the ¹H nuclear magnetic resonance (NMR) spectra of SFC yielded the methacrylate signals of $R_1$, $R_2$, furan signals of $R_3$, and urethane bond signals of $R_1$-$R_3$. This result confirmed a single SFC involved the flavor group and binding groups.

As shown in Fig. 2b, the temperature responsiveness of the SFC was investigated by heating at 37 °C, 80 °C, or 150 °C up to 24 h in a closed system. In the 37 °C heating case, the SFC hardly yielded any noticeable signals. Interestingly, a weak absorbance signal started to appear after heating at 80 °C and this absorbance signal became obvious by 150 °C heating. In particular, the increasing peaks near 335 nm indicated the enhanced mobility of the furan group of furfuryl mercaptan, the Maillard reaction product[30]. Hence, these results confirm that the SFC exhibited temperature-responsive flavoring via the release of the Maillard reaction product. The release of the Maillard reaction product exhibited a positive correlation with the heating temperature. Moreover, the onset temperature for flavoring was 80 °C and this thermos-responsivity became saturated after 12 h (Supplementary Fig. 3).

It was evaluated whether the Maillard reaction product was stably bound to the SFC during a prolonged cell cultured period at 37 °C. To this end, the SFC and pristine furfuryl mercaptan without disulfide bond were incubated at 37 °C in an open system with sufficient air circulation. As shown in Supplementary Fig. 4a, b, pure furfuryl mercaptan evaporated rapidly from day 1 of the experiment. In particular, its residual weights were 60.9% and 6.76% after 3 and 14 days of 37 °C incubation, respectively. In contrast, SFC maintained 93.8% of its weight even at the same end point. Moreover, ¹H NMR experiments were conducted in situ to prove the weight variation-based finding about the stability of SFC under the open system with 37 °C heating below the onset point. Remarkably, ¹H NMR spectra showed that the SFC retained the features of its chemical structure during 14 days of incubation at 37 °C (Supplementary Fig. 4c). These results show that the SFC selectively released the flavor compound upon cooking and the flavor compound was non-volatile during the cultured meat production.

After confirming the stability of the SFC itself, the flavor stability of the gelatin-based hydrogel containing SFC(Gel+SFC) was evaluated. The hydrogel without SFC (Gel-SFC) was also produced as a control group. Our strategy aimed to induce the strong interaction between the SFC and the scaffold, and then selectively release the SFC-derived Maillard reaction product upon heating. Accordingly, the SFC was introduced into the gelatin-based hydrogel matrix based on the robust covalent bonds. In particular, the methacrylate of SFC's binding group was connected with the methacrylate of gelatin methacryloyl through an ultraviolet (UV)-based radical reaction. To demonstrate the enhanced flavor stability within the hydrogel achieved through our strategy, a hydrogel just physically mixed with pure furfuryl mercaptan (Gel + FM) was also produced. Namely, this Gel + FM hardly presents any covalent bonds between the gelatin matrix and FM.

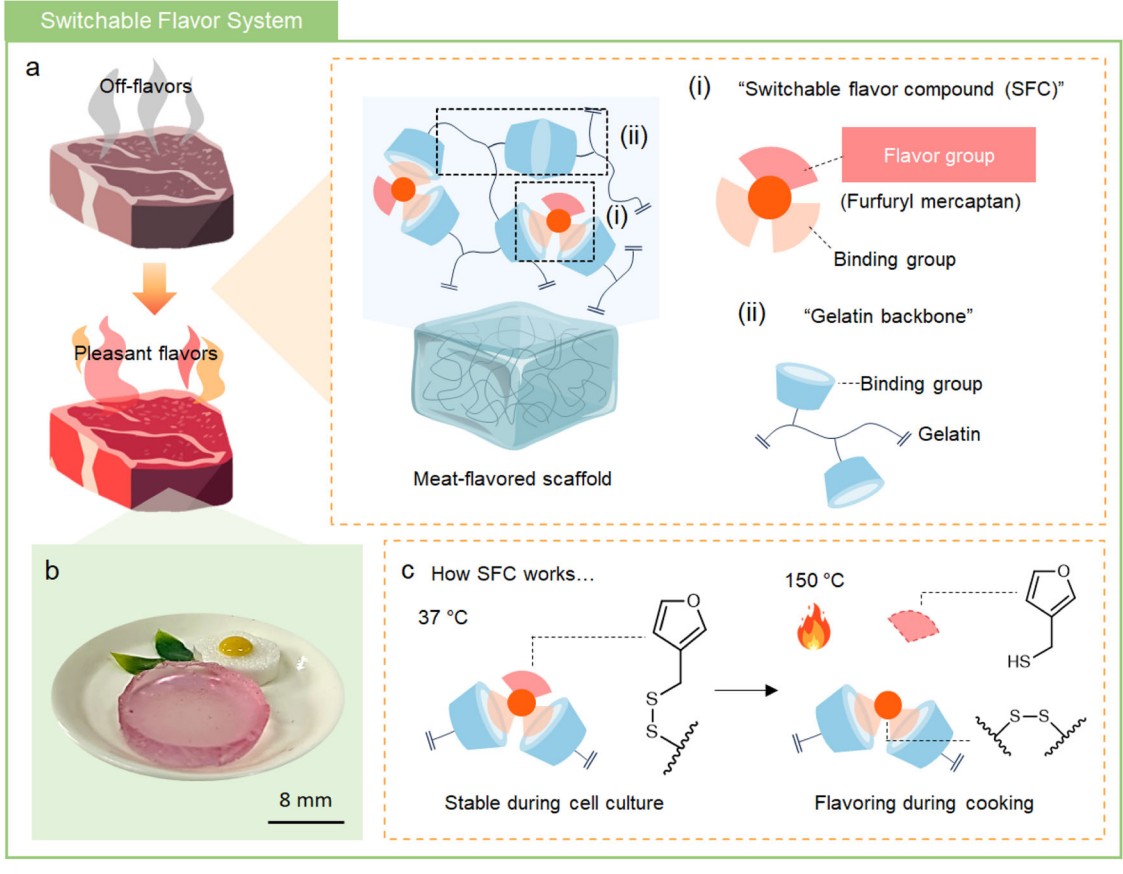

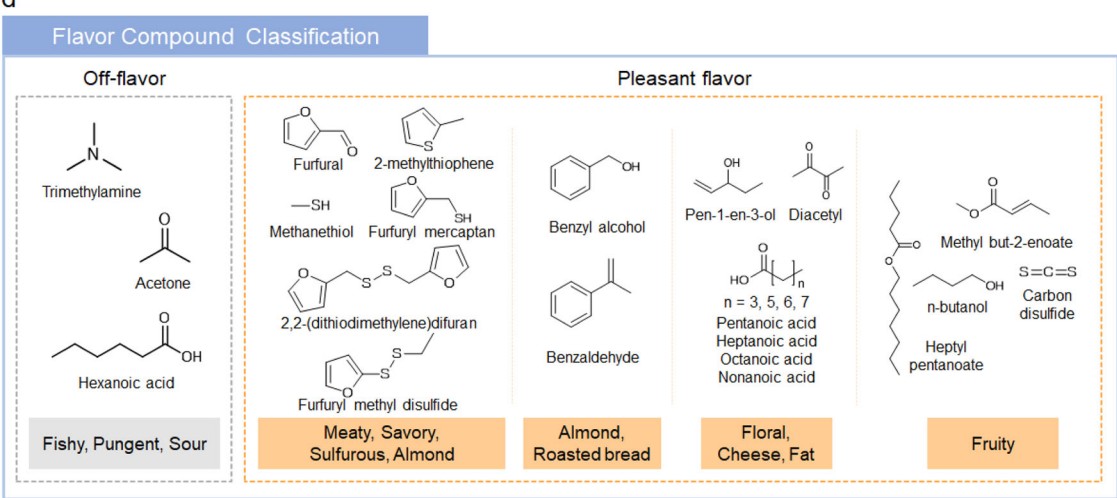

**Fig. 1 | Schematic illustration of switchable flavor system. a** Illustrative description of scaffold structure conjugated with switchable flavor compound (SFC). **b** Photograph of cultured meat fabricated using the scaffold with SFC (CM + SFC). Scale bar: 8 mm. **c** Illustration explaining the mechanism of the SFC system depending on temperature. **d** Classification of the flavor compounds analyzed in this study.

Images of Gel-SFC, Gel + SFC, and Gel + FM can be identified with the illustration representing the structure of each group in Fig. 2c. The yellow color of Gel + FM is due to the color of furfuryl mercaptan. Then, the flavor stability of the hydrogels at aqueous conditions was evaluated by gas chromatography–mass spectrometry (GC–MS) after immersing them in distilled water for 15 days, corresponding to the myoblast culture period (Fig. 2d and Supplementary Fig. 5). To assess the flavor-releasing capability of the hydrogel as a function of the temperature, the concentration of the volatile compounds from each group at room temperature and at the Maillard reaction temperature (150 °C) was measured by GC–MS. The flavor profiles of the detected

volatile compounds were assigned by the flavor library of the Flavor and Extract Manufacturers Association (FEMA) of the United States database (Supplementary Table 1). Then, the volatile compounds with flavor profiles assigned were classified into two groups: Pleasant flavor and off-flavor. This classification was based on the flavor classification in Fig. 1. The volatile compounds with no flavor profile are classified as non-flavor compounds. Also, the specific flavor notes that were identified by the FEMA library are shown in the pie chart.

Before heating, the concentration of the compounds with off-flavor was dominant for Gel-SFC. The concentration of these compounds was decreased whereas the concentration of volatile

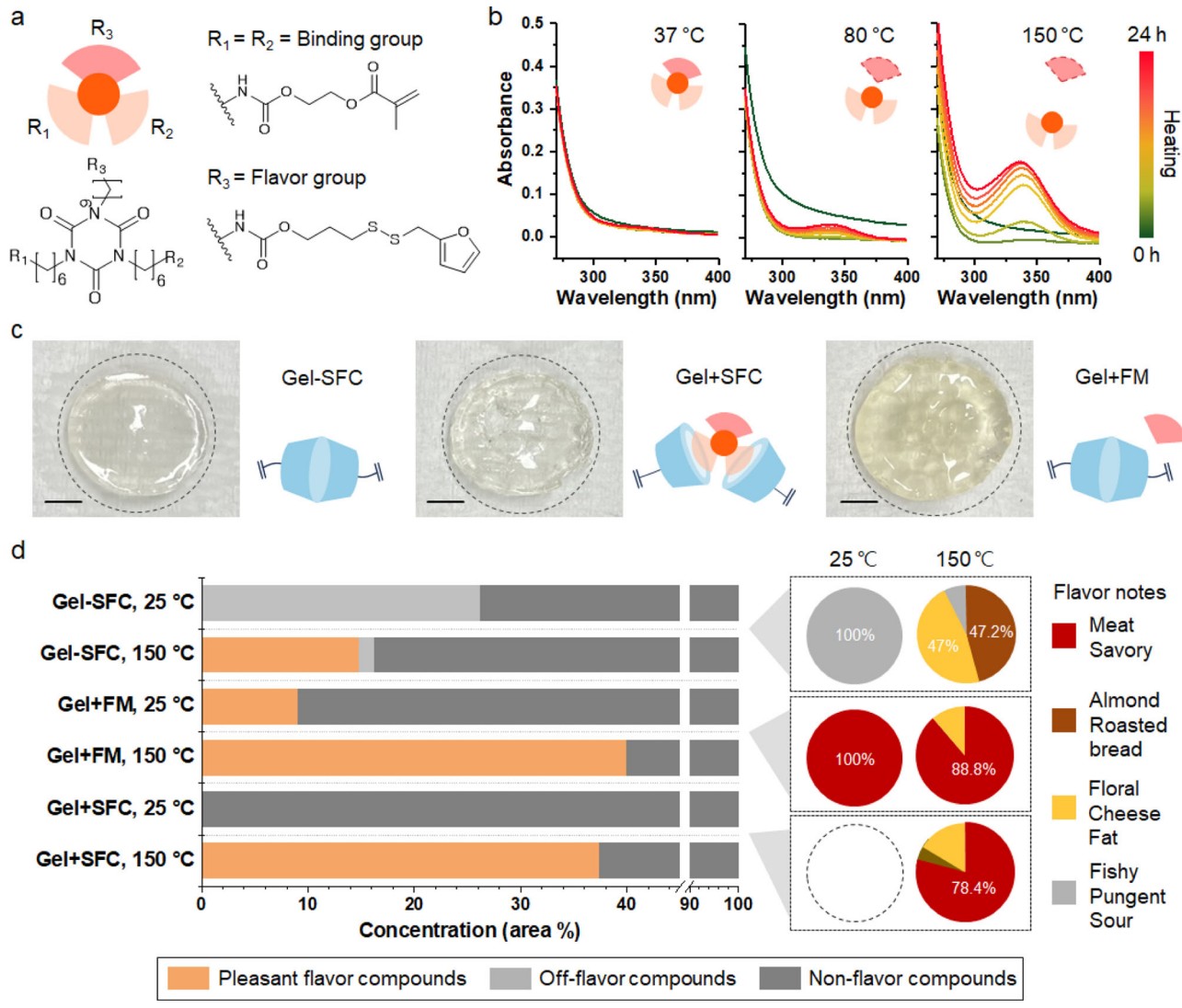

**Fig. 2 | Synthesis of SFC and thermal responsivity evaluation. a** Chemical structure of the switchable flavor compound (SFC) involving two binding groups (R₁, R₂) with methacrylate end and one flavor group (R₃) with a thermal responsive disulfide bridge. **b** Ultraviolet-visible (UV-Vis) spectra of SFC to monitor the mobility of furfuryl mercaptan upon heating of SFC. The peak at 335 nm indicates the mobility of the furan group of furfuryl mercaptan thermal-responsively generated from SFC. The intervals for heating are 0 min, 10 min, 1, 4, 7, 12, and 24 h. **c** Photographs of the hydrogel without SFC (Gel-SFC), hydrogel with SFC (Gel+SFC), and hydrogel mixed with pure furfuryl mercaptan (Gel+FM) with illustrations of

their network structure. Gel+SFC features the robust covalent bonds between the gelatin matrix and SFC. Meanwhile, Gel+FM exhibits a weak interaction between the gelatin matrix and furfuryl mercaptan. Scale bars: 0.4 cm (**d**) Flavor analysis of hydrogels before and after heating at Maillard temperature, 150 °C. All samples are pre-incubated in the distilled water for 15 days. The pie chart presents the ratio of the flavor compounds classified according to the specific flavor notes. Non-flavor compounds are excluded in the pie chart and each flavor note is represented by a different color (*n* = 3). Source data are provided as a Source Data file.

compounds with pleasant flavor was increased after heating. Nevertheless, the off-flavor compounds still accounted for approximately 1.4% in Gel-SFC. Also, the concentration of the flavor compounds among total volatile compounds detected from Gel-SFC was decreased upon heating. For Gel+SFC, the increase in concentration of pleasant flavor compounds after heating was confirmed which indicates that the introduction of SFC can provide pleasant flavor characteristics to the scaffold. The temperature-responsive release of the flavor compounds was only confirmed in Gel + SFC. At room temperature, no flavor compounds were detected from Gel + SFC. Upon heating at the Maillard reaction temperature, high levels of the flavor compounds with pleasant flavor notes such as meat, savory, almond, roasted bread, floral, cheese, and fat were detected from Gel + SFC. On the other hand, the flavor compound release before heating was detected for the Gel+FM group, indicating that flavor loss can occur during the cell culture period. These results confirm that the SFC can contribute to

the controlled release of the meaty flavor compounds from the scaffold, eventually enabling the fabrication of flavor-rich cultured meat.

## Cell culture on scaffold for flavor-enhanced cultured meat production

After confirming the stability of the SFC and the temperature-responsive volatilization behavior of aroma compounds, the hydrogels were lyophilized before cell culture. Then, primary bovine myoblasts were cultured on the scaffold to assess the cell proliferation and differentiation behavior. After 15 days of cell culture on the scaffold, cultured meat (CM) was fabricated and cooked at the Maillard reaction temperature (150 °C) to volatilize the meaty flavor compound of SFC (Fig. 3a). The cell-cultured groups using Gel-SFC and Gel + SFC were denoted as cultured meat without SFC (CM-SFC) and cultured meat with SFC (CM + SFC), respectively. CM-SFC was used as the control group to understand the influence of SFC on cell viability and myotube

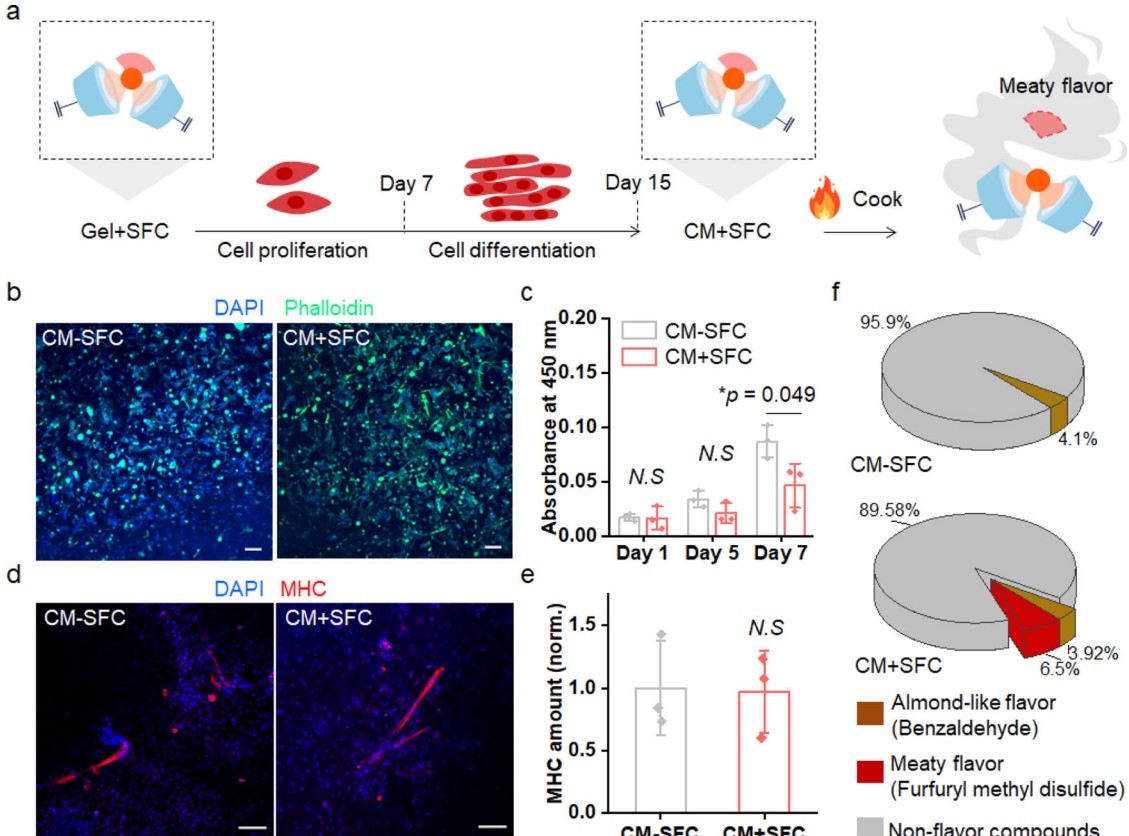

**Fig. 3 | Biological evaluation and flavor analysis of cell-cultured scaffold.**
**a** Illustration of the flavor enriching process of the switchable flavor compound (SFC) under the cultured meat fabrication process. **b** Immunofluorescence images of proliferated myoblasts on each group. Scale bars: 100 μm. **c** Cell viability on day 1, day 5, and day 7 using CCK-8 assay kit (mean ± SD, $n$ = 3 independent experiments, One-way ANOVA with Tukey method). Gray color indicates cultured meat without SFC (CM-SFC) and red color indicates cultured meat with SFC (CM + SFC). **d** Confocal images showing myosin heavy chain (MHC) and nuclei immunostained with MF20 (red) and DAPI (blue), respectively. Scale bars: 100 μm. **e** Quantitative assessment of MHC amount by bovine myosin-1 enzyme-linked immunosorbent assay (ELISA) (mean ± SD, $n$ = 3 independent experiments, One-way ANOVA with Tukey method). The MHC amount of the CM + SFC is normalized to that of the CM-SFC. **f** Assessment of volatile compounds in CM-SFC and CM + SFC after heating at 150 °C ($n$ = 3). Source data are provided as a Source Data file.

formation. First, immunostaining of actin filaments and nuclei of the cells was performed on day 7 of proliferation (Fig. 3b). The morphology and distribution of the attached cells were not significantly different between the scaffolds. To quantitatively compare the cell viabilities of the scaffolds, a cell counting kit-8 (CCK-8) assay was performed for CM-SFC and CM + SFC on the 1st, 5th, and 7th day of proliferation (Fig. 3c). The cell viability on day 7 was lower in CM + SFC compared to that of CM-SFC. However, the absorbance value did not decrease from day 1 to day 7 in CM + SFC. These results indicate that rather than cell death in CM + SFC, the proliferation rate was slow in CM + SFC, compared to that of CM-SFC. To further study this difference in cell viability between CM-SFC and CM + SFC, the swelling degree and compressive strength of each group were compared (Supplementary Fig. 6 and Supplementary Fig. 7). The radical-based covalent linkage between methacrylate of GelMA and the binding groups of SFC can decrease the crosslinking density of GelMA backbone which can eventually affect the swelling capacity and mechanical properties of the scaffold. Because the compressive modulus of a scaffold affects long-term cell proliferation, it was hypothesized that the lower cell viability of CM + SFC on day 7 is due to the lower compressive modulus of the scaffold[31]. Therefore, the swelling degree and compressive modulus of each scaffold were evaluated. As expected, the swelling capacity of Gel + SFC was higher whereas the compressive modulus was slightly lower, compared to those of Gel-SFC. The scaffold stiffness can be one of the factors that slow down the cell growth in Gel + SFC. However, further investigation is required to determine whether the slower cell growth in Gel + SFC is due to the chemistry. After evaluating cell proliferation, differentiation was induced by replacing the growth medium with the differentiation medium.

Myogenic differentiation was induced on day 7 of proliferation and then continued for another 8 days. On the last day of myogenesis, myosin heavy chain (MHC) immunostaining was performed to compare the degrees of myotube formation of the different groups (Fig. 3d). Branched myotubes were observed in both samples. Furthermore, the MHC amount per scaffold was quantified by bovine myosin-1 enzyme-linked immunosorbent assay (ELISA), which showed similar levels of MHC in CM-SFC and CM + SFC (Fig. 3e). The results of these biological tests indicate that the SFC introduction in the scaffold did not inhibit cellular functions.

After myogenic differentiation, flavor analysis was performed after heating the scaffolds at 150 °C, to compare the flavor profiles corresponding to the different scaffolds (Fig. 3f). Only benzaldehyde with almond-like flavor was detected from CM-SFC. On the other hand, meaty flavor was confirmed in CM + SFC, due to the formation of furfuryl methyl disulfide derived from the volatilization of the conjugated flavor compounds. These results indicate that SFC can provide a meat-like flavor profile to cultured meat.

## Variation of switchable flavor compounds to regulate the flavor pattern of cultured meat
The results of Figs. 2 and 3 confirmed that the Maillard flavor compounds in cultured meat with SFC were stably released upon heating.

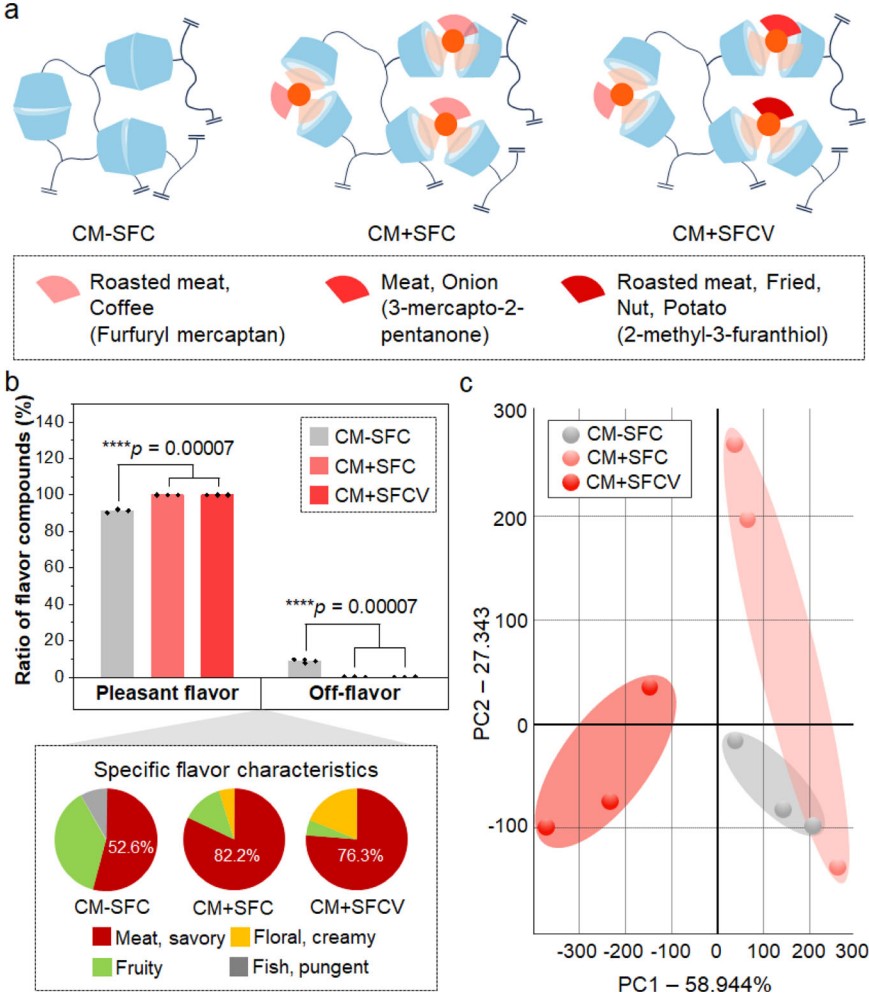

**Fig. 4 | Electronic nose analysis of cultured meat specimens. a** Illustration of cultured meat specimens with different scaffolds: cultured meat without SFC (CM-SFC), cultured meat with SFC (CM + SFC), and cultured meat with flavor variated SFC (CM + SFCV). **b** Ratio of the flavor compounds detected from CM-SFC, CM + SFC, and CM + SFCV (mean ± SD, $n = 3$ independent experiments, One-way ANOVA with Tukey method). The pie chart is shown to present the specific flavor profiles of each group. Flavor notes are presented with different colors. **c** Principal component analysis (PCA) of the flavor compounds of each group (discrimination index = 90, $n = 3$). Source data are provided as a Source Data file.

In conventional meat, various Maillard reaction products with diverse flavor profiles are formed, rather than a single flavor compound. Therefore, the range of SFC was expanded with three different Maillard reaction molecules to verify if the SFC system can be applied not only to single-flavor compounds but also to various flavor compounds (Fig. 4a). Specifically, 3-mercapto-2-pentanone and 2-methyl-3-furanthiol and furfuryl mercaptan were introduced into SFC. 3-mercapto-2-pentanone is also a sulfur compound with meat, onion-like flavors, whereas 2-methyl-3-furanthiol is known to possess fried, nut, and roasted meat-like flavors. Furthermore, the thiol-ends of 3-mercapto-2-pentanone and 2-methyl-3-furanthiol contributed to preparing the thermo-responsive disulfide linkage, identical to the thiol-end of furfuryl mercaptan. It was hypothesized that diversifying the flavor agents in SFCs can mimic the complex Maillard reaction of conventional meat in cultured meat systems. Therefore, the SFC with three kinds of flavor compounds was synthesized to introduce different Maillard flavor compounds into the scaffold (Supplementary Fig. 8). These three types of SFCs were then introduced into the GelMA hydrogel in the same proportion. Then, cultured meat with flavor variated SFC (CM + SFCV) was obtained after culturing bovine myoblasts on scaffolds with the three SFCs.

CM-SFC, CM + SFC, and CM + SFCV were cooked at 150 °C for 5 min before the flavor evaluation using an electronic nose. In Fig. 4b

and Supplementary Table 2, the flavor characteristics of CM-SFC, CM + SFC, and CM + SFCV were assessed after heating at 150 °C by electronic nose. Pleasant flavor and off-flavor were compared, and then the ratio of specific flavor notes were analyzed for each group. Compared to the results in Fig. 3f, different and more various flavor compounds were detected due to the different polarity of the used columns in the electronic nose performance. Here, non-polar and mild-polar volatile compounds were detected whereas polar volatile compounds were detected in Fig. 3. As a result, the volatile compounds with off-flavor (fish, pungent flavor) which were classified based on the flavor compound classification of Fig. 1 were only detected from CM-SFC. On the other hand, the volatile compounds with pleasant flavors were detected from all groups but highly detected from CM + SFC and CM + SFCV. The ratio of the compounds with meaty flavor was higher in CM + SFC and CM + SFCV compared to that of CM-SFC. Especially, the flavor characteristic of CM + SFCV was most similar to that of traditional beef among the three cultured meat specimens (Supplementary Fig. 9). The flavor ratio of the flavor compounds detected according to the Maillard reaction of beef was high in the following order: Meat, floral and creamy, and fruity. This pattern was also confirmed in CM + SFCV. To analyze the flavor similarity between the groups, the principal component analysis (PCA) was performed (Fig. 4c). As a result, it was confirmed that the SFC and SFCV affect the

flavor characteristics of the cultured meat. PCA was also performed to demonstrate the similarities in flavor characteristics between conventional beef and cultured meat specimens. Also, the flavor difference was identified according to the distance between the area of each cultured meat group and the area of traditional beef shown in the PCA graph (Supplementary Fig. 10). The results show that CM + SFC had a more similar flavor to beef compared to CM-SFC, owing to the SFC volatilization upon heating. CM + SFCV had the most similar flavor pattern to beef among the three cultured meat groups, owing to the introduction of various Maillard reaction compounds.

These results indicate that the strategy of introducing switchable flavor compounds can control the flavor characteristics of cultured meat. Moreover, this approach is not limited to a specific compound: the introduction of various flavor compounds can mimic the complex flavor pattern of conventional meat.

## Discussion

Volatile small molecules, representatively, gas molecules, volatile organic compounds, and flavor compounds, have attracted considerable attention in broad research and application fields: the capture of environmentally impacting carbon dioxide gas[32], electrochemical production of hydrogen gas as a green energy resource[33], deactivation of volatile chemical warfare agents[34], disease diagnosis by exhaled breath gas[35,36], vascular disease treatment by nitric oxide gas[37,38], and carbon monoxide gas-based gastrointestinal inflammation suppression[39]. However, given the utilization in open systems, such as air-permeable cell culture processes, these volatile small molecules suffer from their fast mass transport rate. Therefore, strategies for stable entrapment and controllable release of volatile small molecules have been developed. The emulsion and perfluorocarbon were representatives for physical retention[40,41]. Meanwhile, because these physical capsules degrade rapidly, the selective generation of the volatile small molecules is a challenge. Although the chemical binding promises significant durability, it requires external energies (*e.g.*, proteolysis[36], magnetic force[42], and catalysis[43]) to trigger the volatile compound release on demand.

In this study, we employed robust covalent bonds to secure the flavor stability during a prolonged open system process for cell proliferation and differentiation. Moreover, we devised thermal energy as a desirable trigger resource, given the heating-based cooking and the positive correlation between temperature and diffusion coefficient of the Maillard reaction product. In particular, among the dynamic covalent bonds, such as imine[44], boronate–ester[45], and hydrazine[46] bonds, the disulfide bond was adopted considering its temperature responsivity[27,28]. Accordingly, this thermo-responsively Maillard reaction product releasing system, so-called SFC, enabled the cell scaffold-based replication of the Maillard reaction of slaughtered meat. Remarkably, the SFC remained stable in the scaffold during the long-term cell culture period. Moreover, it released the flavor compounds such as furfuryl mercaptan selectively at the cooking temperature. At first, we expected only the flavor compounds included in the SFC, furfuryl mercaptan, to be released. However, more diverse flavor compounds with meaty and savory flavor notes were found at 150 °C. These unexpected products would be due to the hydrogen peroxide, an oxidation reagent for forming the disulfide bond in the Maillard reaction product. The hydrogen peroxide would be residual in the CM + SFC with a concentration of 0.386 mM (which is within the food-grade concentration range). When the furfuryl mercaptan is generated upon cooking, the oxidative reactivity of hydrogen peroxide can induce the disulfide bond within the furfuryl mercaptan, forming the 2,2-(dithiodimethylene)difuran[47]. Moreover, the hydrogen peroxide could generate the hydroxyl or hydroperoxyl radicals upon heating[48]. Hence, these radicals could induce the bond scission and thereby result in diverse volatile compounds with meaty and savory flavor notes, such as 2-methylthiophene or furfuryl methyl disulfide. Finally,

we fabricated the flavor-enriched cultured meat after carrying out cell proliferation and differentiation of bovine myoblasts on the SFC-introduced scaffold. Because this SFC exhibited savory and meaty flavors upon volatilization and also generated various flavor molecules by dynamic disulfide exchange, the flavor pattern of cultured meat became similar to that of slaughtered beef.

However, irrelevant to these remarkable advances, this study still encounters a critical limitation. It should be cautious to regard the used reagents as food-grade, although they are proven to be biocompatible. For instance, the methacrylic anhydride is generally employed to functionalize the methacrylate in the Food and Drug Administration (FDA)-approved proteins (such as gelatin[49–51], silk fibroin[52]) and to prepare biocompatible scaffolds via photo-crosslinking. However, until recently, no sub-type of GelMA was granted by the FDA for food purposes. Namely, although the biocompatibility of CM + SFC was identified by the primary bovine myoblast experiments, it is still distant from establishing its food safety. Nevertheless, the proposed flavoring strategy yields the potential to be implemented solely by food-grade chemicals. In particular, the enzymatic gelatin crosslink using FDA-approved transglutaminase could be a promising approach to realize the robust binding of Maillard reaction products using only food-grade chemicals[13]. Moreover, the thermo-responsivity could be fulfilled by the natural disulfide chemistry of cysteine amino acids[53]. In conclusion, beyond the current stage limitation related to food-grade chemistry, we anticipate that the interesting flavoring strategy developed in this study will contribute to the production of cultured meat that reaches the organoleptic properties of conventional meat.

Here, we developed the SFC system to introduce meaty flavor compounds in cultured meat. Also, we further develop the strategy to diversify the flavor compounds of SFC to mimic the flavor properties of traditional meat in Fig. 4. However, there is still a significant difference between the flavor characteristics of cultured meat with SFC and that of traditional meat even with the diversification of flavor compounds in SFC. To overcome this limitation, we think two future strategies would be possible: (1) development of SFC with more diverse flavor compounds than SFCV, (2) convergence of the SFC system with the strategy to increase the cell content in the scaffold to take the advantages of cell-derived flavors. As traditional meat creates dozens of Maillard reaction compounds as flavor compounds when cooked, three kinds of flavor molecules in SFCV would still be insufficient to mimic the flavor of traditional meat perfectly. Therefore, developing the SFC system to admit dozens of Maillard compounds would be one of the future directions. Also, we only focused on the flavor characteristics depending on SFC in this paper. However, we expect the synergistic effect of SFC and cultured cells if three-dimensional culture strategies to increase cell content are further applied to this study. Through our previous research, we have reported that proteins synthesized by cells participate in the Maillard reaction and affect the flavor characteristics of cultured meat[13,54]. Therefore, if a strategy to increase the content of the cell-derived proteins is converged into the SFC system reported in this study, it is expected that cultured meat with the flavor of traditional meat will be possible.

This study focuses on material engineering for controlling the flavor characteristics of cultured meat. Therefore, the use of animal-derived components which is an issue in the cultured meat field is overlooked in this research. However, considering that the cultured meat industry ultimately aims for sustainable meat development, it is essential to discuss the feasibility of our SFC system even without animal-derived materials such as gelatin and serum-containing media. In this study, SFC is conjugated to the GelMA backbone by the reaction between the methacryloyl and the binding group of SFC. The introduction of methacryloyl in gelatin is possible by the substitution of methacrylate groups on the amine-containing side groups of gelatin. Since proteins have amine groups, methacryloyl can also be

introduced to plant-derived proteins. Therefore, we expect that the SFC system can be possible to various plant-derived proteins. Also, serum-free media are being actively studied in the field of cultured meat. It is anticipated that if non-animal derived nutrients capable of replacing serum in media are developed, it would be able to verify the flavor-enriching effects of the SFC system more precisely. Because serum-containing culture medium contains various animal-derived proteins and glucose, the Maillard reaction may occur at 150 °C. The products from the Maillard reaction between animal serum and glucose in culture media might react with the flavor compounds derived from SFC because the Maillard reaction produces various unstable and highly reactive volatile compounds[8]. This reaction might converse the flavor compounds from SFC to non-flavor compounds or might create undesirable or desirable flavor compounds. In Figs. 2 and 3, it can be verified in both CM-SFC and CM + SFC that the number of detected flavor compounds are decreased after culturing cells. This might be due to the effect of volatile compounds created from serum-containing media which remain in the cultured meat specimens. If the serum-free medium is applied to this study, unwanted reactions between the volatile compounds from serum-containing media and the compounds from SFC would be excluded, which would help analyzing the effect of the SFC system on cultured meat's flavor characteristics more precisely.

## Methods

### Inclusion & ethics
All the experimental procedures for animal slaughter were conducted in compliance with the Animal Care and Use Guidelines of Kangwon National University and were approved by the Institutional Animal Care and Use Committee (IACUC) of Kangwon National University (IACUC approval no. KW-220714-1).

### Synthesis of the switchable flavor compound (SFC)
To induce the formation of disulfide bonds at the thiol groups of Maillard reaction products, 11.6 mmol of 3-mercapto-1-propanol (TCI-SEJIN CI, Korea) was reacted with equimolar amounts of furfuryl mercaptan (TCI-SEJIN CI, Korea), 3-mercapto-2-pentanone (TCI-SEJIN CI, Korea), or 2-methyl-3-furanthiol (TCI-SEJIN CI, Korea) for 48 h at 80 °C, along with 2.56 mmol of hydrogen peroxide. SFC was prepared via urethane reaction with the isocyanate trimer (hexamethylene diisocyanate isocyanurate trimer; BLD Pharm, China). The molar ratio between isocyanate trimer, 2-hydroxyethyl methacrylate (Sigma–Aldrich, #128635), and disulfide-linked Maillard reaction products was 1:2:1. The SFC based on furfuryl mercaptan was synthesized with a 30% (w/v) concentration of all reagents within the solvent (propylene carbonate; Sigma–Aldrich, #110264). Technically, 0.5 mmol isocyanate trimer (0.267 g) was reacted in 1 mL propylene carbonate (1.2 g, 11.8 mmol) where the molar ratio with 2-hydroxyethyl methacrylate and disulfide-linked furfuryl mercaptan was set to 2 and 1, respectively. In contrast, SFCs with 3-mercapto-2-pentanone or 2-methyl-3-furanthiol were prepared with 20% (w/v) concentration. In detail, 0.3 mmol isocyanate (0.156 g) was reacted in 1 mL propylene carbonate (1.2 g, 11.8 mmol) where the molar ratio with 2-hydroxyethyl methacrylate and disulfide-linked Maillard reaction products was identical to the furfuryl mercaptan case. The reaction was performed at 80 °C for 4 days for the SFC conjugated with furfuryl mercaptan, and for 3 days for the SFC conjugated with 3-mercapto-2-pentanone and with 2-methyl-3-furanthiol.

The chemical structure was investigated by the Raman spectroscopy (XploRA PLUS, HORIBA, France) and ${}^1$H NMR (Avance III HD 300, Bruker Biospin, USA). For the Raman spectroscopy measurement, a 10x objective and a 532 nm laser (75 mW intensity) were employed. The peak at 520 cm$^{-1}$ was used as the calibration standard. The laser was irradiated for 30 s per cycle, and each analysis was repeated five times to acquire reliable spectra. For the NMR studies, the SFC was dispersed

in the dimethyl sulfoxide-d$_6$ (Sigma–Aldrich) with the volume concentration of 10%.

### Fabrication of hydrogels
First, gelatin methacryloyl (GelMA) was synthesized by conjugating methacrylic anhydride to fish gelatin, referring to the previous research[55,56]. Fish gelatin (GELTECH, Korea) was dissolved in deionized water (DW) at 65 °C to make a 20% (w/v) gelatin solution. Then, 0.08 mL of methacrylic anhydride (Sigma-Aldrich #276685) was added in the gelatin solution per 1 g of fish gelatin. After stirring 2 h at 500 rpm, the gelatin solution was diluted 2-fold with distilled water. Then, the GelMA solution was dialyzed for 5 days at 80 °C hot plate using a 12–14 kDa membrane (Thermo Fisher, #08667E). Distilled water was used as dialysate. The dialyzed GelMA solution was lyophilized for 5 days using the lyophilzer (FreeZone −50 Benchtop Freeze Dryer, Labconco) after 1 day of freezing at − 20 °C.

To obtain the scaffold precursor solution, the lyophilized GelMA was dissolved in deionized water at 50 °C to make a 20% (w/v) solution, followed by adding 0.1% (w/v) 2-hydroxy-4-(2-hydroxyethoxy)-2-methylpropiophenone (I2959; Sigma-Aldrich, #410896) and 0.5% (w/v) SFC. To prepare the GelMA scaffold without SFC (denoted as "Gel-SFC"), we used 20% (w/v) GelMA solution with 0.1% (w/v) I2959. For the GelMA scaffold with single furfuryl mercaptan (denoted as "Gel+FM), 0.5% (w/v) furfuryl mercaptan was added instead of SFC. The scaffold with furfuryl mercaptan containing SFC (named as "Gel+SFC") was prepared with the same concentration of GelMA solution and I2959 as Gel-SFC, but the addition of 0.5% (w/v) SFC containing furfuryl mercaptan. The scaffold containing various SFC types (named as "Gel + SFCV") was also obtained by adding SFCs into the same hydrogel precursor of Gel-SFC. Specifically, 0.16% (w/v) SFC containing furfuryl mercaptan, 0.16% (w/v) SFC with 3-mercapto-2-pentanone, and 0.16% (w/v) SFC with 2-methyl-3-furanthiol were added instead of solely adding 0.5% (w/v) SFC. This hydrogel precursor was distributed into a 24-well plate (SPL), followed by 3 h of ultraviolet (UV) exposure to complete polymerization. To remove any unreacted molecules, the hydrogels were washed with distilled water for four times. The first two times were performed by immersing the hydrogels with water for 1 min for each wash. Then, the hydrogels were immersed in water for 6 hours at 37 °C to allow the hydrogels to swell. Then, the hydrogels were taken from the water and washed with new distilled water for 1 min.

### Thermo-responsivity analysis of SFC
The UV quartz cuvette (chamber volume 3.5 mL, path-length 10 mm) was filled with 2 mL of 1.86 µM SFC. SFC was heated at 37 °C, 80 °C, and 150 °C for 24 h by maintaining the closed system after sealing with the polytetrafluoroethylene stopper. The UV-VIS spectra within 200–600 nm were obtained for the desired heating time points with the settings of bandwidth 1.0 nm, scan speed 240 nm/min, and data interval 1.0 nm. The peak at 335 nm was assigned to the furan-related absorbance. Here, three replication experiments ($n$ = 3) were performed for each heating temperature for reliability.

To understand the long-term stability under open system, the vials (chamber volume 5 mL, 18 × 38 mm$^2$) were filled with 1 mL of 0.37 mM SFC ($n$ = 3) and 1 mL of pure furfuryl mercaptan ($n$ = 3), respectively. The vials with SFC and furfuryl mercaptan was positioned at 37 °C up to 2 weeks. In particular, to emulate the open system during cell culture procedures, the vials were not closed with a cap, and the air exchange kept sufficient. The weight variation was measured for specific time points. ${}^1$H NMR (Avance III HD 300, Bruker, Bruker Biospin, USA) spectra of residual SFC were obtained by diluting in the dimethyl sulfoxide-d$_6$ (10 vol%).

### Stability analysis of flavor molecules in hydrogel
To assess the stability of the flavor molecules in the hydrogels, the three types of samples (GelMA scaffold without switchable flavor

compound (Gel-SFC), GelMA scaffold with switchable flavor compound (Gel + SFC), and GelMA scaffold with furfuryl mercaptan mixed (Gel + FM)) were immersed in distilled water for the whole cell culture period (15 days). Then, the hydrogels were transferred to a vial and kept at room temperature or heated at 150 °C for 5 min. A heating plate was used for temperature control. Specifically, the heating plate was covered with aluminum foil and the temperature was adjusted. When the temperature reached to the setting point, the samples were put on the foil covered plate for 5 min. Using GC–MS (Agilent 8890 GC system–Agilent 5677B MSD, Agilent Technologies), volatile and semi-volatile compounds were detected for each group[5,57]. In particular, carboxen/polydimethylsiloxane/divinylbenzene (CAR/PDMS/DVB) fibers were used to adsorb the volatile compounds from the samples by headspace solid-phase microextraction (HS–SPME). At this time, the groups that were placed at room temperature were heated to 30 °C for 20 min, while the others were heated to 80 °C for 20 min using the autosampler (Agilent 7693, CombiPAL sampler 80, Agilent) to allow the volatile compounds to be adsorbed onto the fibers. Then, the samples were maintained at room temperature with the fibers for an additional 40 min to complete the adsorption of the compounds. Subsequently, the volatile compounds adsorbed onto the fibers were analyzed. Helium with 99.999% purity was used for the carrier gas with a split flow rate of 40 mL/min. Fibers were injected into the inlet at the temperature of 250 °C using the split (20:1) injection mode. The initial temperature of the column oven was 40 °C and the hold time was 5 min. Then, the temperature increased to 240 °C at a rate of 4 °C/min, and the set temperature was hold for 20 min to analyze the volatile compounds. The GC column used in this experiment was DB-WAX (Agilent 123-7063). The temperature of the column oven was kept at 40 °C for 5 min and then increased to 240 °C at a rate of 4 °C/min. Finally, the column oven temperature was kept at 240 °C for 20 min to analyze the volatile and semivolatile compounds of each group.

## Cell culture

Before culturing cells, the hydrogels were frozen at −20 °C for one day and freeze-dried for 4 days at −50 °C by lyophilizer to obtain the scaffolds. Then, the lyophilized aerogels were washed with 70% (v/v) ethanol and sterilized by exposure to UV irradiation for 2 h. The sterilized gels were swollen in high-glucose Dulbecco's modified Eagle medium (HG-DMEM, high glucose; Gibco) at 37 °C before cell seeding.

Bovine myoblasts were sub-cultured on TPP® tissue culture dishes (Sigma–Aldrich) with growth media composed of HG-DMEM (Thermo Fisher Scientific) with 10% (v/v) fetal bovine serum (FBS; Welgene), 1% (v/v) penicillin–streptomycin–amphotericin (PS; Gibco® Life Technologies), and 5 ng/mL basic fibroblast growth factor (bFGF; Peprotech, Rocky Hill, NJ, USA). For cell subculture, cells were washed with 1X PBS (Gibco® Life Technologies, USA) and detached using 0.025% (v/v) trypsin EDTA (Welgene). Myoblasts at passage 3 were seeded on the scaffolds at a density of $2 \times 10^4$ cell/scaffold (16.6 mm² x 0.8 mm). The growth medium was replaced every two days to feed the cells during proliferation. After 7 days of cell proliferation, cell differentiation was induced for another 8 days using a differentiation medium, consisting of HG-DMEM supplemented with 5% (v/v) horse serum (Gibco™ Horse Serum, Thermo Fischer Scientific) and 1% (v/v) PS. The differentiation medium was replaced every two days. After cell differentiation, samples were renamed to indicate that the samples were in the form of cultured meat. The cultured meat group using Gel-SFC was named "CM-SFC" whereas the cultured meat group using Gel + SFC was named as "CM + SFC". The cultured meat group with Gel + SFCV was then named as "CM + SFCV".

## Biological assessments of cell proliferation and differentiation

For cell proliferation analysis, myoblasts were stained with CCK-8 (D-Plus™ CCK cell viability assay kit, Dongin LS, Korea), following the protocol provided with the kit. To confirm cell adhesion,

immunostaining was performed as follows: first, cell fixation was performed with formalin solution, followed by washing with 1X PBS thoroughly. Then, a blocking solution composed of 2% (w/v) bovine serum albumin (BSA; Sigma–Aldrich), 0.3% (v/v) Triton X-100 (Triton™ X-100 solution, Sigma–Aldrich), and 10% (v/v) horse serum in 1X PBS was added to the cells overnight. F-actin filaments of myoblasts were stained using the staining solution which includes 0.165 uM Alexa Fluor 488™ phalloidin (A12379, Thermo Fisher Scientific) and 0.08 mg/mL DAPI (Thermo Fisher Scientific, #D9542) in the solvent composed of 1X PBS and 1% BSA. For myosin heavy chain staining, MHC antibody was diluted 100-fold using the MF 20 concentrate solution (MF 20, DSHB, ID:AB2147781) in a solvent composed of 10% (v/v) horse serum and 2% (w/v) BSA. The MHC antibody solution was then added to the cells for 2 h at room temperature. Then, the samples were washed twice with 1X PBS and once with 0.025% Triton X-100. Subsequently, 0.005 mg/mL secondary antibodies (Donkey anti-mouse Alexa Flour 594, Thermo Fischer, #A21203) in the same solvent as MF20 were added to the samples for 1 h at room temperature, followed by washing. Confocal laser scanning microscopy (CLSM; LSM 980, Carl Zeiss) was used for stained cell observation.

MHC quantification was performed using a bovine myosin-1 (MYH1) ELISA kit (MyBioSource).

## Flavor analysis

GC–MS and an e-nose (HERACLES-II-E-NOSE, Alpha MOS, France) were used for flavor analysis. To compare the aromatic notes of CM-SFC and CM + SFC, GC–MS was used to quantify the ratio of flavor compounds from each group, using the same stability analysis method employed for flavor molecules in the hydrogel. Briefly, CM-SFC and CM + SFC were placed in a vial separately and heated at 150 °C for 5 min before the GC–MS analysis. To prevent the compound from leaking, each vial was sealed with parafilm before heating. Then, the HS-SPME fibers were placed in each vial and the temperature was set to 80 °C to adsorb all the volatile and semivolatile compounds. GC–MS analysis was performed using the fibers with the adsorbed compounds from each group, employing the same method used for the stability analysis.

To compare the flavors of the experimental groups to that of beef brisket, an e-nose was used to investigate the flavor pattern of each group. Hanwoo beef brisket was purchased from SIR.LOIN, Korea. For precise comparison. Then, 0.4 g of CM-SFC, CM + SFC, CM + SFCV, and beef brisket were transferred into a 20 mL headspace vial, followed by heating at 150 °C for 5 min. Volatile and semivolatile compounds produced in the headspace of the vial were then injected into the inlet of the e-nose at an injection speed of 125 μl/s. The injector temperature was 200 °C and the inject time was 45 seconds. For carrier gas supply, $H_2$ was used for the injector as well as for the detector. Two independent chromatographic columns and two flame ionization detectors were used for compound detection. The non-polar column was MXT-5 GC metal capillary column (Restek) was used whereas the second column with medium polarity was MXT-1701 GC metal capillary column (Restek). For the detectors, flame ionization detector (FID)1 (non-polar) and FID2 (slightly polar) were used, and the detector temperature was 260 °C. The detected compounds were then analyzed using the Alpha Software (Alpha MOS, France).

## Flavor profile assignment

The specific flavor note was assigned to each detected volatile compound using the flavor ingredient library of the Flavor and Extract Manufacturers Association of the United States (FEMA) database.

## Statistics and reproducibility

The data are reported as mean ± standard deviation (SD). Statistical analysis was performed with a significance level of 0.05 using One-way ANOVA with the Tukey method via OriginPro 2018 Software. All experiments were repeated three times independently.

**Reporting summary**

Further information on research design is available in the Nature Portfolio Reporting Summary linked to this article.

## Data availability

All data supporting the findings of this study are available within the paper and its Supplementary Information. Also, raw data of the study is provided in the Source data file. Source data are provided with this paper.

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

## Acknowledgements

This research was supported by (1) the Korea Institute of Marine Science & Technology Promotion (KIMST) funded by the Ministry of Oceans and Fisheries (RS-2024-00402200, J.H.) and by (2) the National Research Foundation of Korea (NRF) (RS-2024-00354178, J.H.).

## Author contributions

M.L., W.C., and J.H. contributed to designing the study and performed experiments. J.M.L., S.T.L., and W.G.K. contributed to the in vitro experiments.

## Competing interests

The authors declare no competing interests.
