## [Peer Review File · Nature Communications]

REVIEWER COMMENTS

Reviewer #1 (Remarks to the Author):

Dear authors, the manuscript is very interesting, however there are some lacks that need to be improved. It needs revision to clarify and improve some points in the manuscript, such as in the results, discussion, and methods sections. The main weakness lies in a better methods description and a more clear presentation of results and discussion sections. The lack of comparison of all analyses with beef samples is also an issue. Other than that, the use of nonfood grade reagents for pre-scaffolding must be discussed in this paper and improved for new research. All my points are highlighted in the document attached.

Reviewer #2 (Remarks to the Author):

In this study, the authors generate thermal-responsive gelatin-based scaffolds which, through temperature sensitive covalent linking, are able to release key flavor compounds upon cooking. They demonstrate that these flavor compounds remain stably linked to the scaffolds over long-term cell culture with bovine muscle cells, and that only when the resulting cultured meat constructs are cooked are the compounds released. The authors show that this release is associated with an improvement in overall aromatic likeness to conventional beef. This is a clever and innovative approach, which is indeed novel to the field. As such, it shows substantial promise as an exciting contribution to the literature. That said, there are several minor and major revisions that the authors must perform before the manuscript is sufficiently clear and detailed for publication

Minor revisions:

- Throughout the manuscript are several areas where editing for clarity and readability would be helpful. I have suggested some specific changes in the attached markup document, but the authors are welcome to make their own changes, if they do not deem the markup to be optimal.
- Line 46 states that “cultured meat is far from being recognized as a meat product,” however, two products have already been approved for sale as meat (in the USA and/or Singapore) and regulatory agencies label these as “cell cultured meat” (for instance, in the USA). As such, it is incorrect to say that cultured meat is not recognized as a meat product.
- Line 53-54 states that cultured meat faces an “absence of the Maillard reaction;” however, no evidence is provided for this. Indeed, it is likely that cultured meat produced even without temperature-sensitive hydrogels would undergo some Maillard reaction (for instance, there is suggested evidence of a maillard reaction in DOI: 10.1016/j.ijbiomac.2023.129134 and again in DOI: 10.1038/s41467-023-44359-9 and DOI: 10.1021/acsami.2c10988, which are the authors own work). The authors should remove this statement, as the evidence suggests otherwise.
- Line 55 states that “the amino acid types of cells are not as diverse as those of meat;” however, no evidence is provided to validate this claim. Indeed, this paper: (DOI: 10.5851/kosfa.2021.e72) suggests that all amino acids present in traditional beef or chicken are also present in cultured beef or chicken. The authors must back up their claim with a reference from the literature, or else remove it.
- In line 68, the authors claim a two-week period. That may be what the authors use in this paper, but it is not universal to all cultured meat production processes. A wider range should be mentioned, given the range of timelines that might be required for cultured meat production.

- Line 90 should include a scale-bar of the photograph of cultured meat.
- Throughout, the authors use active voice rather than passive voice (e.g., line 107 says “we investigated the temperature responsivity” rather than something like “temperature responsiveness was investigated”). The latter would be more appropriate. Throughout the manuscript, the authors should change to passive voice (avoiding reference of “we”).
- Line 109: “significant” should be changed to some other word, such as “notable” or “noticeable” or “detectable,” as no statistics were performed.
- Line 148 (example, though there are other instances as well). There are some scenarios where “flavor” is used where “flavor compound” or “aroma” should be used instead. For instance, in 148, the authors say that “flavor was detected,” however, since flavor is a sense rather than a chemical, this should either be “aroma” or “flavor compound”
- Line 162: the authors should specify the name of the maillard reaction product (e.g., chemical name) in the figure legend.
- Line 199: It is not clear to me why cell infiltration would result in insufficient CCK-8 data. If the sponges are porous, then transfer of metabolites into/out of the scaffolds should be possible, and so the assay should work as expected.
- Line 216: Were no other compounds from the scaffold detected? Only this singular molecule?
- Line 220: It would be helpful to clarify which flavor compounds these were with their chemical names, as it would help the reader track these compounds throughout the paper.
- Line 265: This line claims that cultured meat without the flavor enhancement strategy had undesirable olfactory characteristics. However, this is only validated for this system. The authors should clarify that this might not be true for other systems (e.g., other culture media, other scaffolds, etc.).
- Line 292, 293: the chemical 3-mercapto-1-propanol is mentioned twice. Is this correct, or a typo?
- Line 320: Please describe how long / how many times the hydrogels were washed.
- Line 334: Please make sure that all experimental details are given, including injection speed, etc.
- Line 339: Please provide the parameters of freezing (e.g., temperature, time, etc.) before lyophilization.
- Line 355: Please indicate how frequently cells were fed during proliferation.
- Line 367-380: Please provide catalog numbers for antibodies and reagents/kits used (please do this throughout the methods section). Additionally, please specify the concentration of MF20, since DSHB antibodies are provided with variable concentrations, and so a dilution factor does not exactly indicate the ug/mL of antibody used.
- Line 393: Please specify how much of the various samples were transferred to the 20 mL headspace vials.

Major revisions:

- The authors claim that no (or at least very few) studies have looked at flavor of cultured meat.

However, the authors fail to reference key papers on this topic, including:

o DOI: 10.5851/kosfa.2021.e72

o DOI: 10.1021/acs.jafc.2c08004

o DOI: 10.1038/s41467-023-44359-9 (the authors own work which only just came out, so they may be planning to add it in anyways).

o DOI: 10.1016/j.foodres.2022.111636

As such, the authors overstate the paucity of research in this area. Instead, the authors should take more time discussing the existing literature on this topic and putting the current study in the context of that broader literature.

- In line 180 (and in figure 3), the authors point towards meaty flavors being “roasted bread, meaty” and “bitter almond, burnt sugar, malt” – However, these do not match exactly with the flavors in figure 2D. Similarly, the flavors described in figure 4b(iii) do not clearly track with the same compounds. The authors should make sure that the words used to categorize flavors are consistent and include chemical names, where possible, so that the connection between the figures and flavors is clear.

- Throughout, the use of “flavor” and “aroma” is inconsistent (e.g., line 232 vs line 234). For consistency, the authors should stick to one term. I would propose “flavor compound” be used throughout the manuscript. This is because the context of the paper is around the flavor of the meats, but “flavor” is a general sense caused by a complex mixture of flavor compounds, and so the use of “flavor” alone might be inaccurate. The use of “flavor compound” is both clear and accurate.

- In lines 194-195, the authors say that a lack of significant difference between the groups “indicates non-cytotoxicity” – however, this is inappropriate statistics for proving the null hypothesis. Rather, with the statistical tests used, the authors should only claim that “no significant difference in cell viability existed between CM-SFC and CM+SFC on days 1 and 5, though a significant reduction in viability was present on day 7 for CM+SFC.” Relatedly, it is inappropriate in line 197 to claim that the difference is negligible because of a high-ish p-value. Indeed, numerically the absorbance looks to be two-fold higher for SFM-SFC than SFM+SFC on day 7, which could be seen as a very non-negligible difference. The results clearly suggest (both absolutely and statistically) that SFM+SFC reduces cell growth over a seven day period. This is okay, and does not diminish the study – the authors should present the facts frankly and avoid couching their findings in caveats.

- Line 198-199: the cell infiltration images are unconvincing, showing very limited penetration. It would be helpful if the authors could provide additional images to give a clearer picture of the infiltration, and if they could provide 3D images of the confocal images through the scaffolds.

- The timing of various heating experiments is a bit confusing. For instance, in Fig. 2b and supplementary figure 3, >1 hour is needed to see detectable compound release; however, later cooking experiments only heat for 1 hour (e.g., line 246). How is it possible that these shorter cooking times are releasing sufficient flavor compounds for detection? The authors should clarify this process, the choice of a 5 minute cooking time, and how this is mechanistically possible, otherwise it is hard to follow and seems implausible that 5 minutes would yield detectable flavor compound release.

- Line 247: Why is PCA used as the analytical method, here? The first principle component accounts for 96% of variability, which suggests that this is a simpler cause-and-effect association than would require PCA. Relatedly, the PCA analysis reveals dramatic differences between cultivated and slaughtered meat. This should be noted after line 250, which says that CM+SFCV had the most similar pattern to beef. Specifically, the authors should note that while this was the most similar, it was still significantly different from beef, suggesting that much work is still required to get these flavor profiles closer to that of slaughtered meat. Line 283, too, should be amended to

say that the flavor pattern of cultured meat became “more similar” to that of beef, rather than just “similar.” Indeed, the PCA data suggests that the flavor pattern of cultured meat is still mostly different from conventional beef.

- Conclusion: The manuscript is lacking in discussion of the study’s limitations. The authors should address the following questions:
 - o The cultured meat is still quite different from conventional meat. Why might this be the case, and how can further engineering get it closer to beef?
 - o What about the use of gelatin? That is an animal-derived component, and so might not be appropriate for cultured meat. The authors should discuss opportunities for transitioning this technology to animal-free systems.
 - o Similarly, the study was performed with serum containing media, which could itself provide flavor compounds to the system. The authors should discuss how/if serum-free culture might affect the system.
 - o The authors do not discuss the safety of the chemistries used from a food safety standpoint. Are any toxic chemicals used? How can we ensure that no residual chemicals remain? Are toxic byproducts potentially produced? If so, how can the technology be transitioned to food-safe chemistry? Or, alternatively, is the chemistry already food-safe?

- Similarly, the manuscript is lacking in discussion of future research directions. Please include these in the discussion.

- Line 343: Please provide details for the myoblast isolation procedure, as well as data validation for the identity of these cells (e.g., staining for key satellite cell / myoblast markers such as Pax7 and MyoD)

- Methods: The methods are missing several experiments. Please provide methods for:
 - o The stability of non-bound compounds (fig. 2)
 - o NMR studies
 - o Raman studies
 - o Residual weight studies
 - o SEM
 - o Tissue sectioning / penetration studies
 - o Statistical methodsAs well as any other experimental details that are missing. All experiments must be adequately described.

Overall, I congratulate the authors on a clever and novel approach; however, additional clarity, consistency, detail, and discussion will greatly elevate the manuscript, and should be implemented before it is accepted for publication.

Reviewer #3 (Remarks to the Author):

In this manuscript the authors investigate the effect of adding Maillard reaction products into the scaffold of cultured meat on flavor release upon storage and cooking. This research is very interesting and has a very valuable aim, which is the improvement of the flavor profile of cultured meat. In my opinion it should be accepted for publication after further improvement. A general comment is on the R&D, there is little comparison with literature. Detailed comments are below, and they are all about clarity because I think the experiments/experimental design are well performed.

Introduction

Line 55: “the amino acid types of cells” check this sentence.

Line 73. The author should explain exactly what is the switchable flavor compound. What do they

mean with "compound", because of this lack of explanation, the sentence is not clear.

Line 75. Is the gelatin methacryloyl the constituent of the scaffold? Mention it.

Line 80-81. "Because the SFC is selectively volatilized upon heating, CM+SFC can replicate the Maillard reaction of conventional meat." What does mean "selectively"? and how this lead to the formation of maillard products formed in conventional meat? The term "selectively" is vague.

Figure 1. the caption can be improved. For panel a, it is mentioned "a Fabrication of cultured meat with switchable flavor compound (CM+SFC). Scale bars: 100 μ m." I think the scale does not refer to the whole panel a.

The authors need to explain here (line 64) on in the R&D the rational behind the selection of the compound furfuryl mercaptan among all the Maillard reaction products with meaty notes.

Moreover, in the paragraph 72-86 or at the beginning of the R&D they should introduce well the samples and why they worked on theses samples, what was they hypothesis.

Results

Figure 2. Explain in the caption what are R1-R3. Moreover, "B Investigation of Maillard reaction product mobility from SFC upon heating." too vague. what is panel b? what measure was done? Moreover, was the SFC in the gel?

Line 108-110. The authors should explicitly mention how the temperature responsivity of the SFC was measure... what was monitored?

Line 116. The reference to paper 21 is not clear. Why the authors refer to it?

Line 139. "which may be due to the inhibition of the polymerization of the hydrogel by furfuryl mercaptan" why the author think so? Could you add a reference.

Line 142. " after immersing them in distilled water" why at in distilled water? What is the rational behind it?

Line 147. In the sentence "Then, the ratios of volatile compounds with and without flavor notes were calculated from GC-MS peak areas" what the authors mean with "with and without flavor"? do they meant before and after heat treatment or Gel+SFC and Gel+FM? Mention it.

Line 147 and 148. "pungent aroma" and "nutty aroma" were they a sensorial observation of the researcher, or detected by the GCMS and in the latter case, what compounds were those? Mention it.

Line 148-152. In the first sentence, the authors talk about a trend in the Gel-SFC but in the next sentence, they said that the opposite trend was observed for samples Gel-SFQ and Gel-FM. Please, clarify this.

Line 149. What the authors mean with "the overall proportion of aroma compounds"? why "proportion" and not "concentration"?

Figure 2 panel d. I agree with showing in the main manuscripts only the attributes of the detected volatiles but, I think, the authors should report the identified compounds in the supplementary material. Moreover, they need to explain the abbreviations. Each figure and table should be understood even out of the manuscript.

In all the figures: introduce abbreviation in the captions.

Line 219. "which might be due to thermal degradation of GelMA" why? Add reference.

Methods : no need of reference in the Methods?

Response letter

Journal: *Nature Communications*

Manuscript ID: NCOMMS-23-63842-T

Title: Flavor-Switchable Scaffold for Cultured Meat with Enhanced Aromatic Properties

Authors: Milae Lee, Woojin Choi, Jeong Min Lee, Seung Tae Lee, Won-Gun Koh, Jinkee Hong

Response to Reviewer #1

Reviewer(s)'s Comments to Author:

Reviewer 1.

Comments

Dear authors, the manuscript is very interesting, however there are some lacks that need to be improved. It needs revision to clarify and improve some points in the manuscript, such as in the results, discussion, and methods sections. The main weakness lies in a better methods description and a more clear presentation of results and discussion sections. The lack of comparison of all analyses with beef samples is also an issue. Other than that, the use of nonfood grade reagents for pre-scaffolding must be discussed in this paper and improved for new research. All my points are highlighted in the document attached.

Response

The authors sincerely appreciate the reviewer's valuable comments. By carefully reflecting the comments, we fully revised the manuscript and figures. Mainly, we classified "desirable flavors" and "offensive flavors" in the first place, then we revised the figures to present that our strategy can increase the desirable flavors in cultured meat. Desirable flavors indicate meaty, savory, almond-like, floral, fatty, and fruity flavors which comprise the flavor notes of conventional grilled meat¹⁻⁵. Offensive flavors are defined as the flavors that are fishy, pungent, and sour flavors which are not desirable as food flavors according to previous research⁶⁻⁸. In the revised discussion section, we clarify that our strategy is rather mimicking the flavor notes than implementing the exact flavor chemicals of conventional meat.

After reading the reviewer's comments, we deeply realized that the method description and discussion in the manuscript were very lacking. Also, the consistency of the analyses was low,

and the presentation of the results was ambiguous. Therefore, we revised the whole figures with clearer descriptions, and we supplemented the research discussion. As the reviewer pointed out, our research contains nonfood grade reagents. We fully discussed about this issue in the revised manuscript. Again, we sincerely appreciate the reviewer for the valuable comments that helped us improve the quality of our manuscript.

To respond to all the valuable comments from the reviewer in detail and sincerely, point-by-point responses to the reviewer's comments were written. The changes in the revised manuscript related each comment are presented in blue font color. Once again, we appreciate the reviewer for giving us the opportunity to revise our manuscript.

[References]

1. Food Research International, 157, 111385 (2022)
2. Food Chemistry, 129, 1-4, 432-438 (2013)
3. Meat Science, 80, 3, 728-737 (2008)
4. npj Science of Food, 7, 13 (2013)
5. Nature Communications, 15, 77 (2024)
6. Human brain mapping, 17, 17-27 (2022)
7. International Journal of Food Science & Technology, 57, 2277-2284 (2022)
8. The Journal of clinical and aesthetic dermatology, 6, 45 (2013)

Comment 1. Line 54 -56: Of what cells? Please revise the sentence and don't make generic colocations.

Response:

We apologize for the ambiguous expression that can make the fallacy of generalization. As we already explained that the in vitro cell culture alone lacks the Maillard reaction in in the line 53-54, we removed the sentence pointed by the reviewer.

Comment 2. Line 58: What cell lines?

Response:

We apologize for the unclear expression in Line 58. We used murine and bovine myoblasts in our previous works, and we added this information in the revised introduction as below.

Related changes in the manuscript:

(Line 61-62)

Recently, our group revealed that the enhanced proliferation and differentiation of murine and bovine myoblasts could induce relatively more meat-like flavors.

Comment 3. Line 76 – 78: Sentence is confusing, please revise.

Response:

We regret the confusing description in line 76-78. We removed such sentence in the revised manuscript.

Comment 4. Line 88 (Figure 1): This figure could be a graphical abstract. The real image of the cultured meat must be improved and highlighted.

Response:

We sincerely appreciate the reviewer's comment. We regret the poor quality of our cultured meat's real image. We improved the image and revised the Fig. 1 to highlight it. In the revised figure 1, we also changed the illustrative description of the switchable flavor compound system. Moreover, we denoted the chemical structure and name of the volatile compounds detected in our research which were classified by the flavor notes (Fig. R1d).

Revised Fig. 1:

Fig. R1 (Fig. 1): Schematic illustration of switchable flavor system. a Illustrative description of scaffold structure conjugated with switchable flavor compound (SFC). **b** Photograph of cultured meat fabricated using the scaffold with SFC (CM+SFC). Scale bar: 8 mm. **c** Illustration explaining the mechanism of SFC system depending on temperature. **d** Classification of the flavor compounds analyzed in this study.

Comment 5. Line 97 – 100: There are a confusion between results and discussion. The authors must revise these entire sections and present results in this one and discussion in the other.

Response:

The authors sincerely appreciate the considerate reviewer's comment. Reflecting on your meaningful opinion, we thoroughly revised the discussion section as suggested below.

Related changes in the manuscript:

(Line 133-135)

In case of the thermal responsive group (R₃), a disulfide bond to the roasted meat-flavored Maillard reaction product, furfuryl mercaptan, was introduced (Supplementary Fig. 1a).

(Line 321-342)

Volatile small molecules, representatively, gas molecules, volatile organic compounds, and flavor compounds, have attracted considerable attention in broad research and application fields: capture of environmentally impacting carbon dioxide gas¹, electrochemical production of hydrogen gas as a green energy resource², deactivation of volatile chemical warfare agents³, disease diagnosis by exhaled breath gas^{4,5}, vascular disease treatment by nitric oxide gas^{6,7}, and carbon monoxide gas-based gastrointestinal inflammation suppression⁸. However, given the utilization in open systems, such as air-permeable cell culture processes, these volatile small molecules suffer from their fast mass transport rate. Therefore, strategies for stable entrapment and controllable release of volatile small molecules have been developed. The emulsion and perfluorocarbon were representatives for physical retention^{9,10}. Meanwhile, because these physical capsules degrade rapidly, the selective generation of the volatile small molecules is a challenge. Although the chemical binding promises significant durability, it requires external energies (*e.g.*, proteolysis¹¹, magnetic force¹², and catalysis¹³) to trigger the volatile compound release on demand.

In this study, we employed robust covalent bonds to secure the flavor stability during a

prolonged open system process for cell proliferation and differentiation. Moreover, we devised thermal energy as a desirable trigger resource, given the heating-based cooking and the positive correlation between temperature and diffusion coefficient of Maillard reaction product. In particular, among the dynamic covalent bonds, such as, imine¹⁴, boronate-ester¹⁵, and hydrazine bonds¹⁶, the disulfide bond was adopted considering its temperature responsivity^{17,18}. Accordingly, this thermo-responsively Maillard reaction product releasing system, so-called SFC, enabled the cell scaffold-based replication of the Maillard reaction of slaughtered meat.

[References]

1. Nat. Rev. Mater., 2, 17045 (2017)
2. Nature, 612, 673-678 (2022)
3. Nat. Rev. Chem., 5, 370-387 (2021)
4. ACS Sens., 4, 268-280 (2019)
5. Nat. Nanotechnol., 15, 792-800 (2020)
6. ACS Nano, 17, 8935-8965 (2023)
7. Materials Today, 72, 57-70 (2023)
8. Sci. Transl. Med., 14, ab14135 (2022)
9. Adv. Colloid Interface. Sci., 298, 102544 (2021)
10. Mol. Pharmaceutics, 20, 3254-3277 (2023)
11. Nat. Nanotechnol., 15, 792-800 (2020)
12. Sci. Robot., 5, aaz4239 (2020)
13. J. Am. Chem. Soc., 145, 11019-11032 (2023)
14. Journal of the American Chemical Society, 141, 18048-18055 (2019)
15. Proceedings of the National Academy of Sciences, 109, 4383-4388 (2012)

16. Journal of the American Chemical Society, 135, 17663-17666 (2013)
17. Nature Reviews Materials, 5, 562-583, (2020)
18. Journal of the American Chemical Society, 144, 2022-2033 (2022)

Comment 6. Line 109: This is not well described in the methods, please revise it.

Response:

We apologize for the missing related methods in the original manuscript. We described the experimental details so the reviewer and future readers have more information. Furthermore, thanks to your kind comment, we found that the methods related to Supplementary Fig. 4 were also missing. We updated entire methods as suggested below.

Related changes in the manuscript:

(Line 472-486)

Thermo-responsivity analysis of SFC

The UV quartz cuvette (chamber volume 3.5 mL, path-length 10 mm) was filled with 2 mL of 1.86 μ M SFC. SFC was heated at 37 °C, 80 °C, and 150 °C for 24 h by maintaining the closed system after sealing with the polytetrafluoroethylene stopper. The UV-VIS spectra within 200-600 nm were obtained for the desired heating time points with the settings of bandwidth 1.0 nm, scan speed 240 nm/min, and data interval 1.0 nm. The peak at 335 nm was assigned to the furan-related absorbance. Here, three replication experiments ($n = 3$) were performed for each heating temperature for reliability.

To understand the long-term stability under open system, the vials (chamber volume 5 mL, 18 \times 38 mm²) were filled with 1 mL of 0.37 mM SFC ($n = 3$) and 1 mL of pure furfuryl mercaptan ($n = 3$), respectively. The vials with SFC and furfuryl mercaptan was positioned at 37 °C up to 2 weeks. In particular, to emulate the open system during cell culture procedures, the vials were not closed with a cap, and the air exchange kept sufficient. The weight variation was measured for specific time points. ¹H NMR (Avance III HD 300, Bruker, Bruker Biospin, USA) spectra of residual SFC were obtained by diluting in the dimethyl sulfoxide-d₆ (10 vol%).

Comment 7. Line 155 – 156: All chromatograms must be presented in the supplementary material.

Response:

We appreciate the reviewer's comment, and we regret the insufficient supplementary data. We added all chromatograms of Fig. 2d in the supplementary Fig. 5 (Fig. R2) as below.

Related changes in the supplementary information:

Fig. R2 (Supplementary Fig. 5): GC-MS results of the hydrogels in Fig. 2d. a Chromatograms of the hydrogel without SFC (Gel-SFC) at 25 °C and at 150 °C **b** Chromatograms of the hydrogel with SFC (Gel+SFC) at 25 °C and at 150 °C **c** Chromatograms of the hydrogel mixed with furfuryl mercaptan (Gel+FM) at 25 °C and at

150 °C.

Comment 8. Line 165: Please be specific, what temperature?

Response:

We apologize for the insufficient description. We specified the temperature in the revised manuscript as below:

Related changes in the manuscript:

(Line 124-125)

Flavor analysis of hydrogels before and after heating at Maillard temperature, 150 °C.

Comment 9. Line 187 – 188 (GelMA biocompatibility explanation): And for food applications? It is possible to use it in some country? There are not food grade reagents that could be used instead of methacrylic anhydride?

Response:

First of all, we would like to address your question: GelMA is still far from food application, regardless of its well-proved biocompatibility, given that Food and Drug Administration (FDA) did not approve GelMA as a food-grade ingredient yet. The authors truly acknowledge your helpful opinion that the original statement could lead to a misunderstanding between biocompatibility and food-grade safety. We sympathize that the biocompatibility could not guarantee its food-grade safety. Thus, we modified the related section to avoid the overexpression. Furthermore, in the discussion section, we analyzed the current limitation of our study associated with the utilization of food-grade chemicals.

Related changes in the manuscript:

(Line 226-229)

The cell-cultured groups using Gel-SFC and Gel+SFC were denoted as cultured meat without SFC (CM-SFC) and cultured meat with SFC (CM+SFC), respectively. CM-SFC was used as control group to understand the influence of SFC on cell viability and myotube formation.

(Line 362-377)

However, irrelevant to these remarkable advances, this study still encounters a critical limitation. It should be cautious to regard the used reagents as food-grade, although they are proven to be biocompatible. For instance, the methacrylic anhydride is generally employed to functionalize the methacrylate in the Food and Drug Administration (FDA)-approved proteins (such as gelatin¹⁻³, silk fibroin⁴) and to prepare biocompatible scaffolds via photo-crosslinking. However, until recently, no sub-type of GelMA was granted by the FDA for food purposes. Namely, although the biocompatibility of CM+SFC was identified by the primary bovine

myoblast experiments, it is still distant from establishing its food safety. Nevertheless, the proposed flavoring strategy yields the potential to be implemented solely by food-grade chemicals. In particular, the enzymatic gelatin crosslink using FDA-approved transglutaminase could be a promising approach to realize the robust binding of Maillard reaction products using only food-grade chemicals⁵. Moreover, the thermo-responsivity could be fulfilled by the natural disulfide chemistry of cysteine amino acids⁶. In conclusion, beyond the current stage limitation related to food-grade chemistry, we anticipate that the interesting flavoring strategy developed in this study will contribute to the production of cultured meat that reaches the organoleptic properties of conventional meat.

[References]

1. Engineered Regeneration, 2, 47-56 (2021)
2. Biomaterials, 73, 254-271 (2015)
3. Biomaterials Research, 27, 86 (2023)
4. Nat. Commun., 9, 1620 (2018)
5. Nat. Commun., 15, 77 (2024)
6. Biomacromolecules, 23, 926-936 (2022)

Comment 10. Line 190: These descriptions must be in the method section.

Response:

We apologize for the insufficient description in the method section. We added the sample information in the method section of the revised manuscript as below:

Related changes in the manuscript:

(Line 458-472)

To prepare the GelMA scaffold without SFC (denoted as “Gel-SFC”), we used 20% (w/v) GelMA solution with 0.1% (w/v) I2959. For the GelMA scaffold with single furfuryl mercaptan (denoted as “Gel+FM”), 0.5% (w/v) furfuryl mercaptan was added instead of SFC. The scaffold with furfuryl mercaptan containing SFC (named as “Gel+SFC”) was prepared with the same concentration of GelMA solution and I2959 as Gel-SFC, but addition of 0.5% (w/v) SFC containing furfuryl mercaptan. The scaffold containing various SFC types (named as “Gel+SFCV”) was also obtained by adding SFCs into the same hydrogel precursor of Gel-SFC. Specifically, 0.16% (w/v) SFC containing furfuryl mercaptan, 0.16% (w/v) SFC with 3-mercapto-2-pentanone, and 0.16% (w/v) SFC with 2-methyl-3-furanthiol were added instead of solely adding 0.5% (w/v) SFC. This hydrogel precursor was distributed into a 24-well plate (SPL), followed by 3 h of ultraviolet (UV) exposure to complete polymerization. To remove any unreacted molecules, the hydrogels were washed with distilled water for four times. First two times were performed by immersing the hydrogels with water for 1 min for each wash. Then, the hydrogels were immersed in water for 6 hours at 37 °C to allow the hydrogels to swell. Then, the hydrogels were taken from the water and washed with new distilled water for 1 min.

(Line 531-535)

After cell differentiation, samples were renamed to indicate that the samples were in the form of cultured meat. The cultured meat group using Gel-SFC was named “CM-SFC” whereas the

cultured meat group using Gel+SFC was named as “CM+SFC. The cultured meat group with Gel+SFCV was then named as “CM+SFCV”.

Comment11. Line 196 (Cell proliferation significancy): At what level? Please describe the statistic treatment in the methods section.

Response:

We apologize for the insufficient information for the statistical analysis. The significance level used in this study was 0.05. The authors used One-way ANOVA (Tukey method) for the statistic treatment using OriginPro 2018 Software. We supplemented this information in the method section as below:

Related changes in the manuscript:

(Line 585-588)

Statistics and reproducibility

The data are reported as mean \pm standard deviation (SD). Statistical analysis was performed with the significance level of 0.05 using One-way ANOVA with Tukey method via OriginPro 2018 Software. All experiments were repeated three times independently.

Comment 12. Line 198 (Day 7 CCK significance): But it was a statistical difference. Describe the statistical treatment in the methods and discuss it properly.

Comment 13. Line 204 (DNA quantification): Add significance level.

Response to the comment 12 and 13:

We apologize for the insufficient information about the statistical analysis. We performed the statistic treatment at the 0.05 significance level using One-way ANOVA with Tukey method for both CCK analysis and DNA quantification. To discuss the reason of lower cell viability in CM+SFC compared to CM-SFC on day 7, we compared the stiffness of the scaffold. In fact, we firstly compared the swelling degrees between the scaffolds to respond to the reviewer's 32nd comment. Then, we found out that the swelling capacity of the scaffold with SFC (Gel+SFC) was significantly higher than that of the scaffold without SFC (Gel-SFC) (Fig. R3). The authors assume that the SFC may lower the crosslinking density of GelMA, resulting higher swelling capacity of a scaffold.

Fig. R3 (Supplementary Fig. 6). Swelling degree of Gel-SFC and Gel+SFC measured after immersing each scaffold in culture media for 24 hours at 37 °C (mean \pm SD, $n = 6$ independent experiments, One-way ANOVA with Tukey method). The swelling degree was calculated by the following equation.

Swelling degree (%)

$$= \frac{(\text{Weight of swollen scaffold} - \text{Weight of lyophilized scaffold}) * 100}{\text{Weight of lyophilized scaffold}}$$

With these results, the authors thought that the stiffness of scaffold would differ which may affect the cell viability. Therefore, we measured the compressive modulus of Gel-SFC and Gel+SFC after swelling in culture medium for 24 hours at 37 °C. Although the compressive modulus was not significantly different between the scaffolds, the mean value of Gel+SFC was about 20 kPa lower than that of Gel-SFC (Fig. R4). Previous research reported that the substrate stiffness had a significant effect on the viability on cells after 3 days of proliferation rather than the initial cell viability¹. In our study, the initial cell viability was also not much different between CM-SFC and CM+SFC but the viability on day 7 was different (Fig. R5c). These results may be due to differences in scaffold stiffness.

Fig. R4 (Supplementary Fig. 7). Stiffness of Gel-SFC and Gel+SFC measured after immersing each scaffold in culture media for 24 hours at 37 °C (mean ± SD, $n = 3$ independent experiments, One-way ANOVA with Tukey method).

Also, we removed the DNA quantification data because CCK results already show the cell viability in the same figure.

Fig. R5 (Fig. 3): Biological evaluation and flavor analysis of cell-cultured scaffold. a

Illustration of the flavor enriching process of the switchable flavor compound (SFC) under cultured meat fabrication process. **b** Immunofluorescence images of proliferated myoblasts on each group. Scale bars: 100 μm . **c** Cell viability on day 1, day 5, and day 7 using CCK-8 assay kit (mean \pm SD, $n = 3$ independent experiments, One-way ANOVA with Tukey method). Grey color indicates cultured meat without SFC (CM-SFC) and red color indicates cultured meat with SFC (CM+SFC). **d** Confocal images showing myosin heavy chain (MHC) and nuclei immunostained with MF20 (red) and DAPI (blue), respectively. Scale bars: 100 μm . **e** Quantitative assessment of MHC amount by bovine myosin-1 enzyme linked immunosorbent assay (ELISA) (mean \pm SD, $n = 3$ independent experiments, One-way ANOVA with Tukey method). MHC amount of the CM+SFC is normalized to that of CM-SFC. **f** Assessment of volatile compounds in CM-SFC and CM+SFC after heating at 150 °C ($n = 3$).

We supplemented this information in the revised manuscript as below:

Related changes in the manuscript:

(Line 231-246)

To quantitatively compare the cell viabilities of the scaffolds, a cell counting kit-8 (CCK-8) assay was performed for CM-SFC and CM+SFC on the 1st, 5th, and 7th day of proliferation (Fig. 3c). The cell viability on day 7 was lower in CM+SFC compared to that of CM-SFC. However, the absorbance value did not decrease from day 1 to day 7 in CM+SFC. These results indicate that rather than cell death in CM+SFC, the proliferation rate was slow in CM+SFC, compared to that of CM-SFC. To further study this difference in cell viability between CM-SFC and CM+SFC, swelling degree and compressive strength of each group were compared (Supplementary Fig. 6 and Supplementary Fig. 7). The radical-based covalent linkage between methacrylate of GelMA and the binding groups of SFC can decrease the crosslinking density of GelMA backbone which can eventually affect the swelling capacity and mechanical properties of the scaffold. Because the compressive modulus of a scaffold affects the long-term cell proliferation, it was hypothesized that the lower cell viability of CM+SFC on day 7 is due to the lower compressive modulus of the scaffold¹. Therefore, the swelling degree and compressive modulus of each scaffold were evaluated. As expected, the swelling capacity of Gel+SFC was higher whereas the compressive modulus was slightly lower, compared to those of Gel-SFC.

[Reference]

1. RSC Advances, 5, 3539-3551 (2016)

Comment 14. Line 217 – 221: Why the author don't analyse a traditional meat for comparison of the volatile profile? These discuss is missing in the paper.

Comment 15. Line 231 – 232: These are not described in the method section. Why some analyses were done with beef for comparison and others not?

Response to the comment 14 and 15:

We appreciate the reviewer's comment. As the reviewer pointed out, the GC-MS was performed for CM-SFC and CM+SFC but not for traditional meat which can arise the inconsistency throughout the manuscript. Therefore, we revised the Fig. 3 (Fig. R6) and Fig. 4 (Fig. R7), and also revised the discussion section. In Fig. 3, we tried to focus on the flavor change depending on the SFC introduction rather than comparing them to traditional meat. Our SFC system is designed to allow volatile compounds with meaty flavor note to be expressed from cultured meat during cooking, rather than specifically aiming to generate the identical volatile compounds as beef. In other words, we focused on the flavor notes of the compounds than on the chemical compound itself. Therefore, rather than emphasizing the consistency of flavor between our cultured meat and traditional meat, we aimed to focus on whether the SFC alters the flavor characteristics of cultured meat. In Fig. 3, the authors emphasized that the volatile compound with meaty flavor is only detected from CM+SFC compared to CM-SFC. In Fig. 4, we removed the flavor analysis results of beef to improve the consistency throughout the manuscript. The flavor analysis results of traditional beef were relocated to the Supplementary Figures (Supplementary Fig. 9 and Supplementary Fig. 10).

Related changes in the manuscript:

(Line 273-279)

The results of Fig. 2 and Fig. 3 confirmed that the Maillard flavor compounds in cultured meat with SFC were stably released upon heating. In conventional meat, various Maillard reaction products with diverse flavor profiles are formed, rather than a single flavor compound.

Therefore, the range of SFC was expanded with three different Maillard reaction molecules to verify if the SFC system can be applied not only to single flavor compound but also to various flavor compounds (Fig. 4a). Specifically, 3-mercapto-2-pentanone and 2-methyl-3-furanthiol and furfuryl mercaptan were introduced into SFC.

(Line 290-315)

CM-SFC, CM+SFC, and CM+SFCV were cooked at 150 °C for 5 min before the flavor evaluation using electronic nose. In Fig. 4b and Supplementary Table 2, the flavor characteristics of CM-SFC, CM+SFC, and CM+SFCV were assessed after heating at 150 °C by electronic nose. Pleasant flavor and offensive flavor were compared, then the ratio of specific flavor notes were analyzed for each group. Compared to the results in Fig. 3f, different and more various flavor compounds were detected due to the different polarity of the used columns in the electronic nose performance. Here, non-polar and mild-polar volatile compounds were detected whereas polar volatile compounds were detected in Fig. 3. As a result, the volatile compounds with offensive flavor (fish, pungent flavor) which were classified based on the flavor compound classification of Fig. 1 were only detected from CM-SFC. On the other hand, the volatile compounds with pleasant flavor were detected from all groups but highly detected from CM+SFC and CM+SFCV. The ratio of the compounds with meaty flavor was higher in CM+SFC and CM+SFCV compared to that of CM-SFC. Especially, the flavor characteristic of CM+SFCV was most similar to that of traditional beef among the three cultured meat specimens (Supplementary Fig. 9). The flavor ratio of the flavor compounds detected according to the Maillard reaction of beef was high in the following order: Meat, floral and creamy, and fruity. This pattern was also confirmed in CM+SFCV. To analyze the flavor similarity between the groups, the principal component analysis (PCA) was performed (Fig. 4c). As a result, it was confirmed that the SFC and SFCV affect the flavor characteristic of the cultured meat. PCA was also performed to demonstrate the similarities in flavor characteristic

between conventional beef and the cultured meat specimens. Also, the flavor difference was identified according to the distance between the area of each cultured meat group and the area of traditional beef shown in the PCA graph (Supplementary Fig. 10). The results show that CM+SFC had a more similar flavor to beef compared to CM-SFC, owing to the SFC volatilization upon heating. CM+SFCV had the most similar flavor pattern to beef among the three cultured meat groups, owing to the introduction of various Maillard reaction compounds.

Fig. R6 (Fig. 3): Biological evaluation and flavor analysis of cell-cultured scaffold. a Illustration of the flavor enriching process of the switchable flavor compound (SFC) under cultured meat fabrication process. **b** Immunofluorescence images of proliferated myoblasts on each group. Scale bars: 100 μ m. **c** Cell viability on day 1, day 5, and day 7 using CCK-8 assay kit (mean \pm SD, n = 3 independent experiments, One-way ANOVA with Tukey method). Grey

color indicates cultured meat without SFC (CM-SFC) and red color indicates cultured meat with SFC (CM+SFC). **d** Confocal images showing myosin heavy chain (MHC) and nuclei immunostained with MF20 (red) and DAPI (blue), respectively. Scale bars: 100 μ m. **e** Quantitative assessment of MHC amount by bovine myosin-1 enzyme linked immunosorbent assay (ELISA) (mean \pm SD, n = 3 independent experiments, One-way ANOVA with Tukey method). MHC amount of the CM+SFC is normalized to that of CM-SFC. **f** Assessment of volatile compounds in CM-SFC and CM+SFC after heating at 150 $^{\circ}$ C (n = 3).

Fig. R7 (Fig. 4): Electronic nose analysis of cultured meat specimens. a Illustration of

cultured meat specimens with different scaffolds: cultured meat without SFC (CM-SFC), cultured meat with SFC (CM+SFC), and cultured meat with flavor variated SFC (CM+SFCV).

b Ratio of the flavor compounds detected from CM-SFC, CM+SFC, and CM+SFCV (mean \pm SD, $n = 3$ independent experiments, One-way ANOVA with Tukey method). Pie chart is shown to present the specific flavor profiles of each group. Flavor notes are presented with different color.

c Principal component analysis (PCA) of the flavor compounds of each group (discrimination index = 90, $n = 3$).

Comment 16. Line 247: Information must be clarified in the methods section.

Response:

We appreciate the reviewer's comment. We supplemented the information of the beef product used in this study in the revised method section as below:

Related changes in the manuscript:

(Line 567-571)

To compare the flavors of the experimental groups to that of beef brisket, an e-nose was used to investigate the flavor pattern of each group. Hanwoo beef brisket was purchased from SIR.LOIN, Korea. For precise comparison. Then, 0.4 g of CM-SFC, CM+SFC, CM+SFCV, and beef brisket were transferred into a 20 mL headspace vial, followed by heating at 150 °C for 5 min.

Comment 17. Line 258: These must be described in the method.

Response:

We appreciate the reviewer’s comment. We agree with the reviewer, and we supplemented the experimental information of the e-nose measurement in the revised method section. To improve the quality of our manuscript and reflect the reviewer’s previous comments, we changed Fig. 4 (Fig. R8). Also, we revised the description of Fig. 4 in the main text and the method section.

Fig. R8 (Fig. 4): Electronic nose analysis of cultured meat specimens. a Illustration of

cultured meat specimens with different scaffolds: cultured meat without SFC (CM-SFC), cultured meat with SFC (CM+SFC), and cultured meat with flavor varied SFC (CM+SFCV). **b** Ratio of the flavor compounds detected from CM-SFC, CM+SFC, and CM+SFCV (mean \pm SD, $n = 3$ independent experiments, One-way ANOVA with Tukey method). Pie chart is shown to present the specific flavor profiles of each group. Flavor notes are presented with different color. **c** Principal component analysis (PCA) of the flavor compounds of each group (discrimination index = 90, $n = 3$).

Related changes in the manuscript:

(Line 273-315)

The results of Fig. 2 and Fig. 3 confirmed that the Maillard flavor compounds in cultured meat with SFC were stably released upon heating. In conventional meat, various Maillard reaction products with diverse flavor profiles are formed, rather than a single flavor compound. Therefore, the range of SFC was expanded with three different Maillard reaction molecules to verify if the SFC system can be applied not only to single flavor compound but also to various flavor compounds (Fig. 4a). Specifically, 3-mercapto-2-pentanone and 2-methyl-3-furanthiol and furfuryl mercaptan were introduced into SFC. 3-mercapto-2-pentanone is also a sulfur compound with meat, onion-like flavors, whereas 2-methyl-3-furanthiol is known to possess fried, nut, and roasted meat-like flavors. It was hypothesized that diversifying the flavor agents in SFCs can increase similarity in flavor characteristics between cultured meat and beef. Therefore, the SFC with three kinds of flavor compounds was synthesized to introduce different Maillard flavor compounds into the scaffold (Supplementary Fig. 8). These three types of SFCs were then introduced into the GelMA hydrogel in the same proportion. Then, cultured meat with flavor varied SFC (CM+SFCV) was obtained after culturing bovine myoblasts on scaffolds with the three SFCs.

CM-SFC, CM+SFC, and CM+SFCV were cooked at 150 °C for 5 min before the flavor

evaluation using electronic nose. In Fig. 4b and Supplementary Table 2, the flavor characteristics of CM-SFC, CM+SFC, and CM+SFCV were assessed after heating at 150 °C by electronic nose. Pleasant flavor and offensive flavor were compared, then the ratio of specific flavor notes were analyzed for each group. Compared to the results in Fig. 3f, different and more various flavor compounds were detected due to the different polarity of the used columns in the electronic nose performance. Here, non-polar and mild-polar volatile compounds were detected whereas polar volatile compounds were detected in Fig. 3. As a result, the volatile compounds with offensive flavor (fish, pungent flavor) which were classified based on the flavor compound classification of Fig. 1 were only detected from CM-SFC. On the other hand, the volatile compounds with pleasant flavor were detected from all groups but highly detected from CM+SFC and CM+SFCV. The ratio of the compounds with meaty flavor was higher in CM+SFC and CM+SFCV compared to that of CM-SFC. Especially, the flavor characteristic of CM+SFCV was most similar to that of traditional beef among the three cultured meat specimens (Supplementary Fig. 9). The flavor ratio of the flavor compounds detected according to the Maillard reaction of beef was high in the following order: Meat, floral and creamy, and fruity. This pattern was also confirmed in CM+SFCV. To analyze the flavor similarity between the groups, the principal component analysis (PCA) was performed (Fig. 4c). As a result, it was confirmed that the SFC and SFCV affect the flavor characteristic of the cultured meat. PCA was also performed to demonstrate the similarities in flavor characteristic between conventional beef and the cultured meat specimens. Also, the flavor difference was identified according to the distance between the area of each cultured meat group and the area of traditional beef shown in the PCA graph (Supplementary Fig. 10). The results show that CM+SFC had a more similar flavor to beef compared to CM-SFC, owing to the SFC volatilization upon heating. CM+SFCV had the most similar flavor pattern to beef among the three cultured meat groups, owing to the introduction of various Maillard reaction compounds.

Related changes in the manuscript:

(Line 581-584)

Flavor profile assignment

The specific flavor note was assigned to each detected volatile compound using the flavor ingredient library of the Flavor and Extract Manufacturers Association of the United States (FEMA) database.

Comment 18. Line 268 – 275: Please revise results and discussion section. It is confusing.

Response:

We appreciate the reviewer's comment. We realized that our results and discussion section are confusing and insufficient. We revised Fig. 4 and the main text. Also, we supplemented the discussion of Fig. 4 in the revised manuscript.

Related changes in the manuscript:

(Line 290-319)

CM-SFC, CM+SFC, and CM+SFCV were cooked at 150 °C for 5 min before the flavor evaluation using electronic nose. In Fig. 4b and Supplementary Table 2, the flavor characteristics of CM-SFC, CM+SFC, and CM+SFCV were assessed after heating at 150 °C by electronic nose. Pleasant flavor and offensive flavor were compared, then the ratio of specific flavor notes were analyzed for each group. Compared to the results in Fig. 3f, different and more various flavor compounds were detected due to the different polarity of the used columns in the electronic nose performance. Here, non-polar and mild-polar volatile compounds were detected whereas polar volatile compounds were detected in Fig. 3. As a result, the volatile compounds with offensive flavor (fish, pungent flavor) which were classified based on the flavor compound classification of Fig. 1 were only detected from CM-SFC. On the other hand, the volatile compounds with pleasant flavor were detected from all groups but highly detected from CM+SFC and CM+SFCV. The ratio of the compounds with meaty flavor was higher in CM+SFC and CM+SFCV compared to that of CM-SFC. Especially, the flavor characteristic of CM+SFCV was most similar to that of traditional beef among the three cultured meat specimens (Supplementary Fig. 9). The flavor ratio of the flavor compounds detected according to the Maillard reaction of beef was high in the following order: Meat, floral and creamy, and fruity. This pattern was also confirmed in CM+SFCV. To analyze the flavor similarity between the groups, the principal component analysis (PCA) was performed (Fig.

4c). As a result, it was confirmed that the SFC and SFCV affect the flavor characteristic of the cultured meat. PCA was also performed to demonstrate the similarities in flavor characteristic between conventional beef and the cultured meat specimens. Also, the flavor difference was identified according to the distance between the area of each cultured meat group and the area of traditional beef shown in the PCA graph (Supplementary Fig. 10). The results show that CM+SFC had a more similar flavor to beef compared to CM-SFC, owing to the SFC volatilization upon heating. CM+SFCV had the most similar flavor pattern to beef among the three cultured meat groups, owing to the introduction of various Maillard reaction compounds.

These results indicate that the strategy of introducing switchable flavor compounds can control the flavor characteristics of cultured meat. Moreover, this approach is not limited to a specific compound: the introduction of various flavor compounds can mimic the complex flavor pattern of conventional meat.

(Line 378-396)

Here, we developed the SFC system to introduce meaty flavor compounds in cultured meat. Also, we further develop the strategy to diversify the flavor compounds of SFC to mimic the flavor properties of traditional meat in Fig. 4. However, there is still a significant difference between the flavor characteristics of cultured meat with SFC and that of traditional meat even with diversification of flavor compounds in SFC. To overcome this limitation, we think two future strategies would be possible: (1) development of SFC with more diverse flavor compounds than SFCV, (2) convergence of the SFC system with the strategy to increase the cell content in the scaffold to take the advantages of cell-derived flavors. As the traditional meat creates dozens of Maillard reaction compounds as flavor compounds when cooked, three kinds of flavor molecules in SFCV would still be insufficient to mimic the flavor of traditional meat perfectly. Therefore, developing the SFC system to admit dozens of Maillard compounds would be one of the future directions. Also, we only focused on the flavor characteristics

depending on SFC in this paper. However, we expect the synergistic effect of SFC and cultured cells if three-dimensional culture strategies to increase cell content are further applied to this study. Through our previous research, we have reported that proteins synthesized by cells participate in the Maillard reaction and affect the flavor characteristics of cultured meat^{1,2}. Therefore, if a strategy to increase the content of the cell-derived proteins is converged into the SFC system reported in this study, it is expected that cultured meat with the flavor of traditional meat will be possible.

[References]

1. Nature Communications, 15, 77 (2024).
2. Matter (2024): Rice grains integrated with animal cells: A shortcut to a sustainable food system.

Comment 19. Line 278 – 288: The discussion must be fully revised, for know it is just final considerations about the results and not a paper data discussion.

Response:

We apologize for the insufficient discussion about the research. We fully revised the section discussing about the findings and limitations of this research. Also, we added a perspective on how this research should develop in the future to the discussion section.

Related changes in the manuscript:

(Line 322-421)

Volatile small molecules, representatively, gas molecules, volatile organic compounds, and flavor compounds, have attracted considerable attention in broad research and application fields: capture of environmentally impacting carbon dioxide gas¹, electrochemical production of hydrogen gas as a green energy resource², deactivation of volatile chemical warfare agents³, disease diagnosis by exhaled breath gas^{4,5}, vascular disease treatment by nitric oxide gas^{6,7}, and carbon monoxide gas-based gastrointestinal inflammation suppression⁸. However, given the utilization in open systems, such as air-permeable cell culture processes, these volatile small molecules suffer from their fast mass transport rate. Therefore, strategies for stable entrapment and controllable release of volatile small molecules have been developed. The emulsion and perfluorocarbon were representatives for physical retention^{9,10}. Meanwhile, because these physical capsules degrade rapidly, the selective generation of the volatile small molecules is a challenge. Although the chemical binding promises significant durability, it requires external energies (*e.g.*, proteolysis¹¹, magnetic force¹², and catalysis¹³) to trigger the volatile compound release on demand.

In this study, we employed robust covalent bonds to secure the flavor stability during a prolonged open system process for cell proliferation and differentiation. Moreover, we devised thermal energy as a desirable trigger resource, given the heating-based cooking and the positive

correlation between temperature and diffusion coefficient of Maillard reaction product. In particular, among the dynamic covalent bonds, such as, imine¹⁴, boronate–ester¹⁵, and hydrazine bonds¹⁶, the disulfide bond was adopted considering its temperature responsivity^{17,18}. Accordingly, this thermo-responsively Maillard reaction product releasing system, so-called SFC, enabled the cell scaffold-based replication of the Maillard reaction of slaughtered meat. Remarkably, the SFC remained stable in the scaffold during the long-term cell culture period. Moreover, it released the flavor compounds such as furfuryl mercaptan selectively at the cooking temperature. At first, we expected only the flavor compounds included in the SFC, furfuryl mercaptan, to be released. However, more diverse flavor compounds with meaty and savory flavor notes were found at 150 °C. These unexpected products would be due to the hydrogen peroxide, an oxidation reagent for forming the disulfide bond in the Maillard reaction product. The hydrogen peroxide would be residual in the CM+SFC with a concentration of 0.386 mM (which is within the food-grade concentration range). When the furfuryl mercaptan is generated upon cooking, the oxidative reactivity of hydrogen peroxide can induce the disulfide bond within the furfuryl mercaptan, forming the 2,2-(dithiodimethylene)difuran¹⁹. Moreover, the hydrogen peroxide could generate the hydroxyl or hydroperoxyl radicals upon heating²⁰. Hence, these radicals could induce the bond scission and thereby result in the diverse volatile compounds with meaty and savory flavor notes, such as 2-methylthiophene or furfuryl methyl disulfide. Finally, we eventually fabricated the flavor-enriched cultured meat after carrying out cell proliferation and differentiation of bovine myoblasts on the SFC-introduced scaffold. Because this SFC exhibited savory and meaty flavors upon volatilization and also generated various flavor molecules by dynamic disulfide exchange, the flavor pattern of cultured meat became similar to that of slaughtered beef.

However, irrelevant to these remarkable advances, this study still encounters a critical limitation. It should be cautious to regard the used reagents as food-grade, although they are

proven to be biocompatible. For instance, the methacrylic anhydride is generally employed to functionalize the methacrylate in the Food and Drug Administration (FDA)-approved proteins (such as gelatin²¹⁻²³, silk fibroin²⁴) and to prepare biocompatible scaffolds via photocrosslinking. However, until recently, no sub-type of GelMA was granted by the FDA for food purposes. Namely, although the biocompatibility of CM+SFC was identified by the primary bovine myoblast experiments, it is still distant from establishing its food safety. Nevertheless, the proposed flavoring strategy yields the potential to be implemented solely by food-grade chemicals. In particular, the enzymatic gelatin crosslink using FDA-approved transglutaminase could be a promising approach to realize the robust binding of Maillard reaction products using only food-grade chemicals²⁵. Moreover, the thermo-responsivity could be fulfilled by the natural disulfide chemistry of cysteine amino acids²⁶. In conclusion, beyond the current stage limitation related to food-grade chemistry, we anticipate that the interesting flavoring strategy developed in this study will contribute to the production of cultured meat that reaches the organoleptic properties of conventional meat.

Here, we developed the SFC system to introduce meaty flavor compounds in cultured meat. Also, we further develop the strategy to diversify the flavor compounds of SFC to mimic the flavor properties of traditional meat in Fig. 4. However, there is still a significant difference between the flavor characteristics of cultured meat with SFC and that of traditional meat even with diversification of flavor compounds in SFC. To overcome this limitation, we think two future strategies would be possible: (1) development of SFC with more diverse flavor compounds than SFCV, (2) convergence of the SFC system with the strategy to increase the cell content in the scaffold to take the advantages of cell-derived flavors. As the traditional meat creates dozens of Maillard reaction compounds as flavor compounds when cooked, three kinds of flavor molecules in SFCV would still be insufficient to mimic the flavor of traditional meat perfectly. Therefore, developing the SFC system to admit dozens of Maillard compounds

would be one of the future directions. Also, we only focused on the flavor characteristics depending on SFC in this paper. However, we expect the synergistic effect of SFC and cultured cells if three-dimensional culture strategies to increase cell content are further applied to this study. Through our previous research, we have reported that proteins synthesized by cells participate in the Maillard reaction and affect the flavor characteristics of cultured meat^{27,28}. Therefore, if a strategy to increase the content of the cell-derived proteins is converged into the SFC system reported in this study, it is expected that cultured meat with the flavor of traditional meat will be possible.

This study focuses on material engineering for controlling the flavor characteristics of cultured meat. Therefore, the use of animal-derived components which is an issue in the cultured meat field is overlooked in this research. However, considering that the cultured meat industry ultimately aims for sustainable meat development, it is essential to discuss the feasibility of our SFC system even without animal-derived materials such as gelatin and serum-containing media. In this study, SFC is conjugated to the GelMA backbone by the reaction between the methacryloyl and the binding group of SFC. The introduction of methacryloyl in gelatin is possible by the substitution of methacrylate groups on the amine-containing side groups of gelatin. Since proteins have amine groups, methacryloyl can also be introduced to plant-derived proteins. Therefore, we expect that SFC system can be possible to various plant-derived proteins. Also, serum-free media are being actively studied in the field of cultured meat. It is anticipated that if non-animal derived nutrients capable of replacing serum in media are developed, it would be able to verify the flavor-enriching effects of the SFC system more precisely. Because serum-containing culture medium contains various animal-derived proteins and glucose, Maillard reaction may occur at 150 °C. The products from the Maillard reaction between animal serum and glucose in culture media might react with the flavor compounds derived from SFC, because the Maillard reaction produces various unstable and highly reactive

volatile compounds²⁹. This reaction might converse the flavor compounds from SFC to non-flavor compounds or might create undesirable or desirable flavor compounds. In Fig. 2 and 3, it can be verified in both CM-SFC and CM+SFC that the number of detected flavor compounds are decreased after culturing cells. This might be due to the effect of volatile compounds created from serum-containing media which remain in the cultured meat specimens. If serum-free medium is applied to this study, unwanted reactions between the volatile compounds from serum-containing media and the compounds from SFC would be excluded, which would help analyzing the effect of SFC system on cultured meat's flavor characteristics more precisely.

[References]

1. Nat. Rev. Mater., 2, 17045 (2017)
2. Nature, 612, 673-678 (2022)
3. Nat. Rev. Chem., 5, 370-387 (2021)
4. ACS Sens., 4, 268-280 (2019)
5. Nat. Nanotechnol., 15, 792-800 (2020)
6. ACS Nano, 17, 8935-8965 (2023)
7. Materials Today, 72, 57-70 (2024)
8. Sci. Transl. Med., 14, ab14135 (2022)
9. Adv. Colloid Interface. Sci., 298, 102544 (2021)
10. Mol. Pharmaceutics, 20, 3254-3277 (2023)
11. Nat. Nanotechnol., 15, 792-800 (2020)
12. Sci. Robot., 5, aaz4239 (2020)
13. J. Am. Chem. Soc., 145, 11019-11032 (2023)
14. Journal of the American Chemical Society 141, 18048-18055 (2019).
15. Proceedings of the National Academy of Sciences 109, 4383-4388 (2012)

16. Journal of the American Chemical Society 135, 17663-17666 (2013)
17. Nature Reviews Materials 5, 562-583 (2020).
18. Journal of the American Chemical Society 144, 2022-2033 (2022)
19. J. Am. Chem. Soc., (2024): Light in a Heartbeat: Bond Scission by a Single Photon above 800 nm
20. Journal of Endodontics, 30, 45-50 (2004)
21. Engineered Regeneration 2, 47-56 (2021)
22. Biomaterials 73, 254-271 (2015)
23. Biomaterials Research 27, 86 (2023)
24. Nat. Commun., 9, 1620 (2018)
25. Nat. Commun., 15, 77 (2024)
26. Biomacromolecules, 23, 926-936 (2022)
27. Nature Communications 15, 77 (2024)
28. Matter (2024): Rice grains integrated with animal cells: A shortcut to a sustainable food system.
29. Trends in Food Science & Technology, 11, 9-10, 364-373, (2000)

Comment 20. Line 290 (Method section): The authors don't describe how much replicates were done in each analysis. This information must be add for all tests.

Response:

We apologize for the insufficient information about the analysis method. We supplemented the information about the replicates in the "Statistics and reproducibility" of the method section as below. Also, we indicated the replicate numbers in the method section:

Related changes in the manuscript:

(Line 585-588)

Statistics and reproducibility

The data are reported as mean \pm standard deviation (SD). Statistical analysis was performed with the significance level of 0.05 using One-way ANOVA with Tukey method via OriginPro 2018 Software. All experiments were repeated three times independently.

(Line 474-487)

The UV quartz cuvette (chamber volume 3.5 mL, path-length 10 mm) was filled with 2 mL of 1.86 μ M SFC. SFC was heated at 37 °C, 80 °C, and 150 °C for 24 h by maintaining the closed system after sealing with the polytetrafluoroethylene stopper. The UV-VIS spectra within 200-600 nm were obtained for the desired heating time points with the settings of bandwidth 1.0 nm, scan speed 240 nm/min, and data interval 1.0 nm. The peak at 335 nm was assigned to the furan-related absorbance. Here, three replication experiments ($n = 3$) were performed for each heating temperature for reliability.

To understand the long-term stability under open system, the vials (chamber volume 5 mL, 18 \times 38 mm²) were filled with 1 mL of 0.37 mM SFC ($n = 3$) and 1 mL of pure furfuryl mercaptan ($n = 3$), respectively. The vials with SFC and furfuryl mercaptan was positioned at 37 °C up to 2 weeks. In particular, to emulate the open system during cell culture procedures, the vials were not closed with a cap, and the air exchange kept sufficient. The weight variation was measured

for specific time points. ^1H NMR (Avance III HD 300, Bruker, Bruker Biospin, USA) spectra of residual SFC were obtained by diluting in the dimethyl sulfoxide- d_6 (10 vol%).

Comment 21. Line 302: Please be specific, how much time was employed?

Response:

We apologize for the unclear information. We revised the sentence to clarify the reaction period as below:

Related changes in the manuscript:”

(Line 435-437)

The reaction was performed at 80 °C for 4 days for the SFC conjugated with furfuryl mercaptan, and for 3 days for the SFC conjugated with 3-mercapto-2-pentanone and with 2-methyl-3-furanthiol.

Comment 22. Line 307: This is considered safe for food-based applications?

Response:

The authors acknowledge entirely your accurate point. The methacrylic anhydride is generally employed to functionalize the methacrylate in the Food and Drug Administration (FDA)-approved proteins (such as gelatin¹⁻³, silk fibroin⁴) and to prepare biocompatible scaffolds via photo-crosslinking. Namely, its superior biocompatibility was well-proved. Nevertheless, it is still cautious to regard methacrylic anhydride as safe for food applications, and the Food and Drug Administration (FDA) has not yet approved it as a food-grade chemical. Accordingly, we revised the manuscript to avoid the overexpressed relationship between biocompatibility and food-grade safety. Furthermore, in the discussion section, we suggested an idea to realize the proposed SFC-based flavoring strategy using the food-grade reagents: transglutaminase to substitute the methacrylic anhydride and cysteine disulfide bonds to impart thermoresponsivity^{5,6}.

Related changes in the manuscript:

(Line 362-377)

However, irrelevant to these remarkable advances, this study still encounters a critical limitation. It should be cautious to regard the used reagents as food-grade, although they are proven to be biocompatible. For instance, the methacrylic anhydride is generally employed to functionalize the methacrylate in the Food and Drug Administration (FDA)-approved proteins (such as gelatin¹⁻³, silk fibroin⁴) and to prepare biocompatible scaffolds via photo-crosslinking. However, until recently, no sub-type of GelMA was granted by the FDA for food purposes. Namely, although the biocompatibility of CM+SFC was identified by the primary bovine myoblast experiments, it is still distant from establishing its food safety. Nevertheless, the proposed flavoring strategy yields the potential to be implemented solely by food-grade chemicals. In particular, the enzymatic gelatin crosslink using FDA-approved transglutaminase

could be a promising approach to realize the robust binding of Maillard reaction products using only food-grade chemicals⁵. Moreover, the thermo-responsivity could be fulfilled by the natural disulfide chemistry of cysteine amino acids⁶. In conclusion, beyond the current stage limitation related to food-grade chemistry, we anticipate that the interesting flavoring strategy developed in this study will contribute to the production of cultured meat that reaches the organoleptic properties of conventional meat.

[References]

1. *Engineered Regeneration*, **2**, 47-26 (2021)
2. *Biomaterials*, **73**, 254-271 (2015)
3. *Biomaterials Research*, **27**, 86 (2023)
4. *Nat. Commun.*, **9**, 1620 (2018)
5. *Nat. Commun.*, **15**, 77 (2024)
6. *Biomacromolecules*, **23**, 926-936 (2022)

Comment 23. Line 310: Revise English language.

Response:

We apologize for the inappropriate English language. We change the word in the revised manuscript as below:

Related changes in the manuscript:

(Line 450-452)

Then, the GelMA solution was dialyzed for 5 days at 80 °C hot plate using a 12-14 kDa membrane (Thermo Fisher, #08667E). Distilled water was used as dialysate.

Comment 24. Line 324: Please clarify the abbreviations in the previous paragraph.

Response:

We appreciate the reviewer's comment. We clarified the abbreviations of the sample names in the revised method section as below:

Related changes in the manuscript:

(Line 489-492)

To assess the stability of the flavor molecules in the hydrogels, the three types of samples (GelMA scaffold without switchable flavor compound (Gel-SFC), GelMA scaffold with switchable flavor compound (Gel+SFC), and GelMA scaffold with furfuryl mercaptan mixed (Gel+FM)) were immersed in distilled water for the whole cell culture period (15 days).

Comment 25 (1) Line 325: DW mimics the cell culture medium?

(2) At which conditions?

Response:

As the reviewer pointed out, distilled water does not mimic the cell culture medium consisting various components like animal serum, amino acids, growth factors, glucose, etc. In Fig. 2d, the flavor compound in the scaffolds were analyzed after immersing in distilled water for 15 days. Distilled water was replaced every two days. Because culture medium contains animal serum which is composed of various animal-derived proteins, culture medium itself can create flavor compounds when heated. From our previous research, we confirmed that even the increase in cell-derived proteins can affect the flavor profile of the cultured meat by participating in the Maillard reaction¹. The products from the Maillard reaction between animal serum and glucose in culture media might react with the flavor compounds derived from SFC, because the Maillard reaction produces various unstable and highly reactive volatile compounds². This reaction might converse the flavor compounds from SFC to non-flavor compounds or might create undesirable or desirable flavor compounds. In Fig. 2 and 3, it can be verified in both CM-SFC and CM+SFC that the number of detected flavor compounds are decreased after culturing cells with culture medium. This might be due to the effect of volatile compounds created from serum-containing media which remain in the cultured meat specimens. Therefore, distilled water was used to compare the flavor characteristics of Gel-SFC, Gel+SFC, and Gel+FM, excluding the effect of culture medium. Then, in Fig. 3 and Fig. 4, the flavor characteristics of specimens were compared after cell culturing with culture medium. In summary, we conducted DW-based experiments in Fig. 2 as a model study to figure out the thermal responsivity of SFC solely. The cell culture media-based experiments in Fig 3-4 demonstrated the practical potential of CM+SFC evaluated under the real-world cultured meat production process.

We included this in the revised discussion section as below:

Related changes in the manuscript:

(Line 407-421)

Also, serum-free media are being actively studied in the field of cultured meat. It is anticipated that if non-animal derived nutrients capable of replacing serum in media are developed, it would be able to verify the flavor-enriching effects of the SFC system more precisely. Because serum-containing culture medium contains various animal-derived proteins and glucose, Maillard reaction may occur at 150 °C. The products from the Maillard reaction between animal serum and glucose in culture media might react with the flavor compounds derived from SFC, because the Maillard reaction produces various unstable and highly reactive volatile compounds². This reaction might converse the flavor compounds from SFC to non-flavor compounds or might create undesirable or desirable flavor compounds. In Fig. 2 and 3, it can be verified in both CM-SFC and CM+SFC that the number of detected flavor compounds are decreased after culturing cells. This might be due to the effect of volatile compounds created from serum-containing media which remain in the cultured meat specimens. If serum-free medium is applied to this study, unwanted reactions between the volatile compounds from serum-containing media and the compounds from SFC would be excluded, which would help analyzing the effect of SFC system on cultured meat's flavor characteristics more precisely.

[References]

1. Nature Communications, 15, 77 (2024)
2. Trends in Food Science & Technology, 11, 9-10, 364-373, (2000)

Comment 26. Line 327: What kind of heating was used? Please describe the equipment used.

Response:

We apologize for the insufficient information. We used a heating plate for temperature control. Specifically, we covered the heating plate with aluminum foil and adjusted the temperature. When the temperature reached to the setting point, we put the samples on the foil covered plate for 5 minutes. We supplemented this information in the revised manuscript as below:

Related changes in the manuscript:

(Line 492-496)

Then, the hydrogels were transferred to a vial and kept at room temperature or heated at 150 °C for 5 min. A heating plate was used for temperature control. Specifically, the heating plate was covered with aluminum foil and the temperature was adjusted. When the temperature reached to the setting point, the samples were put on the foil covered plate for 5 minutes.

Comment 27. Line 331 – 332: In what equipment? Please describe.

Response:

We apologize for the insufficient information. For the heating the sample to extract the volatile compounds during GC-MS performance, an autosampler (Agilent 7693, CombiPAL sampler 80, Agilent) was used. We supplemented this information in the revised manuscript.

Related changes in the manuscript:

(Line 501-503)

At this time, the groups that were placed at room temperature were heated to 30 °C for 20 min, while the others were heated to 80 °C for 20 min using the autosampler (Agilent 7693, CombiPAL sampler 80, Agilent) to allow the volatile compounds to be adsorbed onto the fibers.

Comment 28. Line 334 – 338: What was the column used? Please add complete information of the methods. The same for the gas used (mobile phase).

Response:

We apologize for the insufficient information about the GC-MS performance. Firstly, the fibers were injected into the inlet at the temperature of 250 °C using the split (20:1) injection mode. The initial temperature of the column oven was 40 °C and hold time was 5 min. Then, the temperature increased to 240 °C at a rate of 4 °C/min, and the set temperature was hold for 20 min to analyze the volatile compounds. The GC column used in this experiment was DB-WAX (Agilent 123-7063). Carrier gas flow was He (99.999%) at initial flow of 2 mL/min and post run flow of 1 mL/min. We added this information in the revised method section as below:

Related changes in the manuscript:

(Line 505-514)

Subsequently, the volatile compounds adsorbed onto the fibers were analyzed. Helium with 99.999% purity was used for the carrier gas with the split flow rate of 40 mL/min. Fibers were injected into the inlet at the temperature of 250 °C using the split (20:1) injection mode. The initial temperature of the column oven was 40 °C and hold time was 5 min. Then, the temperature increased to 240 °C at a rate of 4 °C/min, and the set temperature was hold for 20 min to analyze the volatile compounds. The GC column used in this experiment was DB-WAX (Agilent 123-7063). The temperature of the column oven was kept at 40 °C for 5 min, and then increased to 240 °C at a rate of 4 °C/min. Finally, the column oven temperature was kept at 240 °C for 20 min to analyze the volatile and semivolatile compounds of each group.

Comment 29. Line 356: What was the area of the scaffold?

Response:

We apologize for the incomplete information about the scaffold. The area of the scaffold was 16.6 mm². We revised the sentence as below:

Related changes in the manuscript:

(Line 526-527)

Myoblasts at passage 3 were seeded on the scaffolds at a density of 2×10^4 cell/scaffold (16.6 mm² x 0.8 mm).

Comment 30. Line 359: Please clarify this information, from 48h to 48h?

Response:

We apologize for the unclear description in the method section. For the cell culture procedure, we changed the culture medium every two days. We supplemented this information in the revised manuscript as below:

Related changes in the manuscript:

(Line 527-528)

Growth medium was replaced every two days to feed the cells during proliferation.

(Line 531)

Differentiation medium was replaced every two days.

Comment 31. Line 384: Why CM+SFCV was not analysed? (for GC-MS analysis)

Response:

We appreciate the reviewer's comment. As the reviewer pointed out, CM+SFCV was not analyzed for the GC-MS analysis. For the GC-MS data, the authors tried to show that our strategy of introducing SFC can enrich the flavor of cultured meat compared to the one that does not contain SFC. Therefore, we focused on comparing the GC-MS results comparing the CM-SFC and CM+SFC only. Furthermore, we intentionally designed CM+SFCV to demonstrate the potential of flavor diversification through our SFC-based strategy. Thanks to your kind comment, we realized that it can be confused that the analysis on CM+SFCV is suddenly performed in Fig. 4 with traditional beef. Therefore, we revised Fig. 4 with newly written description and discussion (Fig. R9). The revised Fig. 4 mainly focuses on the diversification of SFC system by introducing more types of flavor compound in SFC molecule, rather than comparing the flavor characterization of the cultured meat with that of beef. We also explain that after confirming the effect of SFC system on the flavor characteristics of cultured meat through Fig. 2 and Fig. 3, we apply this system to more various flavor compounds in Fig. 4 because traditional meat expresses diverse Maillard flavor compounds rather than a single compound.

Related changes in the manuscript:

(Line 273-279)

The results of Fig. 2 and Fig. 3 confirmed that the Maillard flavor compounds in cultured meat with SFC were stably released upon heating. In conventional meat, various Maillard reaction products with diverse flavor profiles are formed, rather than a single flavor compound. Therefore, the range of SFC was expanded with three different Maillard reaction molecules to verify if the SFC system can be applied not only to single flavor compound but also to various flavor compounds (Fig. 4a). Specifically, 3-mercapto-2-pentanone and 2-methyl-3-furanthiol

and furfuryl mercaptan were introduced into SFC.

Fig. R9 (Fig. 4): Electronic nose analysis of cultured meat specimens. **a** Illustration of cultured meat specimens with different scaffolds: cultured meat without SFC (CM-SFC), cultured meat with SFC (CM+SFC), and cultured meat with flavor varied SFC (CM+SFCV). **b** Ratio of the flavor compounds detected from CM-SFC, CM+SFC, and CM+SFCV (mean \pm SD, $n = 3$ independent experiments, One-way ANOVA with Tukey method). Pie chart is shown

to present the specific flavor profiles of each group. Flavor notes are presented with different color. **c** Principal component analysis (PCA) of the flavor compounds of each group (discrimination index = 90, $n = 3$).

Comment 32. Line 386: How much was placed in the vial? How about the proximal composition of the cultured meat? How much water the samples presented? Please describe.

Response:

We apologize for the insufficient information. For the flavor analysis, one sample was placed in one 20 mL vial for both CM-SFC and CM+SFC. The proximal composition of the cultured meat specimen were not measured in our system, but the authors expect that most of the composition of CM-SFC and CM+SFC would be protein. Because the CM specimens are mostly comprised of the gelatin-based scaffold, the main constituent of both CM-SFC and CM+SFC would be protein. To reflect the reviewer's comment, we measured the weight of the scaffolds (Gel-SFC and Gel+SFC) before and after swollen in culture media at 37 °C for 24 hours. As a result, we found out that the swelling degrees of Gel-SFC and Gel+SFC were approximately 570 % and 651 %, respectively (Fig. R10). Due to this difference of swelling capacity, the stiffness of Gel+SFC was slightly lower than that of Gel-SFC (Fig. R11).

Fig. R10 (Supplementary Fig. 6): Swelling degree of Gel-SFC and Gel+SFC measured after immersing each scaffold in culture media for 24 hours at 37 °C (mean \pm SD, $n = 6$ independent experiments, One-way ANOVA with Tukey method). The swelling degree was calculated by the following equation.

Swelling degree (%)

$$= \frac{(\text{Weight of swollen scaffold} - \text{Weight of lyophilized scaffold}) * 100}{\text{Weight of lyophilized scaffold}}$$

Fig. R11 (Supplementary Fig. 7): Stiffness of Gel-SFC and Gel+SFC measured after immersing each scaffold in culture media for 24 hours at 37 °C (mean \pm SD, $n = 3$ independent experiments, One-way ANOVA with Tukey method).

The higher swelling degree of Gel+SFC compared to Gel-SFC may be due to the SFC inhibiting the crosslinking between GelMA chains. We supplemented this information in the revised manuscript as below:

Related changes in the manuscript:

(Line 239-246)

The radical-based covalent linkage between methacrylate of GelMA and the binding groups of SFC can decrease the crosslinking density of GelMA backbone which can eventually affect the swelling capacity and mechanical properties of the scaffold. Because the compressive modulus of a scaffold affects the long-term cell proliferation, it was hypothesized that the lower cell viability of CM+SFC on day 7 is due to the lower compressive modulus of the scaffold¹. Therefore, the swelling degree and compressive modulus of each scaffold were evaluated (Supplementary Fig. 6 and Supplementary Fig. 7). As expected, the swelling capacity of

Gel+SFC was higher whereas the compressive modulus was slightly lower, compared to those of Gel-SFC.

[References]

1. RSC advances, 6, 3539-3551 (2016).

Comment 33. Line 398 – 399: Please insert all details of the method.

Response:

We apologize for the insufficient information of the e-nose experiment. We supplemented the information in the revised manuscript as below:

Related changes in the manuscript:

(Line 572-580)

The injector temperature was 200 °C and the inject time was 45 seconds. For carrier gas supply, H₂ was used for the injector as well as for the detector. Two independent chromatographic columns and two flame ionization detectors were used for compound detection. The non-polar column was MXT-5 GC metal capillary column (Restek) was used whereas the second column with medium polarity was MXT-1701 GC metal capillary column (Restek). For the detectors, flame ionization detector (FID)1 (non-polar) and FID2 (slightly polar) were used, and the detector temperature was 260 °C. The detected compounds were then analyzed using the Alpha Software (Alpha MOS, France).

Response to Reviewer #2

Reviewer(s)'s Comments to Author:

Reviewer 2.

Comments

In this study, the authors generate thermal-responsive gelatin-based scaffolds which, through temperature sensitive covalent linking, are able to release key flavor compounds upon cooking. They demonstrate that these flavor compounds remain stably linked to the scaffolds over long-term cell culture with bovine muscle cells, and that only when the resulting cultured meat constructs are cooked are the compounds released. The authors show that this release is associated with an improvement in overall aromatic likeness to conventional beef. This is a clever and innovative approach, which is indeed novel to the field. As such, it shows substantial promise as an exciting contribution to the literature. That said, there are several minor and major revisions that the authors must perform before the manuscript is sufficiently clear and detailed for publication.

Response:

The authors sincerely appreciate the reviewer's valuable comments. By carefully reflecting the reviewer's comments, we realized that the clarity of the results should be improved and that the research discussion is very lacking. We regret our insufficient information and ambiguous descriptions. To improve the quality of our manuscript, we fully revised the manuscript and all the figures. We removed inappropriate words and revised the sentences in the result section to improve the clarity. Also, we revised the figures to improve the consistency of the data. We supplemented the discussion section with our research discussion about the biological data, chemical toxic issues, and use of non-food grade materials. We also carefully discussed about

the future study to further increase the flavor similarity with conventional meat in the revised manuscript. Again, we sincerely appreciate the reviewer for all the valuable comments that helped us improve the quality of our manuscript.

To respond to all the valuable comments from the reviewer in detail and sincerely, point-by-point responses to the reviewer's comments were written below. The changes in the revised manuscript related each comment are presented in blue font color. Once again, we appreciate the reviewer for giving us the opportunity to revise our manuscript.

Minor revisions:

Comment 1. Throughout the manuscript are several areas where editing for clarity and readability would be helpful. I have suggested some specific changes in the attached markup document, but the authors are welcome to make their own changes, if they do not deem the markup to be optimal.

Response:

The authors appreciate the reviewer's valuable comments. We apologize for the unclear descriptions in the manuscript. We carefully reflected the reviewer's valuable comments in the attached markup document, and we revised the manuscript accordingly.

Comment 2. Line 46 states that “cultured meat is far from being recognized as a meat product,” however, two products have already been approved for sale as meat (in the USA and/or Singapore) and regulatory agencies label these as “cell cultured meat” (for instance, in the USA). As such, it is incorrect to say that cultured meat is not recognized as a meat product.

Response:

We apologize for the expressions making such a strong conclusion. We agree with the reviewer’s comment. We removed the following sentence, “Therefore, despite the efforts of many researchers, cultured meat is still far from being recognized as a meat product.”

Comment 3. Line 53-54 states that cultured meat faces an “absence of the Maillard reaction;” however, no evidence is provided for this. Indeed, it is likely that cultured meat produced even without temperature-sensitive hydrogels would undergo some Maillard reaction (for instance, there is suggested evidence of a maillard reaction in DOI: 10.1016/j.ijbiomac.2023.129134 and again in DOI: 10.1038/s41467-023-44359-9 and DOI: 10.1021/acsami.2c10988, which are the authors own work). The authors should remove this statement, as the evidence suggests otherwise.

Response:

We appreciate for the reviewer’s comments. We apologize for the strong statement without evidence. We agree with the reviewer’s comment about cultured meat without our system can also have Maillard reaction. To avoid making strong statements about other cultured meat system, we revised the setence as below:

Related changes in the manuscript:

(Line 54-65)

Therefore, mimicking the Maillard flavor in cultured meat is also required to increase the sensorial similarity between cultured meat and traditional meat. Recently, many papers reported to improve the flavor characteristics of cultured meat. For example, Joo et al. compared the taste of cultured muscle tissue obtained from chicken satellite cells to that of traditional chicken¹. However, they revealed that it is challenging to mimic the taste of traditional meat in solely cultured tissue without any treatment due to the different amino acid composition. Liu et al. also reported that cultured fat had fruity and creamy flavor characteristics due to the lipid oxidation of fatty acids formed in cultured fat². Recently, our group revealed that the enhanced proliferation and differentiation of murine and bovine myoblasts could induce relatively more meat-like flavors in cultured meat^{3,4}. Nevertheless, in order for cultured meat to present the flavor characteristics of well-cooked meat products, a

strategy to generate the Maillard reaction compounds during cooking is still needed.

[References]

1. Food science of animal resources, 42, 175 (2022).
2. Journal of Agricultural and Food Chemistry, 71, 4113-4122 (2023).
3. ACS Applied Materials & Interfaces, 14, 38235-38245 (2022).
4. Nature Communications, 15, 77 (2024).

Comment 4. Line 55 states that “the amino acid types of cells are not as diverse as those of meat;” however, no evidence is provided to validate this claim. Indeed, this paper: (DOI: 10.5851/kosfa.2021.e72) suggests that all amino acids present in traditional beef or chicken are also present in cultured beef or chicken. The authors must back up their claim with a reference from the literature, or else remove it.

Response:

We apologize for the insufficient evidence. The reviewer provided us with a good reference, allowing us to reconsider our claim and change the sentence effectively. From the reference provided by the reviewer, the paper shows that all the amino acids composing traditional meat are also present in the cultured tissue. However, this paper also say that the amount of some amino acids in cultured meat differ from that of traditional meat which resulted different taste characteristics from traditional meat. Also, Zheng et al. showed that amino acid composition of cultured meat differs from that of conventional meat¹. Therefore, we revised the sentence as below with supplemented references:

Related changes in the manuscript:

(Line 52-59)

The difference in amino acid profiles between in vitro tissues and traditional meat presents challenges in mimicking the Maillard flavor of traditional meat in the field of cultured meat^{1,2}. Therefore, mimicking the Maillard flavor in cultured meat is also required to increase the sensorial similarity between cultured meat and traditional meat. Recently, many papers reported to improve the flavor characteristics of cultured meat. For example, Joo et al. compared the taste of cultured muscle tissue obtained from chicken satellite cells to that of traditional chicken². However, they revealed that it is challenging to mimic the taste of traditional meat in solely cultured tissue without any treatment due to the different amino acid composition.

[References]

1. Food Research International, 161, 111818 (2022).
2. Food Science of Animal Resources, 42, 1, 175-185 (2022).

Comment 5. In line 68, the authors claim a two-week period. That may be what the authors use in this paper, but it is not universal to all cultured meat production processes. A wider range should be mentioned, given the range of timelines that might be required for cultured meat production.

Response:

We appreciate for the reviewer's comment. We agree with the reviewer's comment on the cultured meat production period. The two weeks cell culture period in our system may not be applied to other cultured meat research since cell culture period depends on the cell types, seeding density, media composition, etc. Therefore, we revised the sentence to explain that the cell culture period maybe longer than 2 weeks in other culture system. Also, we provided the range of cell culture timelines with references of other cultured meat research. The revised sentence and the references are shown below:

Related changes in the manuscript:

(Line 73-77)

The long cell culture period could accelerate the loss of flavor, thus leaving no residual flavor compounds within prepared cultured meat. The total cell culture period was 15 days in this work, and it may take 28 days or more for cell proliferation and differentiation depending on the culture system^{1,2}. Therefore, a material science approach to selectively generate the flavor compounds during the cooking process is essential.

[References]

1. Biomaterials, 285, 121543 (2022).
2. Biomaterials, 284, 121487 (2022).

Comment 6. Line 90 should include a scale-bar of the photograph of cultured meat.

Response:

We appreciate the reviewer’s comment. We sincerely apologize for the lack of scale bar of the cultured meat image. In the revised Fig. 1, we changed the image of cultured meat with better quality, and we also revised the caption of Fig. 1 to include the scale bar of the image.

Revised Figure:

Fig. R1 (Fig. 1): Schematic illustration of switchable flavor system. a Illustrative

description of scaffold structure conjugated with switchable flavor compound (SFC). **b** Photograph of cultured meat fabricated using the scaffold with SFC (CM+SFC). Scale bar: 8 mm. **c** Illustration explaining the mechanism of SFC system depending on temperature. **d** Classification of the flavor compounds analyzed in this study.

Comment 7. Throughout, the authors use active voice rather than passive voice (e.g., line 107 says “we investigated the temperature responsivity” rather than something like “temperature responsiveness was investigated”). The latter would be more appropriate. Throughout the manuscript, the authors should change to passive voice (avoiding reference of “we”).

Response:

We appreciate the reviewer’s comments. We replaced the active expression with passive expression as the reviewer pointed out. Also, we revised the manuscript to avoid the word “we” as below:

Related changes in the manuscript:

(Line 129-130)

The switchable flavor compound (SFC) capable of generating the Maillard reaction products in response to heating and cooking was designed.

(Line 165-167)

After confirming the stability of the SFC itself, the flavor stability of the gelatin-based hydrogel containing SFC(Gel+SFC) was evaluated. The hydrogel without SFC (Gel-SFC) was also produced as a control group.

(Line 172-174)

To demonstrate the enhanced flavor stability within the hydrogel achieved through our strategy, a hydrogel just physically mixed with pure furfuryl mercaptan (Gel+FM) was also produced.

(Line 133-135)

In case of the thermal responsive group (R₃), a disulfide bond to the roasted meat-flavored Maillard reaction product, furfuryl mercaptan, was introduced (Supplementary Fig. 1a).

(Line 143-144)

As shown in Fig. 2b, the temperature responsiveness of the SFC was investigated by heating at 37 °C, 80 °C, or 150 °C up to 24 h in a closed system.

(Line 153-154)

It was evaluated whether the Maillard reaction product was stably bound to the SFC during a prolonged cell cultured period at 37 °C.

(Line 178-181)

Then, the flavor stability of the hydrogels at aqueous condition was evaluated by gas chromatography–mass spectrometry (GC–MS) after immersing them in distilled water for 15 days, corresponding to the myoblast culture period (Fig. 2d and Supplementary Fig. 5).

(Line 181-183)

To assess the flavor-releasing capability of the hydrogel as a function of the temperature, the concentration of the volatile compounds from each group at room temperature and at the Maillard reaction temperature (150 °C) was measured by GC-MS.

Comment 8. Line 109: “significant” should be changed to some other word, such as “notable” or “noticeable” or “detectable,” as no statistics were performed.

Response:

We appreciate the reviewer’s comment. We revised the words “significant” to “noticeable” as suggested by the reviewer.

Related changes in the manuscript:

(Line 144-145)

In the 37 °C heating case, the SFC hardly yielded any noticeable signals.

Comment 9. Line 148 (example, though there are other instances as well). There are some scenarios where “flavor” is used where “flavor compound” or “aroma” should be used instead. For instance, in 148, the authors say that “flavor was detected,” however, since flavor is a sense rather than a chemical, this should either be “aroma” or “flavor compound”

Response:

We sincerely appreciate the reviewer’s comment. Through out the manuscript, the word “flavor” is replaced with “flavor compound” when explaining flavor detection. The revised sentences are shown below:

Related changes in the manuscript:

(Line 191-204)

Before heating, the concentration of **the compounds with offensive flavor** was dominant for Gel-SFC. The concentration of **these compounds** was decreased whereas the concentration of **volatile compounds with pleasant flavor** was increased after heating. Nevertheless, the **offensive flavor compounds** still accounted for approximately 1.4% in Gel-SFC. Also, the concentration of **the flavor compounds among total volatile compounds** detected from Gel-SFC was decreased upon heating. For Gel+SFC, the increase in concentration of **pleasant flavor compounds** after heating was confirmed which indicates that the introduction of SFC can provide pleasant flavor characteristics to the scaffold. The temperature-responsive release of the flavor compounds was only confirmed in Gel+SFC. At room temperature, **no flavor compounds** were detected from Gel+SFC. Upon heating at the Maillard reaction temperature, high level of the **flavor compounds with pleasant flavor** notes such as meat, savory, almond, roasted bread, floral, cheese and fat were detected from Gel+SFC. On the other hand, **the flavor compound** release before heating was detected for the Gel+FM group, indicating that flavor loss can occur during the cell culture period.

Comment 10. Line 162: the authors should specify the name of the maillard reaction product (e.g., chemical name) in the figure legend.

Response:

We appreciate the reviewer’s comment. We revised Fig. 1 to include all the chemical names and structures of volatile compounds analyzed in our study (Fig. R2). Also, we added the information of the detected compounds in each figure to the supplementary figures (Table R1).

(Fig. R2) Fig. 1: Schematic illustration of switchable flavor system. a Illustrative

description of scaffold structure conjugated with switchable flavor compound (SFC). **b** Photograph of cultured meat fabricated using the scaffold with SFC (CM+SFC). Scale bar: 8 mm. **c** Illustration explaining the mechanism of SFC system depending on temperature. **d** Classification of the flavor compounds analyzed in this study.

Scaffold type	Temperature	Flavor description	Detected flavor compound (IUPAC name)
Gel-SFC	25 °C	Meaty, Savory	Not detected
		Almond, Roasted bread	Not detected
		Floral, Cheese, Fat	Not detected
		Fishy, Pungent, Sour	Acetone
	150 °C	Meaty, Savory	Not detected
		Almond, Roasted bread	Benzaldehyde, Benzyl alcohol (Phenylmethanol)
		Floral, Cheese, Fat	Nonanoic acid, Octanoic acid
		Fishy, Pungent, Sour	Hexanoic acid
Gel+FM	25 °C	Meaty, Savory	Furfuryl mercaptan (furan-2-ylmethanethiol), Furfural (furan-2-carbaldehyde)
		Almond, Roasted bread	Not detected
		Floral, Cheese, Fat	Not detected
		Fishy, Pungent, Sour	Not detected
	150 °C	Meaty, Savory	Furfuryl mercaptan (furan-2-ylmethanethiol), Furfural (furan-

			2-carbaldehyde), 2,2-(dithiodimethylene)difuran
		Almond, Roasted bread	Not detected
		Floral, Cheese, Fat	Nonanoic acid
		Fishy, Pungent, Sour	Not detected
Gel+SFC	25 °C	Meaty, Savory	Not detected
		Almond, Roasted bread	Not detected
		Floral, Cheese, Fat	Not detected
		Fishy, Pungent, Sour	Not detected
	150 °C	Meaty, Savory	Furfuryl mercaptan (furan-2-ylmethanethiol), Furfural (furan-2-carbaldehyde), 2,2-(dithiodimethylene)difuran
		Almond, Roasted bread	Benzaldehyde, Benzyl alcohol (Phenylmethanol)
		Floral, Cheese, Fat	Nonanoic acid, Heptanoic acid, Octanoic acid
		Fishy, Pungent, Sour	Not detected

Table R1 (Supplementary Table 1): Flavor compounds detected from the hydrogel without switchable flavor compound (Gel-SFC), hydrogel with SFC (Gel+SFC), and hydrogel mixed with pure furfuryl mercaptan (Gel+FM) depending on the temperature.

Comment 11. Line 199: It is not clear to me why cell infiltration would result in insufficient CCK-8 data. If the sponges are porous, then transfer of metabolites into/out of the scaffolds should be possible, and so the assay should work as expected.

Response:

As the reviewer pointed out, CCK-8 data is sufficient to show the cell viability in this research. Therefore, we removed the DNA quantification results in the revised manuscript as below:

Fig. R3 (Fig. 3): Biological evaluation and flavor analysis of cell-cultured scaffold. a Illustration of the flavor enriching process of the switchable flavor compound (SFC) under cultured meat fabrication process. **b** Immunofluorescence images of proliferated myoblasts on each group. Scale bars: 100 μm. **c** Cell viability on day 1, day 5, and day 7 using CCK-8 assay kit (mean ± SD, n = 3 independent experiments, One-way ANOVA with Tukey method). Grey

color indicates cultured meat without SFC (CM-SFC) and red color indicates cultured meat with SFC (CM+SFC). **d** Confocal images showing myosin heavy chain (MHC) and nuclei immunostained with MF20 (red) and DAPI (blue), respectively. Scale bars: 100 μ m. **e** Quantitative assessment of MHC amount by bovine myosin-1 enzyme linked immunosorbent assay (ELISA) (mean \pm SD, $n = 3$ independent experiments, One-way ANOVA with Tukey method). MHC amount of the CM+SFC is normalized to that of CM-SFC. **f** Assessment of volatile compounds in CM-SFC and CM+SFC after heating at 150 $^{\circ}$ C ($n = 3$).

Comment 12. Line 216: Were no other compounds from the scaffold detected? Only this singular molecule?

Response:

In Fig. 3f, only benzaldehyde was detected from CM-SFC. However, more various flavor compounds were detected from CM-SFC when analyzed by electronic nose in Fig. 4. This difference is due to the different column used in GC-MS and electronic nose. In GC-MS, polar column (DB-WAX) was used which is appropriate for detecting polar compounds. Non-polar column (MXT-5) and mild polar column (MXT-1701) were used for electronic nose which detect non-polar and mild polar compounds, respectively. Therefore, the detected flavor compounds are more diverse in Fig. 4 for both CM-SFC and CM+SFC. We included this information in result section to explain the inconsistency of the flavor analysis results in Fig. 3 and Fig. 4.

Related changes in the manuscript:

(Line 294-297)

Compared to the results in Fig. 3f, different and more various flavor compounds were detected due to the different polarity of the used columns in the electronic nose performance. Here, non-polar and mild-polar volatile compounds were detected whereas polar volatile compounds were detected in Fig. 3.

Comment 13. Line 220: It would be helpful to clarify which flavor compounds these were with their chemical names, as it would help the reader track these compounds throughout the paper.

Response:

We appreciate the reviewer's comment. We agree with the reviewer, and we revised the Figures to include the chemical names of volatile compounds with flavors to improve the clarity of the results. Firstly, we added the chemical names and structures of detected flavor compounds in this study in Fig. 1 (Fig. R4). We classified the compounds according to the flavor note assigned to each compound by the Flavor and Extract Manufacturers Association of the United States (FEMA) database. The flavor notes are also classified into two main flavor notes: Offensive flavor and pleasant flavor. Offensive flavor includes fishy, pungent, and sour flavor notes derived from trimethylamine, acetone, and hexanoic acid which are known as unpleasant and irritating odor¹⁻⁴. Pleasant flavor includes meaty, sulfurous, almond-like, fatty, and fruity flavor notes which are known to be detected from traditional meat⁵⁻⁷. Throughout the manuscript, all the flavor analyses in Fig. 2, Fig. 3, and Fig. 4 were performed according to this flavor compound classification in Fig. 1. Also, we added the information of flavor compounds analyzed in each figure in Supplementary Table 1 and Supplementary Table 2 (Table R2 and Table R3) to help the readers to track the compounds.

[References]

1. Human brain mapping, 17, 1, 17-27 (2002)
2. International Journal of Food Science + Technology, 57, 4, 2277-2284 (2022)
3. LWT, 105, 16-22 (2019)
4. The Journal of Clinical and Aesthetic Dermatology, 6, 11, 45-48 (2013).
5. npj Science of Food, 7, 13 (2013)

6. Nature Communications, 15, 77 (2024)

7. Food Research International, 157, 111385 (2022)

Fig. R4 (Fig. 1): Schematic illustration of switchable flavor system. **a** Illustrative description of scaffold structure conjugated with switchable flavor compound (SFC). **b** Photograph of cultured meat fabricated using the scaffold with SFC (CM+SFC). Scale bar: 8 mm. **c** Illustration explaining the mechanism of SFC system depending on temperature. **d** Classification of the flavor compounds analyzed in this study.

Scaffold type	Temperature	Flavor description	Detected flavor compound (IUPAC name)
Gel-SFC	25 °C	Meaty, Savory	Not detected
		Almond, Roasted bread	Not detected
		Floral, Cheese, Fat	Not detected
		Fishy, Pungent, Sour	Acetone
	150 °C	Meaty, Savory	Not detected
		Almond, Roasted bread	Benzaldehyde, Benzyl alcohol (Phenylmethanol)
		Floral, Cheese, Fat	Nonanoic acid, Octanoic acid
		Fishy, Pungent, Sour	Hexanoic acid
Gel+FM	25 °C	Meaty, Savory	Furfuryl mercaptan (furan-2-ylmethanethiol), Furfural (furan-2-carbaldehyde)
		Almond, Roasted bread	Not detected
		Floral, Cheese, Fat	Not detected
		Fishy, Pungent, Sour	Not detected
	150 °C	Meaty, Savory	Furfuryl mercaptan (furan-2-ylmethanethiol), Furfural (furan-2-carbaldehyde), 2,2-(dithiodimethylene)difuran
		Almond, Roasted bread	Not detected
		Floral, Cheese, Fat	Nonanoic acid
		Fishy, Pungent, Sour	Not detected

Gel+SFC	25 °C	Meaty, Savory	Not detected
		Almond, Roasted bread	Not detected
		Floral, Cheese, Fat	Not detected
		Fishy, Pungent, Sour	Not detected
	150 °C	Meaty, Savory	Furfuryl mercaptan (furan-2-ylmethanethiol), Furfural (furan-2-carbaldehyde), 2,2-(dithiodimethylene)difuran
		Almond, Roasted bread	Benzaldehyde, Benzyl alcohol (Phenylmethanol)
		Floral, Cheese, Fat	Nonanoic acid, Heptanoic acid, Octanoic acid
		Fishy, Pungent, Sour	Not detected

Table R2 (Supplementary Table 1): Flavor compounds detected from the hydrogel without switchable flavor compound (Gel-SFC), hydrogel with SFC (Gel+SFC), and hydrogel mixed with pure furfuryl mercaptan (Gel+FM) depending on the temperature.

Cultured meat type	Flavor description	Detected flavor compound
CM-SFC	Meaty, savory	Methanethiol, 2-methylthiophene
	Floral, Creamy	Not detected
	Fruity	Carbon disulfide, n-butanol, Methyl but-2-enoate

	Fish, Pungent	Trimethylamine
CM+SFC	Meaty, savory	Methanethiol
	Floral, Creamy	Butane-2,3-dione, Pent-1-en-3-ol
	Fruity	Carbon disulfide
	Fish, Pungent	Not detected
CM+SFCV	Meaty, savory	Methanethiol, 2-methylthiophene
	Floral, Creamy	Pentanoic acid
	Fruity	Heptyl pentanoate
	Fish, Pungent	Not detected

Table R3 (Supplementary Table 2): Flavor compounds detected from the cultured meat without switchable flavor compound (CM-SFC), cultured meat with SFC (CM+SFC), and cultured meat with flavor varied SFC (CM+SFCV) by electronic nose.

Comment 14. Line 265: This line claims that cultured meat without the flavor enhancement strategy had undesirable olfactory characteristics. However, this is only validated for this system. The authors should clarify that this might not be true for other systems (e.g., other culture media, other scaffolds, etc.).

Response:

We appreciate the reviewer's comment. We agree with the reviewer. We regret that we made such a strong sentence. In the revised manuscript, we removed such sentence.

Comment 15. Line 292, 293: the chemical 3-mercapto-1-propanol is mentioned twice. Is this correct, or a typo?

Response:

We apologize for the incorrect information. We revised the paragraph as below:

Related changes in the manuscript:

(Line 425-435)

To induce the formation of disulfide bonds at the thiol groups of Maillard reaction products, 11.6 mmol of 3-mercapto-1-propanol (TCI-SEJIN CI, Korea) was reacted with equimolar amounts of furfuryl mercaptan (TCI-SEJIN CI, Korea), 3-mercapto-2-pentanone (TCI-SEJIN CI, Korea), or 2-methyl-3-furanthiol (TCI-SEJIN CI, Korea) for 48 h at 80 °C, along with 2.56 mmol of hydrogen peroxide. SFC was prepared *via* urethane reaction with the isocyanate trimer (hexamethylene diisocyanate isocyanurate trimer; BLD Pharm, China). The molar ratio between isocyanate trimer, 2-hydroxyethyl methacrylate (Sigma–Aldrich, #128635), and disulfide-linked Maillard reaction products was 1:2:1. The SFC based on furfuryl mercaptan was synthesized with a 30% (w/v) concentration of all reagents within the solvent (propylene carbonate; Sigma–Aldrich, #110264), whereas SFCs with 3-mercapto-2-pentanone or 2-methyl-3-furanthiol were prepared with 20% (w/v) concentration.

Comment 16. Line 320: Please describe how long / how many times the hydrogels were washed.

Response:

We sincerely apologize for the insufficient information. The hydrogels were washed with distilled water four times. First two times were performed by immersing the hydrogels with water for 1 minute for each wash. Then, the hydrogels were immersed in water for 6 hours at 37 °C to allow the hydrogels to swell. Then, the hydrogels were taken from the water and washed with new distilled water for one minute before lyophilization. We included this information in the revised method section.

Related changes in the manuscript:

(Line 468-472)

To remove any unreacted molecules, the hydrogels were washed with distilled water for four times. First two times were performed by immersing the hydrogels with water for 1 min for each wash. Then, the hydrogels were immersed in water for 6 hours at 37 °C to allow the hydrogels to swell. Then, the hydrogels were taken from the water and washed with new distilled water for 1 min.

Comment 17. Line 334: Please make sure that all experimental details are given, including injection speed, etc.

Response:

We sincerely apologize for the insufficient information. We now fully revised the method section to include all the details of the experimental processes in the revised manuscript as below:

Related changes in the manuscript:

(Line 494-514)

A heating plate was used for temperature control. Specifically, the heating plate was covered with aluminum foil and the temperature was adjusted. When the temperature reached to the setting point, the samples were put on the foil covered plate for 5 minutes. Using GC–MS (Agilent 8890 GC system–Agilent 5677B MSD, Agilent Technologies), volatile and semivolatile compounds were detected for each group^{1,2}. In particular, carboxen/polydimethylsiloxane/divinylbenzene (CAR/PDMS/DVB) fibers were used to adsorb the volatile compounds from the samples by headspace solid-phase microextraction (HS–SPME). At this time, the groups that were placed at room temperature were heated to 30 °C for 20 min, while the others were heated to 80 °C for 20 min using the autosampler (Agilent 7693, CombiPAL sampler 80, Agilent) to allow the volatile compounds to be adsorbed onto the fibers. Then, the samples were maintained at room temperature with the fibers for an additional 40 min to complete the adsorption of the compounds. Subsequently, the volatile compounds adsorbed onto the fibers were analyzed. Helium with 99.999% purity was used for the carrier gas with the split flow rate of 40 mL/min. Fibers were injected into the inlet at the temperature of 250 °C using the split (20:1) injection mode. The initial temperature of the column oven was 40 °C and hold time was 5 min. Then, the temperature increased to 240 °C at a rate of 4 °C/min, and the set temperature was hold for 20 min to analyze the volatile compounds. The GC column

used in this experiment was DB-WAX (Agilent 123-7063). The temperature of the column oven was kept at 40 °C for 5 min, and then increased to 240 °C at a rate of 4 °C/min. Finally, the column oven temperature was kept at 240 °C for 20 min to analyze the volatile and semivolatile compounds of each group.

[References]

1. ACS Applied Materials & Interfaces, 14, 33, 38235-38245 (2022).
2. Korean Society for Biotechnology and Bioengineering Journal, 33, 2 (2018).

Comment 18. Line 339: Please provide the parameters of freezing (e.g., temperature, time, etc.) before lyophilization.

Response:

We apologize for the insufficient information. We supplemented the information for the whole lyophilization process and conditions as below:

Related changes in the manuscript:

(Line 516-517)

Before culturing cells, the hydrogels were freezed at -20 °C for one day and freeze-dried for 4 days at -50 °C by lyophilizer to obtain the scaffolds.

Comment 19. Line 355: Please indicate how frequently cells were fed during proliferation.

Response:

We apologize for the insufficient information. Growth media was replaced every two days to feed the cells during proliferation. Also, differentiation media was replaced every two days during differentiation. We included this information in the revised method section as below:

Related changes in the manuscript:

(Line 527-528)

Growth medium was replaced every two days to feed the cells during proliferation.

(Line 531)

Differentiation medium was replaced every two days.

Comment 20. Line 367-380: Please provide catalog numbers for antibodies and reagents/kits used (please do this throughout the methods section). Additionally, please specify the concentration of MF20, since DSHB antibodies are provided with variable concentrations, and so a dilution factor does not exactly indicate the ug/mL of antibody used.

Response:

We apologize for the insufficient information. We added the catalog numbers for all the reagents and kits used in this study in the revised supplementary information. Also, we added the catalog numbers for the antibodies in the method section of the revised manuscript.

Related changes in the supplementary information:

Reagent name	Company	Catalog number
3-mercapto-1-propanol	TCI-SEJIN CI	M1206
Furfuryl mercaptan	TCI-SEJIN CI	F0077
3-mercapto-2-pentanone	TCI-SEJIN CI	M2026
2-methyl-3-furanthiol	TCI-SEJIN CI	M1847
Hydrogen peroxide	Sigma Aldrich	516813
Hexamethylene diisocyanate isocyanurate trimer	BLD Pharm	3779-63-2
2-hydroxyethyl methacrylate	Sigma Aldrich	128635
Propylene carbonate	Sigma Aldrich	110264
Dimethyl sulfoxide-d6	Sigma Aldrich	151874

Fish gelatin	GELTECH	-
Methacrylic anhydride	Sigma Aldrich	276685
2-hydroxy-4-(2-hydroxyethoxy)-2-methylpropiophenone	Sigma Aldrich	410896
High-glucose Dulbecco's modified Eagle medium (HG-DMEM)	Thermo Fisher Scientific	LB001-05
Heat-inactivated fetal bovine serum (FBS)	Welgene	S101-01
Penicillin-streptomycin-glutamine (PS)	Gibco® Life Technologies	10378016
Basic fibroblast growth factor (bFGF)	Peprotech	100-18B
Phosphate buffer saline (PBS)	Gibco® Life Technologies	10010031
Trypsin-EDTA	Welgene	LS015-10
Horse serum	Thermo Fischer Scientific	26050088
70% Ethanol	DAEJUNG	4018-4410
D-Plus™ CCK cell viability assay kit	Dongin LS	CCK-3000
Formalin solution, neutral buffered, 10%	Sigma Aldrich	HT5012
Bovine serum albumin	Sigma Aldrich	A3311
Triton™ X-100 solution	Sigma Aldrich	93443
Alexa Fluor 488™ phalloidin	Thermo Fisher Scientific	A12379

DAPI	Thermo Fisher Scientific	D9542
MF 20	DSHB	AB2147781
Donkey anti-mouse Alexa Fluor 594	Thermo Fisher Scientific	A21203
Bovine myosin-1 (MYH1) ELISA kit	MyBioSource	MBS7229767

Table R4 (Supplementary Table 3): List of the reagents used in the method section.

Product	Company	Catalog number
12-14 kDa membrane	Thermo Fisher	08667E
24-well plate	SPL	31024
Carboxen/polydimethylsiloxane/divinylbenzene (CAR/PDMS/DVB) fiber	Sigma Aldrich	57348-U
DB-WAX column	Agilent	123-7063
TPP® tissue culture dishes	Sigma Aldrich	93100
MXT-5 GC metal capillary column	Restek	70223
MXT-1701 GC metal capillary column	Restek	72023

Table R5 (Supplementary Table 4): List of the products used in the method section.

Related changes in the manuscript:

(Line 543-552)

F-actin filaments of myoblasts were stained using the staining solution which includes 0.165 μ M Alexa Fluor 488™ phalloidin (Thermo Fisher Scientific) and 0.08 mg/mL DAPI (Thermo Fisher Scientific, #D9542) in the solvent composed of 1X PBS and 1% BSA. For myosin heavy chain staining, MHC antibody was diluted 100-fold using the MF 20 concentrate solution (MF 20, DSHB, ID:AB2147781) in a solvent composed of 10% (v/v) horse serum and 2% (w/v) BSA. The MHC antibody solution was then added to the cells for 2 h at room temperature. Then, the samples were washed twice with 1X PBS and once with 0.025% Triton X-100. Subsequently, 0.005 mg/mL secondary antibodies (Donkey anti-mouse Alexa Fluor 594, Thermo Fischer, #A21203) in the same solvent as MF20 were added to the samples for 1 h at

room temperature, followed by washing.

Comment 21. Line 393: Please specify how much of the various samples were transferred to the 20 mL headspace vials.

Response:

We apologize for the insufficient information. All the samples including beef were prepared at same mass of 0.4 g before transferred to the 20 mL headspace vials. We included this information in the revised method section as below:

Related changes in the manuscript:

(Line 569-571)

Then, 0.4 g of CM-SFC, CM+SFC, CM+SFCV, and beef brisket were transferred into a 20 mL headspace vial, followed by heating at 150 °C for 5 min.

Major revisions:

Comment 22. The authors claim that no (or at least very few) studies have looked at flavor of cultured meat. However, the authors fail to reference key papers on this topic, including:

o DOI: 10.5851/kosfa.2021.e72

o DOI: 10.1021/acs.jafc.2c08004

o DOI: 10.1038/s41467-023-44359-9 (the authors own work which only just came out, so they may be planning to add it in anyways).

o DOI: 10.1016/j.foodres.2022.111636

As such, the authors overstate the paucity of research in this area. Instead, the authors should take more time discussing the existing literature on this topic and putting the current study in the context of that broader literature.

Response:

We sincerely appreciate the reviewer's valuable comment. We revised the introduction to fully discuss the references provided by the reviewer.

Related changes in the manuscript:**(Line 55-65)**

Recently, many papers reported to improve the flavor characteristics of cultured meat. For example, Joo et al. compared the taste of cultured muscle tissue obtained from chicken satellite cells to that of traditional chicken¹. However, they revealed that it is challenging to mimic the taste of traditional meat in solely cultured tissue without any treatment due to the different amino acid composition. Liu et al. also reported that cultured fat had fruity and creamy flavor characteristics due to the lipid oxidation of fatty acids formed in cultured fat². Recently, our group revealed that the enhanced proliferation and differentiation of murine and bovine myoblasts could induce relatively more meat-like flavors in cultured meat^{3,4}. Nevertheless, in order for cultured meat to present the flavor characteristics of well-cooked meat products, a

strategy to generate the Maillard reaction compounds during cooking is still needed.

[References]

1. Food science of animal resources, 42, 175 (2022).
2. Journal of Agricultural and Food Chemistry, 71, 4113-4122 (2023).
3. ACS Applied Materials & Interfaces, 14, 38235-38245 (2022).
4. Nature Communications, 15, 77 (2024).

Comment 23. In line 180 (and in figure 3), the authors point towards meaty flavors being “roasted bread, meaty” and “bitter almond, burnt sugar, malt” – However, these do not match exactly with the flavors in figure 2D. Similarly, the flavors described in figure 4b(iii) do not clearly track with the same compounds. The authors should make sure that the words used to categorize flavors are consistent and include chemical names, where possible, so that the connection between the figures and flavors is clear.

Response:

We appreciate the reviewer’s valuable comments. We revised the Figures by adding the flavor classification as well as re-analyzing the detected flavor compounds according to the classification. In the revised Fig. 1 (Fig. R5), flavor compounds detected in this study are classified according to their flavor notes. Then, the flavor notes are classified into two main flavor groups: Offensive flavor and pleasant flavor. The flavor analysis results of revised Fig. 2 (Fig. R6), Fig. 3 (Fig. R7), and Fig. 4 (Fig. R8) are organized into this flavor classification of Fig. 1 to help the readers to easily follow. Moreover, we provided the detailed information of the detected volatile compounds with their flavor notes in Supplementary Table 1 (Table R6) and Supplementary Table 2 (Table R7).

Fig. R5 (Fig. 1): Schematic illustration of switchable flavor system. a Illustrative description of scaffold structure conjugated with switchable flavor compound (SFC). **b** Photograph of cultured meat fabricated using the scaffold with SFC (CM+SFC). Scale bar: 8 mm. **c** Illustration explaining the mechanism of SFC system depending on temperature. **d** Classification of the flavor compounds analyzed in this study.

Fig. R6 (Fig. 2): Synthesis of SFC and thermal responsivity evaluation. **a** Chemical structure of the switchable flavor compound (SFC) involving two binding groups (R_1 - R_2) with methacrylate end and one flavor group (R_3) with a thermal responsive disulfide bridge. **b** Ultraviolet-visible (UV-Vis) spectra of SFC to monitor the mobility of furfuryl mercaptan upon heating of SFC. The peak at 335 nm indicates the mobility of the furan group of furfuryl mercaptan thermal-responsively generated from SFC. The intervals for heating are 0 min, 10 min, 1 h, 4 h, 7 h, 12 h, and 24 h. **c** Photographs of the hydrogel without SFC (Gel-SFC), hydrogel with SFC (Gel+SFC), and hydrogel mixed with pure furfuryl mercaptan (Gel+FM) with the illustrations of their network structure. Gel+SFC features the robust covalent bonds between the gelatin matrix and SFC. Meanwhile, Gel+FM exhibits the weak interaction

between gelatin matrix and furfuryl mercaptan. Scale bars: 0.4 cm **d** Flavor analysis of hydrogels before and after heating at Maillard temperature, 150 °C. All samples are pre-incubated in the distilled water for 15 days. Pie chart presents the ratio of the flavor compounds classified according to the specific flavor notes. Non-flavor compounds are excluded in the pie chart and each flavor note is represented by a different color ($n = 3$).

Fig. R7 (Fig. 3): Biological evaluation and flavor analysis of cell-cultured scaffold. a Illustration of the flavor enriching process of the switchable flavor compound (SFC) under cultured meat fabrication process. **b** Immunofluorescence images of proliferated myoblasts on each group. Scale bars: 100 μm. **c** Cell viability on day 1, day 5, and day 7 using CCK-8 assay kit (mean ± SD, $n = 3$ independent experiments, One-way ANOVA with Tukey method). Grey color indicates cultured meat without SFC (CM-SFC) and red color indicates cultured meat with SFC (CM+SFC). **d** Confocal images showing myosin heavy chain (MHC) and nuclei

immunostained with MF20 (red) and DAPI (blue), respectively. Scale bars: 100 μ m. **e** Quantitative assessment of MHC amount by bovine myosin-1 enzyme linked immunosorbent assay (ELISA) (mean \pm SD, $n = 3$ independent experiments, One-way ANOVA with Tukey method). MHC amount of the CM+SFC is normalized to that of CM-SFC. **f** Assessment of volatile compounds in CM-SFC and CM+SFC after heating at 150 $^{\circ}$ C ($n = 3$).

Fig. R8 (Fig. 4): Electronic nose analysis of cultured meat specimens. **a** Illustration of cultured meat specimens with different scaffolds: cultured meat without SFC (CM-SFC), cultured meat with SFC (CM+SFC), and cultured meat with flavor varied SFC (CM+SFCV).

b Ratio of the flavor compounds detected from CM-SFC, CM+SFC, and CM+SFCV (mean \pm SD, $n = 3$ independent experiments, One-way ANOVA with Tukey method). Pie chart is shown to present the specific flavor profiles of each group. Flavor notes are presented with different color. **c** Principal component analysis (PCA) of the flavor compounds of each group (discrimination index = 90, $n = 3$).

Scaffold type	Temperature	Flavor description	Detected flavor compound (IUPAC name)
Gel-SFC	25 °C	Meaty, Savory	Not detected
		Almond, Roasted bread	Not detected
		Floral, Cheese, Fat	Not detected
		Fishy, Pungent, Sour	Acetone
	150 °C	Meaty, Savory	Not detected
		Almond, Roasted bread	Benzaldehyde, Benzyl alcohol (Phenylmethanol)
		Floral, Cheese, Fat	Nonanoic acid, Octanoic acid
		Fishy, Pungent, Sour	Hexanoic acid
Gel+FM	25 °C	Meaty, Savory	Furfuryl mercaptan (furan-2-ylmethanethiol), Furfural (furan-2-carbaldehyde)
		Almond, Roasted bread	Not detected
		Floral, Cheese, Fat	Not detected
		Fishy, Pungent, Sour	Not detected
	150 °C	Meaty, Savory	Furfuryl mercaptan (furan-2-ylmethanethiol), Furfural (furan-

			2-carbaldehyde), 2,2-(dithiodimethylene)difuran
		Almond, Roasted bread	Not detected
		Floral, Cheese, Fat	Nonanoic acid
		Fishy, Pungent, Sour	Not detected
Gel+SFC	25 °C	Meaty, Savory	Not detected
		Almond, Roasted bread	Not detected
		Floral, Cheese, Fat	Not detected
		Fishy, Pungent, Sour	Not detected
	150 °C	Meaty, Savory	Furfuryl mercaptan (furan-2-ylmethanethiol), Furfural (furan-2-carbaldehyde), 2,2-(dithiodimethylene)difuran
		Almond, Roasted bread	Benzaldehyde, Benzyl alcohol (Phenylmethanol)
		Floral, Cheese, Fat	Nonanoic acid, Heptanoic acid, Octanoic acid
		Fishy, Pungent, Sour	Not detected

Table R6 (Supplementary Table 1): Flavor compounds detected from the hydrogel without switchable flavor compound (Gel-SFC), hydrogel with SFC (Gel+SFC), and hydrogel mixed with pure furfuryl mercaptan (Gel+FM) depending on the temperature.

Cultured meat type	Flavor description	Detected flavor compound
--------------------	--------------------------

CM-SFC	Meaty, savory	Methanethiol, 2-methylthiophene
	Floral, Creamy	Not detected
	Fruity	Carbon disulfide, n-butanol, Methyl but-2-enoate
	Fish, Pungent	Trimethylamine
CM+SFC	Meaty, savory	Methanethiol
	Floral, Creamy	Butane-2,3-dione, Pent-1-en-3-ol
	Fruity	Carbon disulfide
	Fish, Pungent	Not detected
CM+SFCV	Meaty, savory	Methanethiol, 2-methylthiophene
	Floral, Creamy	Pentanoic acid
	Fruity	Heptyl pentanoate
	Fish, Pungent	Not detected

Table R7 (Supplementary Table 2): Flavor compounds detected from the cultured meat

without switchable flavor compound (CM-SFC), cultured meat with SFC (CM+SFC), and cultured meat with flavor varied SFC (CM+SFCV) by electronic nose.

In the revised Fig. 2d (Fig. R6d), flavor compounds with almond, floral and fat-like flavor notes were detected from Gel-SFC whereas Gel+SFC showed flavor compounds with meaty, almond, floral and fatty flavor notes detected. In Fig 3f (Fig. R7f), floral and fatty flavor compounds (carboxylic acids) were not detected from both CM-SFC and CM+SFC. This difference may arise from the different experimental condition. In Fig. 2d, the flavor compound in the scaffolds were analyzed after immersing in distilled water for 15 days. Distilled water was replaced every two days. Because culture medium contains animal serum which is composed of various animal-derived proteins, culture medium itself can create flavor compounds when heated. The products from the Maillard reaction between animal serum and glucose in culture media might react with the flavor compounds derived from SFC, because the Maillard reaction produces various unstable and highly reactive volatile compounds¹. This reaction might converse the flavor compounds from SFC to non-flavor compounds or might create undesirable or desirable flavor compounds. In Fig. 2 and 3, it can be verified in both CM-SFC and CM+SFC that the number of detected flavor compounds are decreased after culturing cells with culture medium. This might be due to the effect of volatile compounds created from serum-containing media which remain in the cultured meat specimens. Therefore, distilled water was used to compare the flavor characteristics of Gel-SFC, Gel+SFC, and Gel+FM, excluding the effect of culture medium.

We included this in the revised discussion section as below:

Related changes in the manuscript:

(Line 407-421)

Also, serum-free media are being actively studied in the field of cultured meat. It is anticipated

that if non-animal derived nutrients capable of replacing serum in media are developed, it would be able to verify the flavor-enriching effects of the SFC system more precisely. Because serum-containing culture medium contains various animal-derived proteins and glucose, Maillard reaction may occur at 150 °C. The products from the Maillard reaction between animal serum and glucose in culture media might react with the flavor compounds derived from SFC, because the Maillard reaction produces various unstable and highly reactive volatile compounds¹. This reaction might converse the flavor compounds from SFC to non-flavor compounds or might create undesirable or desirable flavor compounds. In Fig. 2 and 3, it can be verified in both CM-SFC and CM+SFC that the number of detected flavor compounds are decreased after culturing cells. This might be due to the effect of volatile compounds created from serum-containing media which remain in the cultured meat specimens. If serum-free medium is applied to this study, unwanted reactions between the volatile compounds from serum-containing media and the compounds from SFC would be excluded, which would help analyzing the effect of SFC system on cultured meat's flavor characteristics more precisely.

Then, in Fig. 3 (Fig. R7) and Fig. 4 (Fig. R8), the flavor characteristics of specimens were compared after cell culturing with culture medium. In Fig. 4, compared to the results in Fig. 3f, different and more various flavor compounds were detected due to the different polarity of the used columns in the electronic nose performance. Non-polar and mild-polar volatile compounds were detected in Fig. 4 whereas polar volatile compounds were detected in Fig. 3. Regardless of the polarity of the used columns, it was confirmed that the flavor compounds with meaty and savory flavor notes were highly detected when the SFC was introduced into the scaffold. We included this information in the revised manuscript as below:

Related changes in the manuscript:

(Line 294-297)

Compared to the results in Fig. 3f, different and more various flavor compounds were detected due to the different polarity of the used columns in the electronic nose performance. Here, non-polar and mild-polar volatile compounds were detected whereas polar volatile compounds were detected in Fig. 3.

Also, it was confirmed that flavor compounds other than furfuryl mercaptan were detected from Gel+SFC and CM+SFC. At first, we expected only the flavor compounds included in the SFC, furfuryl mercaptan, to be released. However, more diverse flavor compounds with meaty and savory flavor notes were found at 150 °C (Table R6 and Table R7). These unexpected products would be due to the hydrogen peroxide, an oxidation reagent for forming the disulfide bond in the Maillard reaction product. The hydrogen peroxide would be residual in the CM+SFC with a concentration of 0.386 mM (which is within the food-grade concentration range). When the furfuryl mercaptan is generated upon cooking, the oxidative reactivity of hydrogen peroxide can induce the disulfide bond within the furfuryl mercaptan, forming the 2,2-(dithiodimethylene)difuran² (Fig. R9). Moreover, the hydrogen peroxide could generate the hydroxyl or hydroperoxyl radicals upon heating³. Hence, these radicals could induce the bond scission and thereby result in the diverse volatile compounds with meaty and savory flavor notes, such as 2-methylthiophene or furfuryl methyl disulfide.

Fig. R9. Illustrative description of the diverse flavor compounds formation derived from the reaction between hydrogen peroxide and furfuryl mercaptan.

We included this in the revised discussion section as below:

Related changes in the manuscript:

(Line 346-358)

At first, we expected only the flavor compounds included in the SFC, furfuryl mercaptan, to be released. However, more diverse flavor compounds with meaty and savory flavor notes were found at 150 °C. These unexpected products would be due to the hydrogen peroxide, an oxidation reagent for forming the disulfide bond in the Maillard reaction product. The hydrogen peroxide would be residual in the CM+SFC with a concentration of 0.386 mM (which is within the food-grade concentration range). When the furfuryl mercaptan is generated upon cooking, the oxidative reactivity of hydrogen peroxide can induce the disulfide bond within the furfuryl mercaptan, forming the 2,2-(dithiodimethylene)difuran². Moreover, the hydrogen peroxide could generate the hydroxyl or hydroperoxyl radicals upon heating³. Hence, these radicals could induce the bond scission and thereby result in the diverse volatile compounds with meaty and savory flavor notes, such as 2-methylthiophene or furfuryl methyl disulfide. Finally, we

fabricated the flavor-enriched cultured meat after carrying out cell proliferation and differentiation of bovine myoblasts on the SFC-introduced scaffold.

[References]

1. Trends in Food Science & Technology, 11, 9-10, 364-373 (2000).
2. Journal of the American Chemical Society (2024): Light in a Heartbeat: Bond Scission by a Single Photon above 800 nm.
3. Journal of endodontics, 30, 45-40 (2004).

Comment 24. Throughout, the use of “flavor” and “aroma” is inconsistent (e.g., line 232 vs line 234). For consistency, the authors should stick to one term. I would propose “flavor compound” be used throughout the manuscript. This is because the context of the paper is around the flavor of the meats, but “flavor” is a general sense caused by a complex mixture of flavor compounds, and so the use of “flavor” alone might be inaccurate. The use of “flavor compound” is both clear and accurate.

Response:

We sincerely appreciate the reviewer’s comment. We agree with the reviewer. To reduce the inconsistency, we revised throughout the manuscript to use only “flavor compound” as well as removing the word “aroma”. We appreciate the reviewer’s comment and suggestion.

Comment 25. In lines 194-195, the authors say that a lack of significant difference between the groups “indicates non-cytotoxicity” – however, this is inappropriate statistics for proving the null hypothesis. Rather, with the statistical tests used, the authors should only claim that “no significant difference in cell viability existed between CM-SFC and CM+SFC on days 1 and 5, though a significant reduction in viability was present on day 7 for CM+SFC.” Relatedly, it is inappropriate in line 197 to claim that the difference is negligible because of a high-ish p-value. Indeed, numerically the absorbance looks to be two-fold higher for SFM-SFC than SFM+SFC on day 7, which could be seen as a very non-negligible difference. The results clearly suggest (both absolutely and statistically) that SFM+SFC reduces cell growth over a seven day period. This is okay, and does not diminish the study – the authors should present the facts frankly and avoid couching their findings in caveats.

Response:

We apologize for the incorrect statement in Fig. 3. We appreciate the reviewer’s comment. The cell viability on day 7 was lower in CM+SFC compared to that of CM-SFC. However, the absorbance value did not decrease from day 1 to day 7 in CM+SFC. These results indicate that rather than cell death in CM+SFC, the proliferation rate was slow in CM+SFC, compared to that of CM-SFC. To discuss about this, we performed compression test and swelling test for both CM-SFC and CM+SFC. As a result, we found out that the swelling capacity of CM+SFC is higher than CM-SFC whereas the compressive modulus was slightly lower in CM+SFC (Fig. R10 and Fig. R11). Because compressive modulus of a scaffold affects long-term cell proliferation, the lower cell viability on day 7 in CM+SFC maybe due to the lower stiffness of the scaffold¹. For this difference in swelling capacity and compressive modulus, inhibition of GelMA crosslinking by SFC would be the reason. The radical-based covalent linkage between methacrylate of GelMA and the binding groups of SFC can decrease the crosslinking density of GelMA backbone which can eventually affect the swelling capacity and mechanical

properties of the scaffold. In the revised manuscript, we added this discussion.

Fig. R10 (Supplementary Fig. 6): Swelling degree of Gel-SFC and Gel+SFC measured after immersing each scaffold in culture media for 24 hours at 37 °C (mean ± SD, $n = 6$ independent experiments, One-way ANOVA with Tukey method). The swelling degree was calculated by the following equation.

Swelling degree (%)

$$= \frac{(\text{Weight of swollen scaffold} - \text{Weight of lyophilized scaffold}) * 100}{\text{Weight of lyophilized scaffold}}$$

Fig. R11 (Supplementary Fig. 7): Stiffness of Gel-SFC and Gel+SFC measured after immersing each scaffold in culture media for 24 hours at 37 °C (mean ± SD, $n = 3$ independent experiments, One-way ANOVA with Tukey method).

independent experiments, One-way ANOVA with Tukey method).

Related changes in the manuscript:

(Line 233-246)

The cell viability on day 7 was lower in CM+SFC compared to that of CM-SFC. However, the absorbance value did not decrease from day 1 to day 7 in CM+SFC. These results indicate that rather than cell death in CM+SFC, the proliferation rate was slow in CM+SFC, compared to that of CM-SFC. To further study this difference in cell viability between CM-SFC and CM+SFC, swelling degree and compressive strength of each group were compared (Supplementary Fig. 6 and Supplementary Fig. 7). The radical-based covalent linkage between methacrylate of GelMA and the binding groups of SFC can decrease the crosslinking density of GelMA backbone which can eventually affect the swelling capacity and mechanical properties of the scaffold. Because the compressive modulus of a scaffold affects the long-term cell proliferation, it was hypothesized that the lower cell viability of CM+SFC on day 7 is due to the lower compressive modulus of the scaffold¹. Therefore, the swelling degree and compressive modulus of each scaffold were evaluated. As expected, the swelling capacity of Gel+SFC was higher whereas the compressive modulus was slightly lower, compared to those of Gel-SFC.

[References]

1. RSC advances, 6, 3539-3551 (2016).

Comment 26. Line 198-199: the cell infiltration images are unconvincing, showing very limited penetration. It would be helpful if the authors could provide additional images to give a clearer picture of the infiltration, and if they could provide 3D images of the confocal images through the scaffolds.

Response:

We appreciate the reviewer's valuable comments. To obtain clearer picture of infiltration, we tried to take 3D confocal images, but we couldn't observe the inner structure of the samples due to the opacity (Fig. R12). We agree with the reviewer's comment pointing out that the confocal images of cross-section are unconvincing to show the cell infiltration. Also, we think that the cell infiltration through the scaffold may occur very limited, because we cultured cells in a static seeding method. It is known that orbital seeding with highly interconnected porous scaffold are critical for cells to be evenly distributed throughout the scaffold^{1,2}. Because we have not focused on strategies for 3D culture of cells, we believe that cell penetration into the scaffold would be very limited, as pointed out by the reviewer. However, if cells are 3D cultured and evenly distributed in Gel+SFC to increase cell content, we expect that the flavor of the cells and the flavor of SFC would exert a synergistic effect, making the flavor characteristics of cultured meat more similar to traditional meat.

Therefore, we removed the confocal images and SEM images which are insufficient to assert the cell infiltration. Rather, we supplemented the discussion section to discuss about the limitation and potential of this research in terms of 3D cell culture.

Fig. R12: Confocal images of Gel-SFC and Gel+SFC.

Related changes in the manuscript:

(Line 389-396)

Also, we only focused on the flavor characteristics depending on SFC in this paper. However, we expect the synergistic effect of SFC and cultured cells if three-dimensional culture strategies to increase cell content are further applied to this study. Through our previous research, we have reported that proteins synthesized by cells participate in the Maillard reaction and affect the flavor characteristics of cultured meat^{3,4}. Therefore, if a strategy to increase the content of the cell-derived proteins is converged into the SFC system reported in this study, it is expected that cultured meat with the flavor of traditional meat will be possible.

[References]

1. Tissue Engineering Part C: Methods, 14, 4 (2008)

2. Journal of orthopaedic research, 38, 6 (2020)
3. Nature Communications, 15, 77 (2024)
4. Matter (2024): Rice grains integrated with animal cells: A shortcut to a sustainable food system.

Comment 27. The timing of various heating experiments is a bit confusing. For instance, in Fig. 2b and supplementary figure 3, >1 hour is needed to see detectable compound release; however, later cooking experiments only heat for 1 hour (e.g., line 246). How is it possible that these shorter cooking times are releasing sufficient flavor compounds for detection? The authors should clarify this process, the choice of a 5 minute cooking time, and how this is mechanistically possible, otherwise it is hard to follow and seems implausible that 5 minutes would yield detectable flavor compound release.

Response:

The authors sincerely appreciate your great comment, which helped us avoid a critical misunderstanding. We believe that this concern emerged because we failed to describe the experimental details of how collecting the released flavor compounds.

Fig. R13: Experimental protocols of a UV-VIS and b GC-MS evaluation.

Fig. R13 depicts the methods of ultraviolet-visible (UV-VIS) experiments requiring more extended time for detection and gas chromatography–mass spectrometry (GC-MS) experiments showing shorter time. When it comes to UV-VIS experiments, the released flavor compounds were diluted in the solvent (Fig. R13a). Accordingly, a relatively prolonged heating time was required to concentrate the sufficient flavor compounds. Meanwhile, in the case of GC-MS experiments, the release flavor compounds were directly adsorbed to the fiber (Fig.

R13b). Hence, a shorter heating time was sufficient for clear detection. Reflecting on your opinion, we thoroughly detailed the related protocols in the revised manuscript.

Related changes in the manuscript:

(Line 474-480)

The UV quartz cuvette (chamber volume 3.5 mL, path-length 10 mm) was filled with 2 mL of 1.86 μ M SFC. SFC was heated at 37 $^{\circ}$ C, 80 $^{\circ}$ C, and 150 $^{\circ}$ C for 24 h by maintaining the closed system after sealing with the polytetrafluoroethylene stopper. The UV-VIS spectra within 200-600 nm were obtained for the desired heating time points with the settings of bandwidth 1.0 nm, scan speed 240 nm/min, and data interval 1.0 nm. The peak at 335 nm was assigned to the furan-related absorbance. Here, three replication experiments ($n = 3$) were performed for each heating temperature for reliability.

Comment 28. Line 247: Why is PCA used as the analytical method, here? The first principle component accounts for 96% of variability, which suggests that this is a simpler cause-and-effect association than would require PCA. Relatedly, the PCA analysis reveals dramatic differences between cultivated and slaughtered meat. This should be noted after line 250, which says that CM+SFCV had the most similar pattern to beef. Specifically, the authors should note that while this was the most similar, it was still significantly different from beef, suggesting that much work is still required to get these flavor profiles closer to that of slaughtered meat. Line 283, too, should be amended to say that the flavor pattern of cultured meat became “more similar” to that of beef, rather than just “similar.” Indeed, the PCA data suggests that the flavor pattern of cultured meat is still mostly different from conventional beef.

Response:

We appreciate the reviewer’s valuable comments. As the reviewer pointed out, the PCA results to compare the flavor pattern of cultured meat specimens and traditional beef still show the big difference between the CM+SFCV and beef in flavor characteristics. Therefore, we relocated the results of beef to the supplementary figures. Rather, we revised Fig. 4 (Fig. R14) to focus of comparing the flavor pattern of CM-SFC, CM+SFC, and CM+SFCV to show the potential of the SFC system to be diversified to admit more various flavor compounds. Also, we added Fig. 4b (Fig. R14b) to simply compare the ratio of pleasant flavor compounds and offensive flavor compounds between the cultured meat specimens. Then, we compare the specific flavor notes between the specimens to show what kind of pleasant flavors are expressed from each group. As a result, it was verified that the ratio of fruity flavor compounds is decreased whereas the ratio of meaty and floral, creamy flavor compounds are increased by introducing SFC and SFCV. The PCA of CM-SFC, CM+SFC, and CM+SFCV in Fig. 4c was performed to analyze the extent of changes in flavor characteristics. The results confirmed that the introduction of SFC and SFCV led to differences in flavor patterns. Also, we compare the

flavor notes between traditional beef and the cultured meat groups (CM-SFC, CM+SFC, and CM+SFCV) in Supplementary Fig. 9 and Supplementary Fig. 10 and confirmed that CM+SFCV has the most similar flavor characteristics of traditional beef among the three cultured meat groups. However, the PCA results in Supplementary Fig. 10 demonstrate that even with diversification of flavor compounds, significant differences still exist in the flavor characteristics between cultured meat and traditional meat. Therefore, we mentioned these limitations in the discussion and further discussed about the future directions to address them. The future direction would be (1) development of SFC with more diverse flavor compounds than SFCV, (2) convergence of the SFC system with the strategy to increase the cell content in the scaffold to take the advantages of cell-derived flavors. As the traditional meat creates dozens of Maillard reaction compounds as flavor compounds when cooked, three kinds of flavor molecules in SFCV would still be insufficient to mimic the flavor of traditional meat perfectly. Therefore, developing the SFC system to admit dozens of Maillard compounds would be one of the future directions. Also, we have reported that proteins synthesized by cells participate in the Maillard reaction which affect the flavor characteristics of cultured meat in our previous research¹. Therefore, if a strategy to increase the content of cell-derived proteins is applied to the SFC system, the synergistic effect of SFC and cultured cells would result the cultured meat with the flavor of traditional meat. We supplemented this discussion the revised discussion section as below.

Fig. R14 (Fig. 4): Electronic nose analysis of cultured meat specimens. **a** Illustration of cultured meat specimens with different scaffolds: cultured meat without SFC (CM-SFC), cultured meat with SFC (CM+SFC), and cultured meat with flavor variated SFC (CM+SFCV). **b** Ratio of the flavor compounds detected from CM-SFC, CM+SFC, and CM+SFCV (mean \pm SD, $n = 3$ independent experiments, One-way ANOVA with Tukey method). Pie chart is shown to present the specific flavor profiles of each group. Flavor notes are presented with different color. **c** Principal component analysis (PCA) of the flavor compounds of each group (discrimination index = 90, $n = 3$).

Fig. R15 (Supplementary Fig. 9): Flavor profiles of beef, cultured meat without switchable flavor compound (CM-SFC), cultured meat with SFC (CM+SFC), and cultured meat with flavor varied SFC (CM+SFCV).

Fig. R16 (Supplementary Fig. 10): Principal component analysis (PCA) of the flavor compounds detected from beef, cultured meat without switchable flavor compound (CM-SFC), cultured meat with SFC (CM+SFC), and cultured meat flavor varied SFC (CM+SFCV). (Discrimination index = 82, $n = 3$)

Related changes in the manuscript:

(Line 378-396)

Here, we developed the SFC system to introduce meaty flavor compounds in cultured meat. Also, we further develop the strategy to diversify the flavor compounds of SFC to mimic the flavor properties of traditional meat in Fig. 4. However, there is still a significant difference between the flavor characteristics of cultured meat with SFC and that of traditional meat even with diversification of flavor compounds in SFC. To overcome this limitation, we think two future strategies would be possible: (1) development of SFC with more diverse flavor compounds than SFCV, (2) convergence of the SFC system with the strategy to increase the cell content in the scaffold to take the advantages of cell-derived flavors. As the traditional meat creates dozens of Maillard reaction compounds as flavor compounds when cooked, three kinds of flavor molecules in SFCV would still be insufficient to mimic the flavor of traditional meat perfectly. Therefore, developing the SFC system to admit dozens of Maillard compounds would be one of the future directions. Also, we only focused on the flavor characteristics depending on SFC in this paper. However, we expect the synergistic effect of SFC and cultured cells if three-dimensional culture strategies to increase cell content are further applied to this study. Through our previous research, we have reported that proteins synthesized by cells participate in the Maillard reaction and affect the flavor characteristics of cultured meat^{1,2}. Therefore, if a strategy to increase the content of the cell-derived proteins is converged into the SFC system reported in this study, it is expected that cultured meat with the flavor of traditional meat will be possible.

[References]

1. Nature Communications, 15, 77 (2024).
2. Matter (2024): Rice grains integrated with animal cells: A shortcut to a sustainable

food system.

Comment 29. Conclusion: The manuscript is lacking in discussion of the study's limitations.

The authors should address the following questions:

(1) The cultured meat is still quite different from conventional meat. Why might this be the case, and how can further engineering get it closer to beef?

(2) What about the use of gelatin? That is an animal-derived component, and so might not be appropriate for cultured meat. The authors should discuss opportunities for transitioning this technology to animal-free systems.

(3) Similarly, the study was performed with serum containing media, which could itself provide flavor compounds to the system. The authors should discuss how/if serum-free culture might affect the system.

(4) The authors do not discuss the safety of the chemistries used from a food safety standpoint. Are any toxic chemicals used? How can we ensure that no residual chemicals remain? Are toxic byproducts potentially produced? If so, how can the technology be transitioned to food-safe chemistry? Or, alternatively, is the chemistry already food-safe?

Response to the question (1):

This question is similar to the reviewer's comment 28, which points out the difference between the flavor of cultured meats and the flavor of traditional beef in Fig. 4. Therefore, we have similar response to this question as the response to the reviewer's comment 28. We agree with the reviewer's comment. The PCA results still show the significant difference between the cultured meat groups and traditional beef in flavor properties. The reason for this could be the limited variety of flavor compounds even in the SFCV introduced scaffold. In SFC and SFCV, we introduced single or triple flavor compounds, respectively. Although various flavor compounds were formed due to the dynamic disulfide exchange reaction in the SFC system, the types of flavor compounds are still limited compared to the dozens of Maillard reaction

products formed from traditional meat. Therefore, to overcome this limitation, we think two future strategies would be possible: (1) development of SFC with more diverse flavor compounds than SFCV, (2) convergence of the SFC system with the strategy to increase the cell content in the scaffold to take the advantages of cell-derived flavors. As the traditional meat creates dozens of Maillard reaction compounds as flavor compounds when cooked, three kinds of flavor molecules in SFCV would still be insufficient to mimic the flavor of traditional meat perfectly. Therefore, developing the SFC system to admit dozens of Maillard compounds would be one of the future directions. Also, we only focused on the flavor characteristics depending on SFC in this paper. However, we expect the synergistic effect of SFC and cultured cells if three-dimensional culture strategies to increase cell content are further applied to this study. Through our previous research, we have reported that proteins synthesized by cells participate in the Maillard reaction and affect the flavor characteristics of cultured meat^{1,2}. Therefore, if a strategy to increase the content of the cell-derived proteins is converged into the SFC system reported in this study, it is expected that cultured meat with the flavor of traditional meat will be possible.

We supplemented this perspective in the revised discussion section.

Related changes in the manuscript:

(Line 378-396)

Here, we developed the SFC system to introduce meaty flavor compounds in cultured meat. Also, we further develop the strategy to diversify the flavor compounds of SFC to mimic the flavor properties of traditional meat in Fig. 4. However, there is still a significant difference between the flavor characteristics of cultured meat with SFC and that of traditional meat even with diversification of flavor compounds in SFC. To overcome this limitation, we think two future strategies would be possible: (1) development of SFC with more diverse flavor compounds than SFCV, (2) convergence of the SFC system with the strategy to increase the

cell content in the scaffold to take the advantages of cell-derived flavors. As the traditional meat creates dozens of Maillard reaction compounds as flavor compounds when cooked, three kinds of flavor molecules in SFCV would still be insufficient to mimic the flavor of traditional meat perfectly. Therefore, developing the SFC system to admit dozens of Maillard compounds would be one of the future directions. Also, we only focused on the flavor characteristics depending on SFC in this paper. However, we expect the synergistic effect of SFC and cultured cells if three-dimensional culture strategies to increase cell content are further applied to this study. Through our previous research, we have reported that proteins synthesized by cells participate in the Maillard reaction and affect the flavor characteristics of cultured meat^{1,2}. Therefore, if a strategy to increase the content of the cell-derived proteins is converged into the SFC system reported in this study, it is expected that cultured meat with the flavor of traditional meat will be possible.

[References]

1. Nature Communications, 15, 77 (2024)
2. Matter (2024): Rice grains integrated with animal cells: A shortcut to a sustainable food system.

Response to the question (2):

We appreciate the reviewer's comments. As the reviewer pointed out, use of animal-derived compounds like gelatin and animal-derived serum might not be appropriate in cultured meat. Though animal-derived proteins such as gelatin and collagen have been used in cultured meat research due to their RGD sequences, there are some studies reporting the cultured meat scaffolds composed of plant-derived proteins. In this study, SFC was conjugated to the GelMA backbone by the reaction between the methacryloyl and the binding group of SFC. The introduction of methacryloyl in gelatin is possible by the substitution of methacrylate groups

on the amine-containing side groups of gelatin. Since proteins have amine groups, methacryloyl can also be introduced to plant-derived proteins. Therefore, we expect that SFC system can be possible to various plant-derived proteins. Although we confirmed the SFC system in GelMA scaffold, we expect this system can further be applied to various plant protein-based scaffolds.

We included this in the revised discussion section as below:

Related changes in the manuscript:

(Line 397-407)

This study focuses on material engineering for controlling the flavor characteristics of cultured meat. Therefore, the use of animal-derived components which is an issue in the cultured meat field is overlooked in this research. However, considering that the cultured meat industry ultimately aims for sustainable meat development, it is essential to discuss the feasibility of our SFC system even without animal-derived materials such as gelatin and serum-containing media. In this study, SFC is conjugated to the GelMA backbone by the reaction between the methacryloyl and the binding group of SFC. The introduction of methacryloyl in gelatin is possible by the substitution of methacrylate groups on the amine-containing side groups of gelatin. Since proteins have amine groups, methacryloyl can also be introduced to plant-derived proteins. Therefore, we expect that SFC system can be possible to various plant-derived proteins.

Response to the question (3):

As the reviewer pointed out, serum-containing media itself can produce flavor compounds when heated, since it contains animal-derived proteins. Although we washed the cultured meat specimens repeatedly, culture media may remain the specimens because the specimens were immersed in the media for long-term. Because serum-containing culture medium contains

various animal-derived proteins and glucose, Maillard reaction may occur at 150 °C. The products from the Maillard reaction between animal serum and glucose in culture media might react with the flavor compounds derived from SFC, because the Maillard reaction produces various unstable and highly reactive volatile compounds¹. This reaction might convert the flavor compounds from SFC to non-flavor compounds or might create undesirable or desirable flavor compounds. In Fig. 2 and 3, it can be verified in both CM-SFC and CM+SFC that the number of detected flavor compounds are decreased after culturing cells. This might be due to the effect of volatile compounds created from serum-containing media which remain in the cultured meat specimens. However, we confirmed that our SFC system can control the flavor profile of the cell scaffold upon cooking in serum-free condition (DW condition) in Fig. 2d (Fig. R17d). Therefore, we expect that if serum-free medium is applied to this study, unwanted reactions between the volatile compounds from serum-containing media and the compounds from SFC would be excluded, which would help analyzing the effect of SFC system on cultured meat's flavor characteristics more precisely. We believe that our strategy is feasible with the serum-free culture condition.

We included this in the revised discussion section as below:

Related changes in the manuscript:

(Line 407-421)

Also, serum-free media are being actively studied in the field of cultured meat. It is anticipated that if non-animal derived nutrients capable of replacing serum in media are developed, it would be able to verify the flavor-enriching effects of the SFC system more precisely. Because serum-containing culture medium contains various animal-derived proteins and glucose, Maillard reaction may occur at 150 °C. The products from the Maillard reaction between animal serum and glucose in culture media might react with the flavor compounds derived from SFC, because the Maillard reaction produces various unstable and highly reactive volatile

compounds¹. This reaction might converse the flavor compounds from SFC to non-flavor compounds or might create undesirable or desirable flavor compounds. In Fig. 2 and 3, it can be verified in both CM-SFC and CM+SFC that the number of detected flavor compounds are decreased after culturing cells. This might be due to the effect of volatile compounds created from serum-containing media which remain in the cultured meat specimens. If serum-free medium is applied to this study, unwanted reactions between the volatile compounds from serum-containing media and the compounds from SFC would be excluded, which would help analyzing the effect of SFC system on cultured meat's flavor characteristics more precisely.

[References]

1. Trends in Food Science & Technology, 11, 9-10, 364-373 (2000)

Fig. R17 (Fig. 2): Synthesis of SFC and thermal responsivity evaluation. **a** Chemical structure of the switchable flavor compound (SFC) involving two binding groups (R_1 - R_2) with methacrylate end and one flavor group (R_3) with a thermal responsive disulfide bridge. **b** Ultraviolet-visible (UV-Vis) spectra of SFC to monitor the mobility of furfuryl mercaptan upon heating of SFC. The peak at 335 nm indicates the mobility of the furan group of furfuryl mercaptan thermal-responsively generated from SFC. The intervals for heating are 0 min, 10 min, 1 h, 4 h, 7 h, 12 h, and 24 h. **c** Photographs of the hydrogel without SFC (Gel-SFC), hydrogel with SFC (Gel+SFC), and hydrogel mixed with pure furfuryl mercaptan (Gel+FM) with the illustrations of their network structure. Gel+SFC features the robust covalent bonds between the gelatin matrix and SFC. Meanwhile, Gel+FM exhibits the weak interaction between gelatin matrix and furfuryl mercaptan. Scale bars: 0.4 cm **d** Flavor analysis of hydrogels before and after heating at Maillard temperature, 150 °C. All samples are pre-incubated in the distilled water for 15 days. Pie chart presents the ratio of the flavor compounds classified according to the specific flavor notes. Non-flavor compounds are excluded in the pie chart and each flavor note is represented by a different color ($n = 3$).

Response to the question (4):

The authors sincerely appreciate your meaningful comments concerning whether the proposed flavoring strategy could be regarded as food-safe chemistry. We also acknowledge that the original limitedly suggested its limitation. Reflecting on your opinion, we clearly analyzed the current stage limitation of using food-grade ingredients in the discussion section. As suggested in Fig. 3, the primary bovine myoblasts cultured on the GelMA scaffold with SFC did not yield any cytotoxic responses after 7 days of proliferation and an additional 8 days of differentiation. Given these results, we would like to say that the SFC and its byproduct chemicals residual during long-term cell culture were not toxic for cells.

After contemplating your considerate comment, we carefully trimmed the manuscript to avoid the misunderstanding between biocompatibility and food safety. Namely, it should be cautious to regard the used reagents as food-grade, although they are proven to be biocompatible. For instance, the methacrylic anhydride is generally employed to functionalize the methacrylate in the Food and Drug Administration (FDA)-approved proteins (such as gelatin¹⁻³, silk fibroin⁴) and to prepare biocompatible scaffolds via photo-crosslinking. However, until recently, no sub-type of GelMA was granted by the FDA for food purposes.

Furthermore, we described an idea to realize the proposed SFC-based flavoring strategy using food-grade reagents, aiming to enlighten the future potential of this work to translate to real-world food-safe chemistry to the readers. In detail, the enzymatic gelatin crosslink using FDA-approved transglutaminase could be a promising approach to realize the robust binding of Maillard reaction products using only food-grade chemicals⁵. Moreover, the thermo-responsivity could be fulfilled by the natural disulfide chemistry of cysteine amino acids⁶.

Again, we sincerely appreciate your excellent comments, which enabled us to improve the manuscript quality considerably.

Related changes in the manuscript:

(Line 362-377)

However, irrelevant to these remarkable advances, this study still encounters a critical limitation. It should be cautious to regard the used reagents as food-grade, although they are proven to be biocompatible. For instance, the methacrylic anhydride is generally employed to functionalize the methacrylate in the Food and Drug Administration (FDA)-approved proteins (such as gelatin¹⁻³, silk fibroin⁴) and to prepare biocompatible scaffolds via photo-crosslinking. However, until recently, no sub-type of GelMA was granted by the FDA for food purposes. Namely, although the biocompatibility of CM+SFC was identified by the primary bovine myoblast experiments, it is still distant from establishing its food safety. Nevertheless, the

proposed flavoring strategy yields the potential to be implemented solely by food-grade chemicals. In particular, the enzymatic gelatin crosslink using FDA-approved transglutaminase could be a promising approach to realize the robust binding of Maillard reaction products using only food-grade chemicals⁵. Moreover, the thermo-responsivity could be fulfilled by the natural disulfide chemistry of cysteine amino acids⁶. In conclusion, beyond the current stage limitation related to food-grade chemistry, we anticipate that the interesting flavoring strategy developed in this study will contribute to the production of cultured meat that reaches the organoleptic properties of conventional meat.

[References]

1. *Engineered Regeneration*, 2, 47-26 (2021)
2. *Biomaterials*, 73, 254-271 (2015)
3. *Biomaterials Research*, 27, 86 (2023)
4. *Nat. Commun.*, 9, 1620 (2018)
5. *Nat. Commun.*, 15, 77 (2024)
6. *Biomacromolecules*, 23, 926-936 (2022)

Comment 30. Similarly, the manuscript is lacking in discussion of future research directions. Please include these in the discussion.

- Line 343: Please provide details for the myoblast isolation procedure, as well as data validation for the identity of these cells (e.g., staining for key satellite cell / myoblast markers such as Pax7 and MyoD)

- Methods: The methods are missing several experiments. Please provide methods for:

- o The stability of non-bound compounds (fig. 2)

- o NMR studies

- o Raman studies

- o Residual weight studies

- o SEM

- o Tissue sectioning / penetration studies

- o Statistical methods

As well as any other experimental details that are missing. All experiments must be adequately described.

Response:

We appreciate the reviewer's comments. We fully revised the discussion section to include the limitations and the future directions of this research as below:

Related changes in the manuscript:

(Line 362-421)

However, irrelevant to these remarkable advances, this study still encounters a critical limitation. It should be cautious to regard the used reagents as food-grade, although they are proven to be biocompatible. For instance, the methacrylic anhydride is generally employed to functionalize the methacrylate in the Food and Drug Administration (FDA)-approved proteins

(such as gelatin¹⁻³, silk fibroin⁴) and to prepare biocompatible scaffolds via photo-crosslinking. However, until recently, no sub-type of GelMA was granted by the FDA for food purposes. Namely, although the biocompatibility of CM+SFC was identified by the primary bovine myoblast experiments, it is still distant from establishing its food safety. Nevertheless, the proposed flavoring strategy yields the potential to be implemented solely by food-grade chemicals. In particular, the enzymatic gelatin crosslink using FDA-approved transglutaminase could be a promising approach to realize the robust binding of Maillard reaction products using only food-grade chemicals⁵. Moreover, the thermo-responsivity could be fulfilled by the natural disulfide chemistry of cysteine amino acids⁶. In conclusion, beyond the current stage limitation related to food-grade chemistry, we anticipate that the interesting flavoring strategy developed in this study will contribute to the production of cultured meat that reaches the organoleptic properties of conventional meat.

Here, we developed the SFC system to introduce meaty flavor compounds in cultured meat. Also, we further develop the strategy to diversify the flavor compounds of SFC to mimic the flavor properties of traditional meat in Fig. 4. However, there is still a significant difference between the flavor characteristics of cultured meat with SFC and that of traditional meat even with diversification of flavor compounds in SFC. To overcome this limitation, we think two future strategies would be possible: (1) development of SFC with more diverse flavor compounds than SFCV, (2) convergence of the SFC system with the strategy to increase the cell content in the scaffold to take the advantages of cell-derived flavors. As the traditional meat creates dozens of Maillard reaction compounds as flavor compounds when cooked, three kinds of flavor molecules in SFCV would still be insufficient to mimic the flavor of traditional meat perfectly. Therefore, developing the SFC system to admit dozens of Maillard compounds would be one of the future directions. Also, we only focused on the flavor characteristics depending on SFC in this paper. However, we expect the synergistic effect of SFC and cultured

cells if three-dimensional culture strategies to increase cell content are further applied to this study. Through our previous research, we have reported that proteins synthesized by cells participate in the Maillard reaction and affect the flavor characteristics of cultured meat^{7,8}. Therefore, if a strategy to increase the content of the cell-derived proteins is converged into the SFC system reported in this study, it is expected that cultured meat with the flavor of traditional meat will be possible.

This study focuses on material engineering for controlling the flavor characteristics of cultured meat. Therefore, the use of animal-derived components which is an issue in the cultured meat field is overlooked in this research. However, considering that the cultured meat industry ultimately aims for sustainable meat development, it is essential to discuss the feasibility of our SFC system even without animal-derived materials such as gelatin and serum-containing media. In this study, SFC is conjugated to the GelMA backbone by the reaction between the methacryloyl and the binding group of SFC. The introduction of methacryloyl in gelatin is possible by the substitution of methacrylate groups on the amine-containing side groups of gelatin. Since proteins have amine groups, methacryloyl can also be introduced to plant-derived proteins. Therefore, we expect that SFC system can be possible to various plant-derived proteins. Also, serum-free media are being actively studied in the field of cultured meat. It is anticipated that if non-animal derived nutrients capable of replacing serum in media are developed, it would be able to verify the flavor-enriching effects of the SFC system more precisely. Because serum-containing culture medium contains various animal-derived proteins and glucose, Maillard reaction may occur at 150 °C. The products from the Maillard reaction between animal serum and glucose in culture media might react with the flavor compounds derived from SFC, because the Maillard reaction produces various unstable and highly reactive volatile compounds⁹. This reaction might converse the flavor compounds from SFC to non-flavor compounds or might create undesirable or desirable flavor compounds. In Fig. 2 and 3,

it can be verified in both CM-SFC and CM+SFC that the number of detected flavor compounds are decreased after culturing cells. This might be due to the effect of volatile compounds created from serum-containing media which remain in the cultured meat specimens. If serum-free medium is applied to this study, unwanted reactions between the volatile compounds from serum-containing media and the compounds from SFC would be excluded, which would help analyzing the effect of SFC system on cultured meat's flavor characteristics more precisely.

[References]

1. Engineered Regeneration 2, 47-56 (2021)
2. Biomaterials 73, 254-271 (2015)
3. Biomaterials Research 27, 86 (2023)
4. Nat. Commun., 9, 1620 (2018)
5. Nat. Commun., 15, 77 (2024)
6. Biomacromolecules, 23, 926-936 (2022)
7. Nature Communications 15, 77 (2024)
8. Matter (2024): Rice grains integrated with animal cells: A shortcut to a sustainable food system.
9. Trends in Food Science & Technology, 11, 9-10, 364-373, (2000)

For the myoblast isolation, the following process was performed. Firstly, the harvested muscle tissues were washed once in 70% ethanol and twice in 2% (v/v) antibiotic-antimycotic solution (AA; Welgene, LS203-01) diluted in Dulbecco's phosphate-buffered saline (DPBS; Welgene, LB001-02). Then, the muscle tissues were cut into small pieces, and digested using 0.2% (w/v) collagenase type II (Worthington Biochemical Corporation, LS004174) dissolved in high glucose-Dulbecco's Modified Eagle Medium (HG-DMEM; Welgene, LB001-05)

supplemented with 1% (w/v) pronase (Calbiochem, 53702). Subsequently, the completely digested skeletal muscle tissues were re-suspended in HG-DMEM supplemented with 2% (v/v) heat-inactivated fetal bovine serum (FBS; Welgene, S101-01) for dissociation enzyme inactivation. After centrifuging the suspension at 1500 x g for 4 minutes, the pellets were re-suspended in red blood cell (RBC) lysis buffer (Sigma-Aldrich, 11814389001) for 10 minutes at room temperature to eliminate RBCs. Then, the RBC-free primary cells were filtered using a 100-um cell strainer (SPL, Korea), followed by filtration through a 70-um cell strainer (SPL). Finally, the muscle-derived primary cells were centrifuged at 1500 x g for 4 minutes, then re-suspended in the myoblast proliferation medium which is composed of HG-DMEM supplemented with 10% (v/v) FBS, 5 ng/mL basic fibroblast growth factor (bFGF; Peprotech, 100-18B), and 1% (v/v) AA. To isolate myoblasts from the muscle-derived primary cells, muscle-derived primary cells (5×10^5) were seeded onto a 35-mm culture dish (SPL) and cultured in the myoblast proliferation medium. After 24 hours, non-adherent cells to the culture dishes were removed and the remaining adherent cells in the fresh myoblast proliferation medium were cultured at 37 °C in a humidified atmosphere of 5% CO₂ in air with medium exchange at 2-day intervals. When cell confluency reached 50-60%, the adherent cells were immunostained with the myoblast marker, MyoD (Santa Cruz, sc-377460 AF488), for myoblast characterization.

Related changes in the supplementary information:

Primary bovine myoblasts were obtained from muscle tissues harvested from 29- to 31-month-old male or female Hanwoo cattle slaughtered at a local slaughterhouse (Kwell LPC, Hongcheon, Korea). All experimental procedures for animal slaughtering performed in this study complied with the Animal Care and Use Guidelines of Kangwon National University and were approved by the Institutional Animal Care and Use Committee (IACUC) of Kangwon National University (IACUC approval no. KW-220714-1).

For the myoblast isolation, the following process was performed. Firstly, the harvested muscle tissues were washed once in 70% ethanol and twice in 2% (v/v) antibiotic-antimycotic solution (AA; Welgene, LS203-01) diluted in Dulbecco's phosphate-buffered saline (DPBS; Welgene, LB001-02). Then, the muscle tissues were cut into small pieces, and digested using 0.2% (w/v) collagenase type II (Worthington Biochemical Corporation, LS004174) dissolved in high glucose-Dulbecco's Modified Eagle Medium (HG-DMEM; Welgene, LB001-05) supplemented with 1% (w/v) pronase (Calbiochem, 53702). Subsequently, the completely digested skeletal muscle tissues were re-suspended in HG-DMEM supplemented with 2% (v/v) heat-inactivated fetal bovine serum (FBS; Welgene, S101-01) for dissociation enzyme inactivation. After centrifuging the suspension at 1500 x g for 4 minutes, the pellets were re-suspended in red blood cell (RBC) lysis buffer (Sigma-Aldrich, 11814389001) for 10 minutes at room temperature to eliminate RBCs. Then, the RBC-free primary cells were filtered using a 100-um cell strainer (SPL, Korea), followed by filtration through a 70-um cell strainer (SPL). Finally, the muscle-derived primary cells were pelleted by centrifuging at 1500 x g for 4 minutes, then re-suspended in the myoblast proliferation medium which is composed of HG-DMEM supplemented with 10% (v/v) FBS, 5 ng/mL basic fibroblast growth factor (bFGF; Peprotech, 100-18B), and 1% (v/v) AA. To isolate myoblasts from the muscle-derived primary cells, muscle-derived primary cells (5×10^5) were seeded onto a 35-mm culture dish (SPL) and cultured in the myoblast proliferation medium. After 24 hours, non-adherent cells to the culture dishes were removed and the remaining adherent cells in the fresh myoblast proliferation medium were cultured at 37 °C in a humidified atmosphere of 5% CO₂ in air with medium exchange at 2-day intervals. When cell confluency reached 50-60%, the adherent cells were immunostained with the myoblast marker, MyoD (Santa Cruz, sc-377460 AF488), for myoblast characterization.

For NMR studies and Raman studies, we supplemented the method description as below:

Related changes in the manuscript:

(Line 438-444)

The chemical structure was investigated by the Raman spectroscopy (XploRA PLUS, HORIBA, France) and ¹H NMR (Avance III HD 300, Bruker Biospin, USA). For the Raman spectroscopy measurement, a 10x objective and a 532 nm laser (75 mW intensity) were employed. The peak at 520 cm⁻¹ was used as the calibration standard. The laser was irradiated for 30 s per cycle, and each analysis was repeated five times to acquire reliable spectra. For the NMR studies, the SFC was dispersed in the dimethyl sulfoxide-d₆ (Sigma–Aldrich) with the volume concentration of 10%.

Also, we supplemented the method of the stability of non-bound compounds (Fig. 2) and residual weight studies as below:

(Line 473-487)

Thermo-responsivity analysis of SFC

The UV quartz cuvette (chamber volume 3.5 mL, path-length 10 mm) was filled with 2 mL of 1.86 μM SFC. SFC was heated at 37 °C, 80 °C, and 150 °C for 24 h by maintaining the closed system after sealing with the polytetrafluoroethylene stopper. The UV-VIS spectra within 200-600 nm were obtained for the desired heating time points with the settings of bandwidth 1.0 nm, scan speed 240 nm/min, and data interval 1.0 nm. The peak at 335 nm was assigned to the furan-related absorbance. Here, three replication experiments (n = 3) were performed for each heating temperature for reliability.

To understand the long-term stability under open system, the vials (chamber volume 5 mL, 18 × 38 mm²) were filled with 1 mL of 0.37 mM SFC (n = 3) and 1 mL of pure furfuryl mercaptan (n = 3), respectively. The vials with SFC and furfuryl mercaptan was positioned at 37 °C up to

2 weeks. In particular, to emulate the open system during cell culture procedures, the vials were not closed with a cap, and the air exchange kept sufficient. The weight variation was measured for specific time points. ¹H NMR (Avance III HD 300, Bruker, Bruker Biospin, USA) spectra of residual SFC were obtained by diluting in the dimethyl sulfoxide-d₆ (10 vol%).

For the SEM images, both Gel-SFC and Gel+SFC were freeze-dried at -20 °C for one day and lyophilized for 4 days at -50 °C by lyophilizer. Then, the surface of the scaffolds were coated with Pt for 90 seconds before taking SEM images. However, this procedure description and the information of the SEM is excluded from the revised manuscript as the SEM images are now removed from the manuscript. Also, the confocal images of cross-sectioned cultured meat to show cell infiltration are now excluded in the revised manuscript. From the reviewer's comment 26, we concluded that the confocal images and SEM images of the cross-sectioned cultured meat and the scaffolds are unconvincing to show the cell infiltration throughout the samples. Therefore, we excluded the confocal and SEM images of cross-sectioned samples.

For the statistical methods, we supplemented in the experimental section as below:

(Line 585-588)

Statistics and reproducibility

The data are reported as mean ± standard deviation (SD). Statistical analysis was performed with the significance level of 0.05 using One-way ANOVA with Tukey method via OriginPro 2018 Software. All experiments were repeated three times independently.

Also, we thoroughly revised the experimental section to include all the experimental details of the research.

Comment 31. Overall, I congratulate the authors on a clever and novel approach; however, additional clarity, consistency, detail, and discussion will greatly elevate the manuscript, and should be implemented before it is accepted for publication.

Response:

We sincerely appreciate the reviewer's comments. We agree that our manuscript lacked clarity, consistency, detail, and discussion. Therefore, we revised the whole figures, main text, experimental section, and supplementary materials to include detailed information and clearer descriptions. From the reviewer's valuable comments, we could improve the quality of our manuscript by reducing inconsistency throughout the figures as well as discussing the limitation and the potential of our research.

Comment 32 ~ 35: Major comments from the reviewer's manuscript correction

Comment 32. Line 114: (Major?) issue – look into these temps and times.

Comment 33. Line 118: (Major?) issue – unless it's commented on.

Response to the comment 32 and 33:

The authors believe your concern has emerged because we missed to describe the detailed methods in the experimental section. Also, we acknowledge that the original statement was somewhat ambiguous, causing a misunderstanding about the heating protocols. Reflecting on your great comments, we thoroughly revised the manuscript to clearly deliver the details.

Related changes in the manuscript:

(Line 143-152)

As shown in Fig. 2b, the temperature responsiveness of the SFC was investigated by heating at 37 °C, 80 °C, or 150 °C up to 24 h in a closed system. In the 37 °C heating case, the SFC hardly yielded any noticeable signals. Interestingly, a weak absorbance signal started to appear after heating at 80 °C and this absorbance signal became obvious by 150 °C heating. In particular, the increasing peaks near 335 nm indicated the enhanced mobility of the furan group of furfuryl mercaptan, the Maillard reaction product. Hence, these results confirm that the SFC exhibited temperature-responsive flavoring *via* release of the Maillard reaction product. The release of the Maillard reaction product exhibited a positive correlation with the heating temperature. Moreover, the onset temperature for flavoring was 80 °C and this thermos-responsivity became saturated after 12 h (Supplementary Fig. 3).

(Line 119-120)

The intervals for heating are 0 min, 10 min, 1 h, 4 h, 7 h, 12 h, and 24 h.

(Line 474-480)

The UV quartz cuvette (chamber volume 3.5 mL, path-length 10 mm) was filled with 2 mL of 1.86 μM SFC. SFC was heated at 37 °C, 80 °C, and 150 °C for 24 h by maintaining the closed

system after sealing with the polytetrafluoroethylene stopper. The UV-VIS spectra within 200-600 nm were obtained for the desired heating time points with the settings of bandwidth 1.0 nm, scan speed 240 nm/min, and data interval 1.0 nm. The peak at 335 nm was assigned to the furan-related absorbance. Here, three replication experiments ($n = 3$) were performed for each heating temperature for reliability.

Comment 34. Line 130: Major(?) issue – check if this makes sense. Just weight generally?
Look into supp and methods.

Response:

Together with your Comments 32-33, we apologize for not suggesting the experimental details in the original manuscript. We devised these experiments to evaluate the stability of SFC under cultured meat production process (*i.e.*, cell culture condition). To this end, the vials (chamber volume 5 mL, $18 \times 38 \text{ mm}^2$) were filled with 1 mL of 0.37 mM SFC (n=3) or 1 mL of pure furfuryl mercaptan (n=3). Subsequently, the vials with SFC and furfuryl mercaptan was positioned at 37 °C up to 2 weeks. In particular, to emulate the open system during cell culture procedures, the vials were not closed with a cap, and the air circulation kept sufficient. The weight variation was measured for specific time points (0 h, 7 h, 12 h, 1 d, 3 d, 5 d, 7 d, 10 d, and 14 d). The authors had the concern same as you about whether we could determine the stability of SFC only through the weight variation results. Hence, we additionally conducted the ^1H NMR studies as shown in Supplementary Fig. 4c. Remarkably, ^1H NMR spectra showed that the chemical structure of the SFC did not change during long-term incubation at 37 °C. In overall, we would like to say that not only the weight variation results but also the ^1H NMR investigation demonstrated SFC was non-volatile and stable during the cultured meat production.

Related changes in the manuscript:

(Line 153-164)

It was evaluated whether the Maillard reaction product was stably bound to the SFC during a prolonged cell cultured period at 37 °C. To this end, the SFC and pristine furfuryl mercaptan without disulfide bond were incubated at 37 °C in an open system with sufficient air circulation. As shown in Supplementary Fig. 4a-b, pure furfuryl mercaptan evaporated rapidly from day 1 of the experiment. In particular, its residual weights were 60.9% and 6.76% after 3 and 14 days

of 37 °C incubation, respectively. In contrast, SFC maintained 93.8% of its weight even at the same end point. Moreover, ¹H NMR experiments were conducted *in situ* to prove the weight variation-based finding about the stability of SFC under open system with 37 °C heating below onset point. Remarkably, ¹H NMR spectra showed that the SFC retained the features of its chemical structure during 14 days of incubation at 37 °C (Supplementary Fig. 4c). These results show that the SFC selectively released the flavor compound upon cooking and the flavor compound was non-volatile during the cultured meat production.

(Line 481-487)

To understand the long-term stability under open system, the vials (chamber volume 5 mL, 18 × 38 mm²) were filled with 1 mL of 0.37 mM SFC (n = 3) and 1 mL of pure furfuryl mercaptan (n = 3), respectively. The vials with SFC and furfuryl mercaptan was positioned at 37 °C up to 2 weeks. In particular, to emulate the open system during cell culture procedures, the vials were not closed with a cap, and the air exchange kept sufficient. The weight variation was measured for specific time points. ¹H NMR (Avance III HD 300, Bruker, Bruker Biospin, USA) spectra of residual SFC were obtained by diluting in the dimethyl sulfoxide-d₆ (10 vol%).

Comment 35. Line 408 (E-nose experiment procedure in method section) : Major – Missing several important methods:

- Stability of non-bound compounds (fig 2)
- - many others.

Response:

We appreciate the reviewer's comments. We included the information of the stability test of non-bound compounds as below:

Related changes in the manuscript:

(Line 474-480)

The UV quartz cuvette (chamber volume 3.5 mL, path-length 10 mm) was filled with 2 mL of 1.86 μ M SFC. SFC was heated at 37 °C, 80 °C, and 150 °C for 24 h by maintaining the closed system after sealing with the polytetrafluoroethylene stopper. The UV-VIS spectra within 200-600 nm were obtained for the desired heating time points with the settings of bandwidth 1.0 nm, scan speed 240 nm/min, and data interval 1.0 nm. The peak at 335 nm was assigned to the furan-related absorbance. Here, three replication experiments ($n = 3$) were performed for each heating temperature for reliability.

Also, for the e-nose experiments, we supplemented the experimental section as below:

(Line 567-580)

To compare the flavors of the experimental groups to that of beef brisket, an e-nose was used to investigate the flavor pattern of each group. Hanwoo beef brisket was purchased from SIR.LOIN, Korea. For precise comparison. Then, 0.4 g of CM-SFC, CM+SFC, CM+SFCV, and beef brisket were transferred into a 20 mL headspace vial, followed by heating at 150 °C for 5 min. Volatile and semivolatile compounds produced in the headspace of the vial were then injected into the inlet of the e-nose at an injection speed of 125 μ l/s. The injector

temperature was 200 °C and the inject time was 45 seconds. For carrier gas supply, H₂ was used for the injector as well as for the detector. Two independent chromatographic columns and two flame ionization detectors were used for compound detection. The non-polar column was MXT-5 GC metal capillary column (Restek) was used whereas the second column with medium polarity was MXT-1701 GC metal capillary column (Restek). For the detectors, flame ionization detector (FID)1 (non-polar) and FID2 (slightly polar) were used, and the detector temperature was 260 °C. The detected compounds were then analyzed using the Alpha Software (Alpha MOS, France).

(Line 581-584)

Flavor profile assignment

The specific flavor note was assigned to each detected volatile compound using the flavor ingredient library of the Flavor and Extract Manufacturers Association of the United States (FEMA) database.

Response to Reviewer #3

Reviewer(s)'s Comments to Author:

Reviewer 3.

Comments

In this manuscript the authors investigate the effect of adding Maillard reaction products into the scaffold of cultured meat on flavor release upon storage and cooking. This research is very interesting and has a very valuable aim, which is the improvement of the flavor profile of cultured meat. In my opinion it should be accepted for publication after further improvement. A general comment is on the R&D, there is little comparison with literature. Detailed comments are below, and they are all about clarity because I think the experiments/experimental design are well performed.

Response:

We sincerely appreciate all the valuable comments from the reviewer. By carefully reflecting the reviewer's comments, we have realized that many expressions, descriptions of experimental procedures, and research discussion of our manuscript need to be revised. Therefore, we fully revised all the figures, main context, experimental section, research discussion, and supplementary information to improve the clarity of our manuscript. In the revised manuscript, we have provided clearer information of the detected flavor compounds and removed exaggerated expressions regarding the results. Additionally, we have supplemented the discussion on the limitations of our study and future directions to overcome them in the revised discussion section. Throughout the reviewer's comments, we could improve the quality of our manuscript. The point-by-point response to the reviewer's comments is written below, and the related changes in the revised manuscript are presented in blue font color. Once again, we appreciate the reviewer for giving us the opportunity to revise our

manuscript.

Introduction

Comment 1. Line 55: “the amino acid types of cells” check this sentence.

Response:

We appreciate the reviewer’s comment. The sentence was removed in the revised introduction. Rather, the sentence was changed as below.

Related changes in manucript:

(Line 52-54)

The difference in amino acid profiles between in vitro tissues and traditional meat presents challenges in mimicking the Maillard flavor of traditional meat in the field of cultured meat^{1,2}.

[References]

1. Food science of animal resources, 42, 175 (2022).
2. Food Research International, 161, 111818 (2022).

Comment 2. Line 73. The author should explain exactly what is the switchable flavor compound. What do they mean with “compound”, because of this lack of explanation, the sentence is not clear.

Response:

We appreciate the reviewer’s comment. We supplemented the explanation of SFC in the revised introduction.

Related changes in manucript:

(Line 78-86)

In this study, we develop a cultured meat that generates grilled beef flavors upon cooking. In particular, we develop a switchable flavor compound (SFC) that can release the conjugated flavor group (which is the Maillard reaction compound) upon heated at the cooking temperature, 150 °C (Fig. 1a). The SFC mainly consists of flavor group and two binding groups. The flavor group is the volatile compound with meaty flavor which is furfuryl mercaptan in this research. Furfuryl mercaptan was chosen because it is known as Maillard reaction product formed from cooked beef^{1,2}. Moreover, the thiol-end group of furfuryl mercaptan enables the formation of thermo-responsive disulfide bond. The binding group of SFC is the functional group which can react with methacryloyl, the binding group of the gelatin backbone.

[References]

1. Meat Science, 107, 12-19, (2015).
2. Journal of Chromatography A, 1147, 1, 85-94, (2007).

Comment 3. Line 75. Is the gelatin methacryloyl the constituent of the scaffold? Mention it.

Response:

We appreciate the reviewer's comment. We added the sentence mentioning the constituent of the scaffold.

Related changes in manuscript:

(Line 86-89)

After introducing SFC into gelatin methacryloyl (GelMA), a three-dimensional and temperature-responsively beef flavoring scaffold is fabricated. In other words, this scaffold is composed of SFC introduced GelMA.

Comment 4. Line 80-81. “Because the SFC is selectively volatilized upon heating, CM+SFC can replicate the Maillard reaction of conventional meat.” What does mean “selectively”? and how this lead to the formation of maillard products formed in conventional meat? The term “selectively” is vague.

Response:

We appreciate the reviewer’s comment. We removed the word “selectively”. Rather, we revised such sentence as below.

Related changes in manucript:

(Line 78-81)

In this study, we develop a cultured meat that generates grilled beef flavors upon cooking. In particular, we develop a switchable flavor compound (SFC) that can release the conjugated flavor group (which is the Maillard reaction compound) upon heated at the cooking temperature, 150 °C (Fig. 1a).

Comment 5. Figure 1. the caption can be improved. For panel a, it is mentioned “a Fabrication of cultured meat with switchable flavor compound (CM+SFC). Scale bars: 100 μm .” I think the scale does not refer to the whole panel a.

Response:

We appreciate the reviewer’s comment. Fig. 1 is now fully revised as below. The confocal images with the scale bars are now removed from Fig. 1. Rather, we included the scale bar for the revised image of CM+SFC.

Revised Figure:

Fig. R1 (Fig. 1): Schematic illustration of switchable flavor system. a Illustrative description of scaffold structure conjugated with switchable flavor compound (SFC). **b** Photograph of cultured meat fabricated using the scaffold with SFC (CM+SFC). Scale bar: 8 mm. **c** Illustration explaining the mechanism of SFC system depending on temperature. **d** Classification of the flavor compounds analyzed in this study.

Comment 6. The authors need to explain here (line 64) on in the R&D the rational behind the selection of the compound furfuryl mercaptan among all the Maillard reaction products with meaty notes.

Moreover, in the paragraph 72-86 or at the beginning of the R&D they should introduce well the samples and why they worked on these samples, what was their hypothesis.

Response:

We appreciate your considerate comment. We rationally selected the furfuryl mercaptan as a representative Maillard reaction product considering its cooked beef aroma and the convenience of forming the thermo-responsive disulfide bond to its thiol end. Especially in the context of disulfide bond formation, we have broadened our thermos-responsive flavoring strategy to other thiol-end compounds: 3-mercapto-2-pentanone and 2-methyl-3-furanthiol. After fully respecting your comment, we have suggested the detailed discussion about selecting the fundamental flavor compound.

Related changes in the manuscript:

(Line 83-86)

Furfuryl mercaptan was chosen because it is known as Maillard reaction product formed from cooked beef^{1,2}. Moreover, the thiol-end group of furfuryl mercaptan enables the formation of thermo-responsive disulfide bond. The binding group of SFC is the functional group which can react with methacryloyl, the binding group of the gelatin backbone.

(Line 279-283)

3-mercapto-2-pentanone is also a sulfur compound with meat, onion-like flavors, whereas 2-methyl-3-furanthiol is known to possess fried, nut, and roasted meat-like flavors. Furthermore, the thiol-ends of 3-mercapto-2-pentanone and 2-methyl-3-furanthiol contributed to preparing the thermo-responsive disulfide linkage, identical to the thiol-end of furfuryl mercaptan.

Also, we revised the introduction to explain the samples and our hypothesis as below.

(Line 89-94)

Cultured meat with the switchable flavor compound (CM+SFC) is produced after inducing differentiation of bovine primary myoblasts in the scaffold (Fig. 1b). Because of the covalent bond between the binding group of SFC and GelMA, SFC is stably conjugated in the scaffold during cell culture condition. Then, it was hypothesized that the SFC can release the flavor group upon heating due to the dynamic disulfide exchange of flavor group. Eventually, CM+SFC can replicate the Maillard reaction of conventional meat (Fig. 1c).

[References]

1. Meat Science, 107, 12-19, (2015)
2. Journal of Chromatography A, 1147, 1, 85-94, (2007)

Results

Comment 7. Figure 2. Explain in the caption what are R1-R3. Moreover, “B Investigation of Maillard reaction product mobility from SFC upon heating.” too vague. what is panel b? what measure was done? Moreover, was the SFC in the gel?

Comment 8. Line 108-110. The authors should explicitly mention how the temperature responsivity of the SFC was measure... what was monitored?

Response to the comment 7 and 8:

The authors sincerely appreciate your helpful comments and apologize for the unfriendly readability of the original manuscript. In the revised manuscript, we thoroughly described the functional roles of R1-R3 groups, the details of UV-Vis experiments in Fig. 2b, and the covalent bond-based connection between SFC and gelatin for robust stability. We hopefully believe that the below summary could relieve your questions that originated from our insufficient explanation in the original manuscript.

SFC was designed to involve three distinct functional moieties: R₁, R₂, and R₃. In detail, R₁ and R₂ were responsible for strong binding with the gelatin matrix, while R₃ contributed to temperature-responsive flavoring function. When preparing the thermal responsive group (R₃), we introduced a disulfide bridge to the thiol end of roasted meat-flavored Maillard reaction product, furfuryl mercaptan. Fig. 2b depicted the UV-Vis spectra of SFC including two binding groups (R₁-R₂) and one flavoring group (R₃). Remarkably, the increasing peaks near 335 nm indicated the enhanced mobility of the furan group of furfuryl mercaptan. Here, we investigated the temperature responsivity of the SFC by heating at 37, 80, or 150 °C up to 24 h in a closed system. In the 37 °C heating case, the SFC hardly yielded any significant signals. Interestingly, a weak absorbance signal started to appear after heating at 80 °C and this absorbance signal became obvious by 150 °C heating. Accordingly, Fig. 2b confirms that the SFC exhibited temperature-responsive flavoring attributed to its R₃ group.

Furthermore, two methacrylate groups were connected in the positions of R₁ and R₂ for strongly binding with the gelatin-based hydrogel matrix based on the robust covalent bonds. In particular, the methacrylate of SFC's binding group was covalently connected with the methacrylate of gelatin matrix through ultraviolet (UV)-based radical generation of photoinitiator (I2959). This covalent bond-based strategy enabled the long-term stability of SFC, unlike the Gel+FM group, in which the furfuryl mercaptan was just physically mixed with the gelatin hydrogel matrix.

Related changes in the manuscript:

(Line 129-142)

The switchable flavor compound (SFC) capable of generating the Maillard reaction products in response to heating and cooking was designed. Fig. 2a shows the chemical structure of the SFC involving three distinct functional moieties (R₁, R₂, and R₃). R₁ and R₂ were responsible for robust binding with the gelatin matrix, while R₃ contributed to temperature-responsive flavoring function. In case of the thermal responsive group (R₃), a disulfide bond to the roasted meat-flavored Maillard reaction product, furfuryl mercaptan, was introduced (Supplementary Fig. 1a). In particular, the disulfide bonds exhibit temperature responsivity through the disulfide exchange process. The Raman signals at 486 cm⁻¹, 515 cm⁻¹, and 2,570 cm⁻¹ confirm the disulfide bridge formation at the thiol-end of furfuryl mercaptan (Supplementary Fig. 1b). Two methacrylate groups were connected in the positions of R₁ and R₂ for serving as binding groups with gelatin methacryloyl (Supplementary Fig. 2a). As shown in Supplementary Fig. 2b, the ¹H nuclear magnetic resonance (NMR) spectra of SFC yielded the methacrylate signals of R₁-R₂, furan signals of R₃, and urethane bond signals of R₁-R₃. This result confirmed a single SFC involved the flavor group and binding groups.

(Line 143-152)

As shown in Fig. 2b, the temperature responsiveness of the SFC was investigated by heating

at 37 °C, 80 °C, or 150 °C up to 24 h in a closed system. In the 37 °C heating case, the SFC hardly yielded any noticeable signals. Interestingly, a weak absorbance signal started to appear after heating at 80 °C and this absorbance signal became obvious by 150 °C heating. In particular, the increasing peaks near 335 nm indicated the enhanced mobility of the furan group of furfuryl mercaptan, the Maillard reaction product. Hence, these results confirm that the SFC exhibited temperature-responsive flavoring *via* release of the Maillard reaction product. The release of the Maillard reaction product exhibited a positive correlation with the heating temperature. Moreover, the onset temperature for flavoring was 80 °C and this thermos-responsivity became saturated after 12 h (Supplementary Fig. 3).

(Line 165-175)

After confirming the stability of the SFC itself, the flavor stability of the gelatin-based hydrogel containing SFC(Gel+SFC) was evaluated. The hydrogel without SFC (Gel-SFC) was also produced as a control group. Our strategy aimed to induce the strong interaction between the SFC and the scaffold, and then selectively release the SFC-derived Maillard reaction product upon heating. Accordingly, the SFC was introduced into the gelatin-based hydrogel matrix based on the robust covalent bonds. In particular, the methacrylate of SFC's binding group was connected with the methacrylate of gelatin methacryloyl through ultraviolet (UV)-based radical reaction. To demonstrate the enhanced flavor stability within the hydrogel achieved through our strategy, a hydrogel just physically mixed with pure furfuryl mercaptan (Gel+FM) was also produced. Namely, this Gel+FM hardly presents any covalent bonds between the gelatin matrix and FM.

(Line 114-128)

Fig. 2: Synthesis of SFC and thermal responsivity evaluation. **a** Chemical structure of the switchable flavor compound (SFC) involving two binding groups (R₁-R₂) with methacrylate end and one flavor group (R₃) with a thermal responsive disulfide bridge. **b** Ultraviolet-visible

(UV-Vis) spectra of SFC to monitor the mobility of furfuryl mercaptan upon heating of SFC. The peak at 335 nm indicates the mobility of the furan group of furfuryl mercaptan thermal-responsively generated from SFC. The intervals for heating are 0 min, 10 min, 1 h, 4 h, 7 h, 12 h, and 24 h. **c** Photographs of the hydrogel without SFC (Gel-SFC), hydrogel with SFC (Gel+SFC), and hydrogel mixed with pure furfuryl mercaptan (Gel+FM) with the illustrations of their network structure. Gel+SFC features the robust covalent bonds between the gelatin matrix and SFC. Meanwhile, Gel+FM exhibits the weak interaction between gelatin matrix and furfuryl mercaptan. Scale bars: 0.4 cm **d** Flavor analysis of hydrogels before and after heating at Maillard temperature, 150 °C. All samples are pre-incubated in the distilled water for 15 days. Pie chart presents the ratio of the flavor compounds classified according to the specific flavor notes. Non-flavor compounds are excluded in the pie chart and each flavor note is represented by a different color ($n = 3$).

Comment 9. Line 116. The reference to paper 21 is not clear. Why the authors refer to it?

Response:

The authors truly acknowledge your helpful opinion. To address your question first, in the original manuscript, we intended to refer to paper 21 to suggest interdisciplinary examples for releasing volatile small molecules, such as gas molecules, volatile organic compounds, and flavor compounds. Thanks to your comment, we found this section should be moved to the discussion section and dealt with in-depth for improved comprehension of readers. In particular, we broadly discussed the current approaches to utilizing the volatile small molecules, as described below.

Related changes in the manuscript:

(Line 322-335)

Volatile small molecules, representatively, gas molecules, volatile organic compounds, and flavor compounds, have attracted considerable attention in broad research and application fields: capture of environmentally impacting carbon dioxide gas¹, electrochemical production of hydrogen gas as a green energy resource², deactivation of volatile chemical warfare agents³, disease diagnosis by exhaled breath gas^{4,5} vascular disease treatment by nitric oxide gas^{6,7} and carbon monoxide gas-based gastrointestinal inflammation suppression⁸. However, given the utilization in open systems, such as air-permeable cell culture processes, these volatile small molecules suffer from their fast mass transport rate. Therefore, strategies for stable entrapment and controllable release of volatile small molecules have been developed. The emulsion and perfluorocarbon were representatives for physical retention^{9,10}. Meanwhile, because these physical capsules degrade rapidly, the selective generation of the volatile small molecules is a challenge. Although the chemical binding promises significant durability, it requires external energies (*e.g.*, proteolysis¹¹, magnetic force¹², and catalysis¹³) to trigger the volatile compound release on demand.

[References]

1. Nat. Rev. Mater., 2, 17045 (2017)
2. Nature, 612, 673-678 (2022)
3. Nat. Rev. Chem., 5, 370-387 (2021)
4. ACS Sens., 4, 268-280 (2019)
5. Nat. Nanotechnol., 15, 792-800 (2020)
6. ACS Nano, 17, 8935-8965 (2023)
7. Materials Today, 72, 57-70 (2024)
8. Sci. Transl. Med., 14, abl4135 (2022)
9. Adv. Colloid Interface. Sci., 298, 102544 (2021)
10. Mol. Pharmaceutics, 20, 3254-3277 (2023)
11. Nat. Nanotechnol., 15, 792-800 (2020)
12. Sci. Robot., 5, aaz4239 (2020)
13. J. Am. Chem. Soc., 145, 11019-11032 (2023)

Comment 10. Line 139. “which may be due to the inhibition of the polymerization of the hydrogel by furfuryl mercaptan” why the author think so? Could you add a reference.

Response:

We appreciate the reviewer’s comment. We attempted to explain the reason for the visible difference in the swollen hydrogel by the difference in polymerization degree, but there was a lack of reference. Therefore, removed such sentence. Rather we revised the description of Fig. 2c as below:

Related changes in the manuscript:

(Line 176-178)

Images of Gel-SFC, Gel+SFC, and Gel+FM can be identified with the illustration representing the structure of each group in Fig. 2c. The yellow color of Gel+FM is due to the color of furfuryl mercaptan.

Comment 11. Line 142. “after immersing them in distilled water” why at in distilled water?
What is the rational behind it?

Response:

We appreciate the reviewer’s comment. In Fig. 2d, the flavor compound in the scaffolds were analysed after immersing in distilled water for 15 days. Distilled water was replaced every two days. Because culture medium contains animal serum which is composed of various animal-derived proteins, culture medium itself can create flavor compounds when heated. The products from the Maillard reaction between animal serum and glucose in culture media might react with the flavor compounds derived from SFC, because the Maillard reaction produces various unstable and highly reactive volatile compounds¹. This reaction might converse the flavor compounds from SFC to non-flavor compounds or might create undesirable or desirable flavor compounds. In Fig. 2 and 3, it can be verified in both CM-SFC and CM+SFC that the number of detected flavor compounds are decreased after culturing cells with culture medium. This might be due to the effect of volatile compounds created from serum-containing media which remain in the cultured meat specimens. Therefore, distilled water was used to compare the flavor characteristics of Gel-SFC, Gel+SFC, and Gel+FM, excluding the effect of culture medium. Then, in Fig. 3 and Fig. 4, the flavor characteristics of specimens were compared after cell culturing with culture medium. We included this in the revised discussion section as below:

Related changes in the manuscript:

(Line 407-421)

Also, serum-free media are being actively studied in the field of cultured meat. It is anticipated that if non-animal derived nutrients capable of replacing serum in media are developed, it would be able to verify the flavor-enriching effects of the SFC system more precisely. Because serum-containing culture medium contains various animal-derived proteins and glucose, Maillard reaction may occur at 150 °C. The products from the Maillard reaction between animal

serum and glucose in culture media might react with the flavor compounds derived from SFC, because the Maillard reaction produces various unstable and highly reactive volatile compounds¹. This reaction might convert the flavor compounds from SFC to non-flavor compounds or might create undesirable or desirable flavor compounds. In Fig. 2 and 3, it can be verified in both CM-SFC and CM+SFC that the number of detected flavor compounds are decreased after culturing cells. This might be due to the effect of volatile compounds created from serum-containing media which remain in the cultured meat specimens. If serum-free medium is applied to this study, unwanted reactions between the volatile compounds from serum-containing media and the compounds from SFC would be excluded, which would help analyzing the effect of SFC system on cultured meat's flavor characteristics more precisely.

[References]

1. Trends in Food Science & Technology, 11, 9-10, 364-373, (2000)

Comment 12. Line 147. In the sentence “Then, the ratios of volatile compounds with and without flavor notes were calculated from GC–MS peak areas” what the authors mean with “with and without flavor”? do they meant before and after heat treatment or Gel+SFC and Gel+FM? Mention it.

Response:

We appreciate the reviewer’s comment. We revised the sentence to clarify the meaning.

Related changes in the manuscript:

(Line 183-190)

The flavor profiles of the detected volatile compounds were assigned by the flavor library of the Flavor and Extract Manufacturers Association (FEMA) of the United States database (Supplementary Table 1). Then, the volatile compounds with flavor profiles assigned were classified into two groups: Pleasant flavor and offensive flavor. This classification was based on the flavor classification in Fig. 1. The volatile compounds with no flavor profile are classified as non-flavor compounds. Also, the specific flavor notes which were identified by the FEMA library are shown.

Comment 13. Line 147 and 148. “pungent aroma” and “nutty aroma” were they a sensorial observation of the researcher, or detected by the GCMS and in the latter case, what compounds were those? Mention it.

Response:

The flavor characteristics of volatile compounds was analyzed by assigning the flavor notes to each volatile compounds by the Flavor and Extract Manufacturers Association of the United States (FEMA) library. By GC-MS, we obtained the concentration and chemical name of each detected volatile compound. We included this information in the revised method section. Then, each name of detected compound was searched on the flavor library in FEMA which provides the flavor information of the chemical compounds. Also, we revised the supplementary materials to identify the specific information of detected flavor compounds in the Supplementary Table 1 (Table R1) and Supplementary Table 2 (Table R2).

Related changes in the manuscript:

(Line 183-186)

The flavor profiles of the detected volatile compounds were assigned by the flavor library of the Flavor and Extract Manufacturers Association (FEMA) of the United States database (Supplementary Table 1).

(Line 581-584)

Flavor profile assignment

The specific flavor note was assigned to each detected volatile compound using the flavor ingredient library of the Flavor and Extract Manufacturers Association of the United States (FEMA) database.

Scaffold type	Temperature	Flavor description	Detected flavor compound (IUPAC name)
Gel-SFC	25 °C	Meaty, Savory	Not detected
		Almond, Roasted bread	Not detected
		Floral, Cheese, Fat	Not detected
		Fishy, Pungent, Sour	Acetone
	150 °C	Meaty, Savory	Not detected
		Almond, Roasted bread	Benzaldehyde, Benzyl alcohol (Phenylmethanol)
		Floral, Cheese, Fat	Nonanoic acid, Octanoic acid
		Fishy, Pungent, Sour	Hexanoic acid
Gel+FM	25 °C	Meaty, Savory	Furfuryl mercaptan (furan-2-ylmethanethiol), Furfural (furan-2-carbaldehyde)
		Almond, Roasted bread	Not detected
		Floral, Cheese, Fat	Not detected
		Fishy, Pungent, Sour	Not detected
	150 °C	Meaty, Savory	Furfuryl mercaptan (furan-2-ylmethanethiol), Furfural (furan-2-carbaldehyde), 2,2-(dithiodimethylene)difuran
		Almond, Roasted bread	Not detected
		Floral, Cheese, Fat	Nonanoic acid
		Fishy, Pungent, Sour	Not detected

Gel+SFC	25 °C	Meaty, Savory	Not detected
		Almond, Roasted bread	Not detected
		Floral, Cheese, Fat	Not detected
		Fishy, Pungent, Sour	Not detected
	150 °C	Meaty, Savory	Furfuryl mercaptan (furan-2-ylmethanethiol), Furfural (furan-2-carbaldehyde), 2,2-(dithiodimethylene)difuran
		Almond, Roasted bread	Benzaldehyde, Benzyl alcohol (Phenylmethanol)
		Floral, Cheese, Fat	Nonanoic acid, Heptanoic acid, Octanoic acid
		Fishy, Pungent, Sour	Not detected

Table R1 (Supplementary Table 1): Flavor compounds detected from the hydrogel without switchable flavor compound (Gel-SFC), hydrogel with SFC (Gel+SFC), and hydrogel mixed with pure furfuryl mercaptan (Gel+FM) depending on the temperature.

Cultured meat type	Flavor description	Detected flavor compound
CM-SFC	Meaty, savory	Methanethiol, 2-methylthiophene
	Floral, Creamy	Not detected

	Fruity	Carbon disulfide, n-butanol, Methyl but-2-enoate
	Fish, Pungent	Trimethylamine
CM+SFC	Meaty, savory	Methanethiol
	Floral, Creamy	Butane-2,3-dione, Pent-1-en-3-ol
	Fruity	Carbon disulfide
	Fish, Pungent	Not detected
CM+SFCV	Meaty, savory	Methanethiol, 2-methylthiophene
	Floral, Creamy	Pentanoic acid
	Fruity	Heptyl pentanoate
	Fish, Pungent	Not detected

Table R2 (Supplementary Table 2): Flavor compounds detected from the cultured meat without switchable flavor compound (CM-SFC), cultured meat with SFC (CM+SFC), and cultured meat with flavor varied SFC (CM+SFCV) by electronic nose.

Comment 14. Line 148-152. In the first sentence, the authors talk about a trend in the Gel-SFC but in the next sentence, they said that the opposite trend was observed for samples Gel-SFQ and Gel-FM. Please, clarify this.

Response:

We apologize for the confusing description. We now revised the sentence to clearly describe the results of Fig. 2d.

Related changes in the manuscript:

(Line 191-204)

Before heating, the concentration of the compounds with offensive flavor was dominant for Gel-SFC. The concentration of these compounds was decreased whereas the concentration of volatile compounds with pleasant flavor was increased after heating. Nevertheless, the offensive flavor compounds still accounted for approximately 1.4% in Gel-SFC. Also, the concentration of the flavor compounds among total volatile compounds detected from Gel-SFC was decreased upon heating. For Gel+SFC, the increase in concentration of pleasant flavor compounds after heating was confirmed which indicates that the introduction of SFC can provide pleasant flavor characteristics to the scaffold. The temperature-responsive release of the flavor compounds was only confirmed in Gel+SFC. At room temperature, no flavor compounds were detected from Gel+SFC. Upon heating at the Maillard reaction temperature, high level of the flavor compounds with pleasant flavor notes such as meat, savory, almond, roasted bread, floral, cheese and fat were detected from Gel+SFC. On the other hand, the flavor compound release before heating was detected for the Gel+FM group, indicating that flavor loss can occur during the cell culture period.

Comment 15. Line 149. What the authors mean with “the overall proportion of aroma compounds”? why “proportion” and not “concentration”?

Response:

We appreciate the reviewer’s comment. As the reviewer pointed out, “concentration” would be appropriate to be used rather than proportion. Therefore, we changed the word meaning the concentration of the flavor compounds in Fig. 2d. We appreciate the reviewer’s suggestion.

Revised Figure:

Fig. R2 (Fig. 2): Synthesis of SFC and thermal responsivity evaluation. a Chemical structure of the switchable flavor compound (SFC) involving two binding groups (R_1 - R_2) with methacrylate end and one flavor group (R_3) with a thermal responsive disulfide bridge. **b**

Ultraviolet-visible (UV-Vis) spectra of SFC to monitor the mobility of furfuryl mercaptan upon heating of SFC. The peak at 335 nm indicates the mobility of the furan group of furfuryl mercaptan thermal-responsively generated from SFC. The intervals for heating are 0 min, 10 min, 1 h, 4 h, 7 h, 12 h, and 24 h. **c** Photographs of the hydrogel without SFC (Gel-SFC), hydrogel with SFC (Gel+SFC), and hydrogel mixed with pure furfuryl mercaptan (Gel+FM) with the illustrations of their network structure. Gel+SFC features the robust covalent bonds between the gelatin matrix and SFC. Meanwhile, Gel+FM exhibits the weak interaction between gelatin matrix and furfuryl mercaptan. Scale bars: 0.4 cm **d** Flavor analysis of hydrogels before and after heating at Maillard temperature, 150 °C. All samples are pre-incubated in the distilled water for 15 days. Pie chart presents the ratio of the flavor compounds classified according to the specific flavor notes. Non-flavor compounds are excluded in the pie chart and each flavor note is represented by a different color ($n = 3$).

Related changes in the manuscript:

(Line 191-198)

Before heating, the **concentration** of the compounds with offensive flavor was dominant for Gel-SFC. The **concentration** of these compounds was decreased whereas the concentration of volatile compounds with pleasant flavor was increased after heating. Nevertheless, the offensive flavor compounds still accounted for approximately 1.4% in Gel-SFC. Also, the **concentration** of the flavor compounds among total volatile compounds detected from Gel-SFC was decreased upon heating. For Gel+SFC, the increase in **concentration** of pleasant flavor compounds after heating was confirmed which indicates that the introduction of SFC can provide pleasant flavor characteristics to the scaffold.

Comment 16. Figure 2 panel d. I agree with showing in the main manuscripts only the attributes of the detected volatiles but, I think, the authors should report the identified compounds in the supplementary material. Moreover, they need to explain the abbreviations. Each figure and table should be understood even out of the manuscript.

In all the figures: introduce abbreviation in the captions.

Response:

We appreciate the reviewer's valuable comments. We sincerely agree with the reviewer. We now added the information of the analyzed compounds in Fig. 1 (Fig. R3), Supplementary Table 1 (Table R3), and Supplementary Table 2 (Table R4) to include the chemical names and structures of the flavor compounds in Fig. 2d (Fig. R4d), Fig. 3f (Fig. R5f), and Fig. 4b (Fig. R6b). In Fig. 1, the flavor compounds were classified into groups depending on their flavor notes. Chemical names and structures are shown. Then, the flavor analysis throughout the manuscript is performed based on this flavor compounds classification. Supplementary Table 1 and Supplementary Table 2 addresses the name of the detected flavor compound in Fig. 2d and Fig. 4b, respectively.

Also, we revised the captions to explain the abbreviations.

Revised Figure and Tables

Fig. R3 (Fig. 1): Schematic illustration of switchable flavor system. a Illustrative description of scaffold structure conjugated with switchable flavor compound (SFC). **b** Photograph of cultured meat fabricated using the scaffold with SFC (CM+SFC). Scale bar: 8 mm. **c** Illustration explaining the mechanism of SFC system depending on temperature. **d** Classification of the flavor compounds analyzed in this study.

Scaffold type	Temperature	Flavor description	Detected flavor compound (IUPAC name)
Gel-SFC	25 °C	Meaty, Savory	Not detected
		Almond, Roasted bread	Not detected
		Floral, Cheese, Fat	Not detected
		Fishy, Pungent, Sour	Acetone
	150 °C	Meaty, Savory	Not detected
		Almond, Roasted bread	Benzaldehyde, Benzyl alcohol (Phenylmethanol)
		Floral, Cheese, Fat	Nonanoic acid, Octanoic acid
		Fishy, Pungent, Sour	Hexanoic acid
Gel+FM	25 °C	Meaty, Savory	Furfuryl mercaptan (furan-2-ylmethanethiol), Furfural (furan-2-carbaldehyde)
		Almond, Roasted bread	Not detected
		Floral, Cheese, Fat	Not detected
		Fishy, Pungent, Sour	Not detected
	150 °C	Meaty, Savory	Furfuryl mercaptan (furan-2-ylmethanethiol), Furfural (furan-2-carbaldehyde), 2,2-(dithiodimethylene)difuran
		Almond, Roasted bread	Not detected
		Floral, Cheese, Fat	Nonanoic acid
		Fishy, Pungent, Sour	Not detected

Gel+SFC	25 °C	Meaty, Savory	Not detected
		Almond, Roasted bread	Not detected
		Floral, Cheese, Fat	Not detected
		Fishy, Pungent, Sour	Not detected
	150 °C	Meaty, Savory	Furfuryl mercaptan (furan-2-ylmethanethiol), Furfural (furan-2-carbaldehyde), 2,2-(dithiodimethylene)difuran
		Almond, Roasted bread	Benzaldehyde, Benzyl alcohol (Phenylmethanol)
		Floral, Cheese, Fat	Nonanoic acid, Heptanoic acid, Octanoic acid
		Fishy, Pungent, Sour	Not detected

Table R3 (Supplementary Table 1): Flavor compounds detected from the hydrogel without switchable flavor compound (Gel-SFC), hydrogel with SFC (Gel+SFC), and hydrogel mixed with pure furfuryl mercaptan (Gel+FM) depending on the temperature.

Cultured meat type	Flavor description	Detected flavor compound
CM-SFC	Meaty, savory	Methanethiol, 2-methylthiophene
	Floral, Creamy	Not detected

	Fruity	Carbon disulfide, n-butanol, Methyl but-2-enoate
	Fish, Pungent	Trimethylamine
CM+SFC	Meaty, savory	Methanethiol
	Floral, Creamy	Butane-2,3-dione, Pent-1-en-3-ol
	Fruity	Carbon disulfide
	Fish, Pungent	Not detected
CM+SFCV	Meaty, savory	Methanethiol, 2-methylthiophene
	Floral, Creamy	Pentanoic acid
	Fruity	Heptyl pentanoate
	Fish, Pungent	Not detected

Table R4 (Supplementary Table 2): Flavor compounds detected from the cultured meat without switchable flavor compound (CM-SFC), cultured meat with SFC (CM+SFC), and cultured meat with flavor varied SFC (CM+SFCV) by electronic nose.

Fig. R4 (Fig. 2): Synthesis of SFC and thermal responsivity evaluation. **a** Chemical structure of the switchable flavor compound (SFC) involving two binding groups (R_1 - R_2) with methacrylate end and one flavor group (R_3) with a thermal responsive disulfide bridge. **b** Ultraviolet-visible (UV-Vis) spectra of SFC to monitor the mobility of furfuryl mercaptan upon heating of SFC. The peak at 335 nm indicates the mobility of the furan group of furfuryl mercaptan thermal-responsively generated from SFC. The intervals for heating are 0 min, 10 min, 1 h, 4 h, 7 h, 12 h, and 24 h. **c** Photographs of the hydrogel without SFC (Gel-SFC), hydrogel with SFC (Gel+SFC), and hydrogel mixed with pure furfuryl mercaptan (Gel+FM) with the illustrations of their network structure. Gel+SFC features the robust covalent bonds between the gelatin matrix and SFC. Meanwhile, Gel+FM exhibits the weak interaction between gelatin matrix and furfuryl mercaptan. Scale bars: 0.4 cm **d** Flavor analysis of

hydrogels before and after heating at Maillard temperature, 150 °C. All samples are pre-incubated in the distilled water for 15 days. Pie chart presents the ratio of the flavor compounds classified according to the specific flavor notes. Non-flavor compounds are excluded in the pie chart and each flavor note is represented by a different color ($n = 3$).

Fig. R5 (Fig. 3): Biological evaluation and flavor analysis of cell-cultured scaffold. a Illustration of the flavor enriching process of the switchable flavor compound (SFC) under cultured meat fabrication process. **b** Immunofluorescence images of proliferated myoblasts on each group. Scale bars: 100 μ m. **c** Cell viability on day 1, day 5, and day 7 using CCK-8 assay kit (mean \pm SD, $n = 3$ independent experiments, One-way ANOVA with Tukey method). Grey color indicates cultured meat without SFC (CM-SFC) and red color indicates cultured meat with SFC (CM+SFC). **d** Confocal images showing myosin heavy chain (MHC) and nuclei

immunostained with MF20 (red) and DAPI (blue), respectively. Scale bars: 100 μm . **e** Quantitative assessment of MHC amount by bovine myosin-1 enzyme linked immunosorbent assay (ELISA) (mean \pm SD, $n = 3$ independent experiments, One-way ANOVA with Tukey method). MHC amount of the CM+SFC is normalized to that of CM-SFC. **f** Assessment of volatile compounds in CM-SFC and CM+SFC after heating at 150 $^{\circ}\text{C}$ ($n = 3$).

Fig. R6 (Fig. 4): Electronic nose analysis of cultured meat specimens. a Illustration of cultured meat specimens with different scaffolds: cultured meat without SFC (CM-SFC), cultured meat with SFC (CM+SFC), and cultured meat with flavor varied SFC (CM+SFCV).

b Ratio of the flavor compounds detected from CM-SFC, CM+SFC, and CM+SFCV (mean \pm SD, $n = 3$ independent experiments, One-way ANOVA with Tukey method). Pie chart is shown to present the specific flavor profiles of each group. Flavor notes are presented with different color. **c** Principal component analysis (PCA) of the flavor compounds of each group (discrimination index = 90, $n = 3$).

Comment 17. Line 219. “which might be due to thermal degradation of GelMA” why? Add reference.

Response:

We appreciate the reviewer’s comment. We stated such sentence because benzaldehyde was detected from the Gel-SFC as well as CM-SFC. Because Gel-SFC is only composed of GelMA, we concluded that benzaldehyde is derived from GelMA. Also, GelMA hydrogel is reported to start degradation from 50 °C¹. In Fig. 2d and the supplementary Table 1, it can be verified that benzaldehyde is only detected from the heated samples at 150 °C. Therefore, we anticipated that benzaldehyde is from the thermal degradation of GelMA. However, it is still unclear whether the compound is due to the thermal degradation of GelMA since there is no research which detected benzaldehyde from thermal degradation of GelMA. Therefore, we removed such sentence in the revised manuscript.

[References]

1. Nanomaterials, 11, 3, 617 (2021)

Comment 18. Methods : no need of reference in the Methods?

Response:

We appreciate the reviewer's comment. We supplemented the reference in the method section as below:

Related changes in the manuscript:

(Line 446-447)

First, GelMA was synthesized by conjugating methacrylic anhydride to fish gelatin, referring to the previous research^{1,2}.

(Line 496-498)

Using GC–MS (Agilent 8890 GC system–Agilent 5677B MSD, Agilent Technologies), volatile and semivolatile compounds were detected for each group^{3,4}.

[References]

1. ACS Applied Polymer Materials, 2, 7, 3016-3023 (2020)
2. Nature Protocols, 11, 727-746 (2016)
3. ACS Applied Materials & Interfaces 14, 33, 38235-38245 (2022)
4. Korean Society for Biotechnology and Bioengineering Journal 33. 2 (2018)

Reviewers' Comments:

Reviewer #1:

Remarks to the Author:

Dear authors,

the paper was improved after all the reviewers comments, now it is clearly written and in good condition and can be recommended for publication.

Reviewer #2:

Remarks to the Author:

The authors have successfully addressed my primary comments, with one exception: the authors have looked into scaffold stiffness as a possible mechanism of slowing cell growth; however, the fact that stiffness could play a role does not negate the possibility that the flavor-controlling chemistry could also play a role. The authors should, somewhere in the manuscript, acknowledge the fact that further investigation is required to determine whether the slower growth in flavor-controlled scaffolds is due to scaffold features or flavor-chemistry features.

Other than that, the authors have produced a very interesting and well-communicated manuscript.

Reviewer #3:

Remarks to the Author:

the authors improved the manuscript according to the comments. I'm not sure if the terminology "offensive flavor" is the commonly used. I would call them off-flavor.

Response letter

Journal: *Nature Communications*

Manuscript ID: NCOMMS-23-63842A

Title: Flavor-Switchable Scaffold for Cultured Meat with Enhanced Aromatic Properties

Authors: Milae Lee, Woojin Choi, Jeong Min Lee, Seung Tae Lee, Won-Gun Koh, Jinkee Hong

Response to Reviewer #1

Reviewer(s)'s Comments to Author:

Reviewer 1.

Comments

Dear authors,

the paper was improved after all the reviewers comments, now it is clearly written and in good condition and can be recommended for publication.

Response:

We sincerely appreciate the reviewer for all the grateful comments that guided us to improve our manuscript. The manuscript contains clearer information. Also, the consistency of the figures are improved. These improvements are made by revising the whole manuscript followed by the reviewer's comments.

Response to Reviewer #2

Reviewer(s)'s Comments to Author:

Reviewer 2.

Comments

The authors have successfully addressed my primary comments, with one exception: the authors have looked into scaffold stiffness as a possible mechanism of slowing cell growth; however, the fact that stiffness could play a role does not negate the possibility that the flavor-controlling chemistry could also play a role. The authors should, somewhere in the manuscript, acknowledge the fact that further investigation is required to determine whether the slower growth in flavor-controlled scaffolds is due to scaffold features or flavor-chemistry features. Other than that, the authors have produced a very interesting and well-communicated manuscript.

Response: We appreciate the reviewer's comments. In the revised manuscript, we stated about the necessity of further investigation to determine the factor that slows down the cell growth in Gel+SFC (flavor-controlled scaffold). The change is made as below:

In the revised manuscript (Line 212-215): The scaffold stiffness can be one of the factors that slows down the cell growth in Gel+SFC. However, further investigation is required to determine whether the slower cell growth in Gel+SFC is due to the flavor-chemistry.

We sincerely appreciate for all the reviewer's valuable comments which helped us to improve our manuscript. The manuscript contains clearer information. Also, the consistency of the figures are improved. These improvements are made by revising the whole manuscript followed by the reviewer's comments.

Response to Reviewer #3

Reviewer(s)'s Comments to Author:

Reviewer 3.

Comments

The authors improved the manuscript according to the comments. I'm not sure if the terminology "offensive flavor" is the commonly used. I would call them off-flavor.

Response: We sincerely appreciate the reviewer's suggestion. We agree with the reviewer. The terminology "offensive flavor" is now changed to "off-flavor" in the revised manuscript. We sincerely appreciate for all the reviewer's valuable comments which helped us to improve our manuscript. The manuscript contains clearer information. Also, the consistency of the figures are improved. These improvements are made by revising the whole manuscript followed by the reviewer's comments.